# UNIFYING LOW DIMENSIONAL SPECTRA IN DEEP LEARNING

## ABSTRACT

Empirical studies have revealed low-dimensional structures in the eigenspectra of weights, Hessians, gradients, and feature vectors of deep networks, consistently observed across datasets and architectures in the overparameterized regime. In this work, we analyze deep unconstrained feature models (UFMs) to provide an analytic explanation of how these structures emerge at the layer-wise level, including the bulk–outlier Hessian spectrum and the alignment of gradient descent with the outlier eigenspace. We show that deep neural collapse underlies these phenomena, deriving explicit expressions for eigenvalues and eigenvectors of many deep learning matrices in terms of class feature means. Furthermore, we demonstrate that the full Hessian inherits its low-dimensional structure from the layer-wise Hessians, and empirically validate our theory in both UFMs and deep networks.

## 1 INTRODUCTION

Several researchers have empirically observed low-dimensional structures emerging in deep neural networks (DNNs) trained on classification tasks. Early evidence came from analyses of Hessian spectra (LeCun et al., 2012; Dauphin et al., 2014; Sagun et al., 2016; 2017; Papyan, 2019; 2020). Histogram plots of eigenvalues revealed that most cluster in a "bulk" near zero, with a few separated outliers. Notably, the number of outliers often matched the number of classes, $K$ (Sagun et al., 2016; 2017). Later, Papyan (2020) identified an additional "mini-bulk" of $K(K-1)$ outliers distinct from the main bulk, yielding a total of $K^2$ separated outliers. Similar bulk–outlier patterns have also been reported in the covariance of gradients (Jastrzebski et al., 2020), the Fisher information matrix (Li et al., 2020), the spectrum of backpropagated errors (Oymak et al., 2019), weight matrices (Mahoney & Martin, 2019), and layer-wise Hessians (Sankar et al., 2021). Another key observation is that gradients tend to align with the Hessian's top-$K$ outlier eigenspace throughout training (Gur-Ari et al., 2018; Ben Arous et al., 2024).

More recently, Papyan et al. (2020) reported related structure in feature matrices and weights, coining the term neural collapse (NC). They showed that in well-trained, overparameterized DNNs, penultimate-layer feature vectors from the same class converge to a single point—the class mean. After global centering, these class means form a simplex equiangular tight frame (ETF). Moreover, the last-layer classifier aligns with these centered means. Subsequent work has investigated whether NC extends beyond the final layer, a phenomenon termed deep neural collapse (DNC) (He & Su, 2022; Rangamani et al., 2023; Parker et al., 2023). These studies found that analogous structure emerges across intermediate layers, with deeper layers exhibiting stronger adherence.

Despite their significance, these low-dimensional phenomena have yet to be unified under a common mathematical explanation. The closest example in the literature is a comment in Papyan et al. (2020) suggesting a link between NC and Hessian spectra, but without a mathematical explanation or recognition of the necessity of DNC for spectral results. This gap is important because network Hessians remain notoriously difficult to analyze theoretically. While many works have attempted to reproduce observed Hessian properties (Choromanska et al., 2015; Pennington & Worah, 2018; Baskerville et al., 2021; Granziol, 2020; Liao & Mahoney, 2021), they capture only subsets of the full behavior. In contrast, DNC offers an intuitive and geometrically interpretable structure. Showing that DNC influences Hessian behavior would provide a new lens through which to study curvature and local geometry, making the Hessian more interpretable and linking it directly to feature separa-

tion. This, in turn, offers a richer foundation for understanding, diagnosing, and improving neural network training in real-world applications.

To address this gap, we study deep unconstrained feature models (UFMs). The UFM (Mixon et al., 2020) assumes that the network can represent data points as arbitrary feature vectors, capturing the expressiveness of modern DNNs. The deep UFM extends this framework by isolating an arbitrary number of layers from the expressive part of the network.

Although global objects such as the full Hessian and gradient remain analytically inaccessible in this model, their layer-wise counterparts for the separated layers can be computed analytically. Empirical evidence suggests that these layer-wise structures closely resemble their global analogues (Sankar et al., 2021; Ben Arous et al., 2024; Parker et al., 2023).

**Our contributions.** We show that the same low-dimensional spectral structures observed in Hessians, gradients, and other deep-learning matrices also arise in deep UFMs. We derive analytic formulas for the eigenvectors of weight matrices and layer-wise Hessians—previously unavailable even empirically—and demonstrate that these structures emerge due to DNC, with eigenvalues and eigenvectors expressed in terms of layer-wise feature means. This provides a unified mathematical explanation for many seemingly disparate empirical observations as manifestations of a single underlying structure governing features and weights. Under minimal assumptions, we further prove that the full Hessian inherits the same eigen-spectrum (up to scaling and noise) as the layer-wise Hessians. Our analysis applies to both linear and ReLU layers in the separated component of the network, and we validate the theory empirically on deep UFMs and standard deep classification networks. Taken together, these results support our central claim: DNC provides a unified explanation for a wide variety of low-dimensional spectral phenomena observed across deep learning.

**Significance.** Our work provides the first analytic explanation that simultaneously reproduces all low-dimensional spectral phenomena described by Papyan while revealing new structure in the corresponding eigenvectors. These robust low-dimensional patterns suggest that they encode fundamental aspects of deep learning, and our results move beyond the partial accounts in prior work to offer a more complete theoretical foundation. Moreover, by showing that the full Hessian inherits its structure from DNC, we provide a principled basis for more systematic design choices. In particular, our analysis clarifies that flatness and NC are not independent under standard training (Han et al., 2025), suggesting that architectures or regularization schemes can target flatness through NC.

## 1.1 RELATED WORKS

Many researchers have studied NC using UFMs (Mixon et al., 2020; Fang et al., 2021), including for multiple loss functions (Zhou et al., 2022b; Mixon et al., 2020; Han et al., 2022) and analyses of the loss landscape (Zhou et al., 2022a; Zhu et al., 2021; Ji et al., 2022). NC has also been examined for large class numbers (Jiang et al., 2023) and imbalanced data (Yang et al., 2022; Thrampoulidis et al., 2022; Hong & Ling, 2023). The influence of dataset properties has also been investigated (Hong & Ling, 2024; Kothapalli & Tirer, 2024). See Kothapalli et al. (2022) for a review. Deep UFMs have been employed to analyze DNC, primarily for mean squared error (MSE) loss (Tirer & Bruna, 2022; Súkeník et al., 2023; Súkeník et al., 2024; Dang et al., 2023). Further literature review of the UFM, including justification for its use as a model of overparameterized networks, appears in Appendix A. DNC has also been studied beyond the UFM framework (Beaglehole et al., 2024).

Hessian spectra have been analyzed using random matrix theory (Pennington & Bahri, 2017; Pennington & Worah, 2018; Baskerville et al., 2022; Liao & Mahoney, 2021; Granziol et al., 2022), spin-glass analogies (Dauphin et al., 2014; Choromanska et al., 2015; Baskerville et al., 2021), neural tangent kernel limits (Fan & Wang, 2020; Jacot et al., 2020), and decoupling conjectures (Wu et al., 2020). Some of these works have leveraged properties of the feature and weight matrices in their Hessian exploration, but they only model subsets of the full behavior and do not incorporate NC/DNC. Given the importance of DNC to our results—and the fundamental geometric component it introduces—these results are only tangentially related to our own. Gradient alignment with outlier eigenspaces has been investigated by Gur-Ari et al. (2018) and Ben Arous et al. (2024), with extensions considering training dynamics (Song et al., 2024). Spectral insights have further informed work on generalization (Li et al., 2020; Wu et al., 2017; Foret et al., 2020), optimization (Gur-Ari et al., 2018; Cosson et al., 2022; Li et al., 2022), and robustness (Zhao et al., 2020; Yao et al., 2018; Moosavi-Dezfooli et al., 2018).

## 2 BACKGROUND

We consider a classification task with $K$ classes and $n$ samples per class. We denote the $i^{\text{th}}$ data point of the $c^{\text{th}}$ class by $x_{ic} \in \mathbb{R}^{d_0}$, with corresponding one-hot encoded labels $y_c \in \mathbb{R}^K$. A deep neural network $f(x; \theta) : \mathbb{R}^{d_0} \to \mathbb{R}^K$, parameterized by $\theta \in \mathbb{R}^p$, models the relationship between training data and labels

$$f(x; \theta) = W_L \sigma(W_{L-1} \sigma(...\sigma(W_1 h(x; \bar{\theta}))...)), \qquad (1)$$

where $\theta = \{W_L, ..., W_1, \bar{\theta}\}$, with weight matrices $W_L \in \mathbb{R}^{K \times d}, W_1, ..., W_{L-1} \in \mathbb{R}^{d \times d}$. The function $h(x; \bar{\theta}) : \mathbb{R}^{d_0} \to \mathbb{R}^d$ is a highly expressive feature map, and $\sigma : \mathbb{R} \to \mathbb{R}$ is an element-wise activation function. The parameters $\theta$ are trained via a variant of gradient descent on a loss function $l$ with $L_2$ regularization

$$\min_\theta \{\mathcal{L}(\theta)\} = \min_\theta \left\{ \text{Av}_{ic}\{l(f(x_{ic}; \theta), y_c)\} + \lambda \|\theta\|_2^2 \right\},$$

where $\text{Av}_{ic}$ denotes the average over indices $i$ and $c$, and $\lambda > 0$ is a regularization coefficient. The feature vectors at each separated layer are defined as

$$h_{ic} = h(x_{ic}; \bar{\theta}), \quad h_{ic}^{(2)} = W_1 h_{ic}, \quad h_{ic}^{(l)} = W_{l-1} \sigma(h_{ic}^{(l-1)}), \text{ for } l = 3, ..., L.$$

We also define the matrices $H_l = [h_{11}^{(l)}, h_{21}^{(l)}, ..., h_{n1}^{(l)}, h_{12}^{(l)}, ..., h_{nK}^{(l)}] \in \mathbb{R}^{d \times Kn}$, whose columns are ordered by class. The class feature means $\mu_c^{(l)}$, global feature means $\mu_G^{(l)}$ and feature mean matrices $\bar{H}_l$ are given by

$$\mu_c^{(l)} = \text{Av}_i\{h_{ic}^{(l)}\}, \quad \mu_G^{(l)} = \text{Av}_c\{\mu_c^{(l)}\}, \quad \bar{H}_l = [\mu_1^{(l)}, \mu_2^{(l)}, ..., \mu_K^{(l)}] \in \mathbb{R}^{d \times K}.$$

Additionally for any vector $v$, denote its normalization by $\hat{v} = v/\|v\|_2$.

In the context of MSE loss, DNC refers to the following phenomena observed in overparameterized DNNs as training progresses (Súkeník et al., 2023).

**Definition 1** (Deep Neural Collapse). *A layer $l$ has DNC structure if the following conditions hold*

**DNC1:** *Feature vectors collapse to their class means $H_l = \bar{H}_l \otimes 1_n^T$.*

**DNC2:** *The class mean matrix forms an orthogonal frame $\bar{H}_l^T \bar{H}_l \propto I_K$.*

**DNC3:** *The rows of the weight matrix $W_l$ are either 0 or colinear with one of the columns of $\bar{H}_l$.*

**Deep Learning Spectra:** The Hessian is defined as $\text{Hess}(\theta) = \nabla_\theta \nabla_\theta^T \mathcal{L}$. Histograms of its eigenspectrum consistently reveal that most eigenvalues cluster near zero, with a small number of large outliers. To investigate this, (Papyan, 2020) applied the Gauss-Newton decomposition, which splits the Hessian into the Fisher information matrix $G$ and a residual matrix $E$. Papyan (2020) provided empirical evidence that $E$ does not contribute to the outliers. He further showed that $G$ exhibits cross-class structure, meaning it can be expressed in the following form

$$G = \sum_{i,c,c'} w_{icc'} g_{icc'} g_{icc'}^T, \quad i \in \{1, ..., n\}, \quad c, c' \in \{1, ..., K\} \qquad (2)$$

where $w_{icc'}$ are non-negative scalars, and $g_{icc'} \in \mathbb{R}^p$ are extended gradients. By analyzing the log-log spectrum, Papyan observed that the Hessian has $K^2$ outliers, separating into $K$ large outliers and $K(K-1)$ smaller ones that form a so-called "mini-bulk". To further dissect this structure, he introduced the decomposition

$$G = G_{\text{class}} + G_{\text{cross}} + G_{\text{within}}, \qquad (3)$$

where $G_{\text{within}}, G_{\text{cross}}$ represent the covariances with respects to the $i$ and $c'$ indices, respectively, and $G_{\text{class}}$ is the second moment matrix with respects to the $c$ index. Successive subtraction of these components revealed that $G_{\text{class}}$ gives rise to the $K$ large outliers, $G_{\text{cross}}$ to the $K(K-1)$ mini-bulk, and $G_{\text{within}}$ to the bulk at zero.

**Gradient Descent Alignment:** Define the gradient of the loss as $g(\theta) = \nabla_\theta \mathcal{L}$. we can project onto the top $K$ eigenspace of $\text{Hess}(\theta)$, producing the vector $g_{\text{top}}$. To quantify alignment with the top eigenspace, we define the projection proportion $f_{\text{top}} = \|g_{\text{top}}\|_2^2 / \|g\|_2^2$.

Gur-Ari et al. (2018) observed that this proportion rapidly approaches one during training, indicating that the gradient becomes increasingly aligned with the top eigenspace. Moreover, they showed that this top eigenspace remains stable after the initial phases of training.

**The Deep Unconstrained Feature Model:** To define the deep UFM, we approximate the feature map $h(x; \bar{\theta})$ as being capable of mapping the training data to arbitrary points in feature space, treating the feature vectors $h_{ic}$ as freely optimized variables. Using MSE loss, the objective becomes

$$\mathcal{L} = \frac{1}{2Kn}\|W_L\sigma(W_{L-1}\sigma(...W_2\sigma(W_1 H_1)...)) - Y\|_F^2 + \frac{1}{2}\lambda\sum_{l=1}^{L}\|W_l\|_F^2 + \frac{1}{2}\lambda\|H_1\|_F^2. \quad (4)$$

Here we use either ReLU or identity activations, and $Y = I_K \otimes 1_n^T$ is the label matrix, where $1_n$ is the $n$-dimensional all-ones vector, and $\otimes$ denotes the Kronecker product. The regularization coefficient $\lambda > 0$ is applied uniformly to all parameters. Note that, due to the UFM approximation, weight decay is also applied to $H_1$.

Dang et al. (2023) showed that DNC is optimal with linear activations in the separated layers when $d \geq K$ and the regularization parameter $\lambda$ lies below a threshold $\lambda_0$ (given in equation 17, Appendix C.7). This condition simply reflects that if $\lambda$ is too large, the zero solution becomes optimal. By contrast, Súkeník et al. (2024) found that DNC is not globally optimal in the ReLU case. Nevertheless, they observed that DNC often emerges under certain hyperparameter regimes, suggesting an inherent bias toward DNC. This is consistent with empirical evidence that real neural networks frequently exhibit both NC and DNC. A full literature review of the UFM, including justification for its use as a model of overparameterized networks, appears in Appendix A.

## 3 LOW DIMENSIONAL STRUCTURE IN THE DEEP LINEAR UFM

We now analyze how low-dimensional structure emerges in the deep UFM. Sections 3.1–3.3 show that the layer-wise Hessians, gradients and weights exhibit low-dimensional spectral structure. These results further reveal that DNC serves as a unifying source of this structure: all such effects arise directly from the geometry of the class means. Section 3.4 demonstrates that the full-network Hessian inherits its low-dimensional structure from the layer-wise Hessians, explaining why DNC induces similar spectral behavior at the global level. Later, in Section 4, we extend these theoretical results to ReLU activations. Additional results on the covariance of gradients and the backpropagation error matrix are deferred to Appendix E.

### 3.1 HESSIAN SPECTRA

The layer-wise Hessian is defined as $\text{Hess}_l = \nabla_{w_l}\nabla_{w_l}^T\mathcal{L}$, where $w_l \in \mathbb{R}^{d^2}$ denotes the flattened weights, with entries $(w_l)_{d(x-1)+y} = (W_l)_{xy}$. Since the regularization terms shift eigenvalues without affecting eigenvectors, we omit them from this definition. In Appendix C.1, We show that the layer-wise Hessian has the following Kronecker structure

$$\text{Hess}_l = A_{l+1}^T A_{l+1} \otimes \left[\text{Av}_{ic}\left\{h_{ic}^{(l)} h_{ic}^{(l)T}\right\}\right], \quad \text{where } A_{l+1} = W_L...W_{l+1}.$$

Using this structure, along with the properties of DNC, we obtain the following result.

**Theorem 1.** *Consider the deep linear UFM described in equation 4. Let the network width satisfy $d \geq K$, and consider a layer $l$ with $1 \leq l < L$. Assume further that the regularization parameter $\lambda$ satisfies the condition in equation 17. Then, at any global optimum of the loss, the layer-wise Hessian at layer $l$ has the following eigen-decomposition*

$$Hess_l = \frac{1}{K}n^{\frac{-1}{L+1}}C^{\frac{2L}{L+1}}\sum_{c,c'=1}^{K}\left[\hat{\mu}_c^{(l+1)}\otimes\hat{\mu}_{c'}^{(l)}\right]\left[\hat{\mu}_c^{(l+1)}\otimes\hat{\mu}_{c'}^{(l)}\right]^T.$$

*As a consequence, $Hess_l$ has rank $K^2$, with nonzero eigenvectors given by $\hat{\mu}_c^{(l+1)} \otimes \hat{\mu}_{c'}^{(l)}$, for $c, c' \in \{1, \ldots, K\}$. Moreover, all nonzero eigenvalues are equal to $\frac{1}{K}n^{\frac{-1}{L+1}}C^{\frac{2L}{L+1}}$. The constant $C$ is given as the larger solution to the following equation, and determines the norms of the feature means*

$$1 = C + K\lambda n^{\frac{1}{L+1}}C^{-\frac{L-1}{L+1}}, \quad \|\mu^{(l)}\|_2 = \frac{1}{\sqrt{n}}(C\sqrt{n})^{\frac{l}{L+1}}.$$

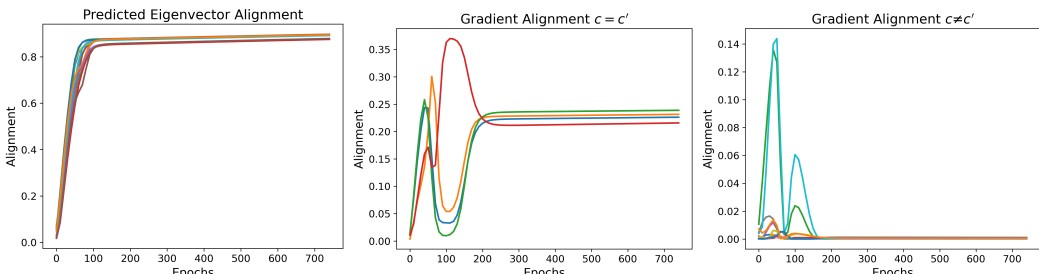

Figure 1: Training of a deep linear UFM. **Left**: Squared cosine similarity between $\mu_c^{(l+1)} \otimes \mu_{c'}^{(l)}$ and $\text{Hess}_l(\mu_c^{(l+1)} \otimes \mu_{c'}^{(l)})$. **Middle & Right**: Decomposition coefficients of $\tilde{g}^{(l)}$ in terms of the predicted eigenvectors $\mu_c^{(l+1)} \otimes \mu_{c'}^{(l)}$, measured by squared cosine similarity. Middle: $c = c'$, right: $c \neq c'$.

The proof appears in Appendix C.1. This result recovers the Hessian eigenvalue structure reported by Papyan (2020), except that here all nonzero eigenvalues are equal. In practice, DNNs do not reach the exact overparameterization and over-training limits, and the resulting noise perturbs the Hessian spectrum. The absence of a mini-bulk in the deep UFM arises from the use of MSE loss rather than cross-entropy (CE); we include the CE case in Appendix B to demonstrate this.

Crucially, the eigenvectors are built from the feature means—the defining elements of DNC. This explains why the class number $K$ consistently appears in empirical observations of Hessian spectra. Mapping data to feature means, and enforcing separation between these class means, combined with the Kronecker structure of the layer-wise Hessian, causes a low-dimensional eigenspectrum.

We next examine Papyan's decomposition, starting with the Gauss-Newton decomposition

$$\text{Hess}_l = \underbrace{\text{Av}_{ic}\left\{ \frac{\partial f(x_{ic};\theta)}{\partial w_l} \frac{\partial^2 l(z,y_c)}{\partial z^2}\Big|_{z=z_{ic}} \frac{\partial f(x_{ic};\theta)}{\partial w_l} \right\}}_{G_l} + \underbrace{\text{Av}_{ic}\left\{ \sum_{c'} \frac{\partial l(z,y_c)}{\partial z_{c'}}\Big|_{z_{ic}} \frac{\partial^2 f_{c'}(x_{ic};\theta)}{\partial w_l^2} \right\}}_{E_l}.$$

Here, $f(x_{ic};\theta) = W_L...W_1 h_{ic}$, and $z_{ic} = f(x_{ic};\theta)$. We denote the two components as $G_l$ and $E_l$. Since we use linear layers, $E_l = 0$ and thus does not contribute to the spectral outliers.

Because Papyan considered the CE loss, while we work with MSE loss, the expression for $G_l$ differs slightly from equation 2. Nevertheless, it retains cross-class structure

$$G_l = \sum_{i,c,c'} \frac{1}{Kn} v_{icc'} v_{icc'}^T,$$

where $v_{icc'} = a_{c'}^{(l+1)} \otimes h_{ic}^{(l)}$, and $(a_{c'}^{(l+1)})_x = (A_{l+1})_{c'x}$. The full derivation is given in Appendix C.2. As in Papyan (2020), we then decompose $G_l$ as

$$G_l = G_{l,\text{class}} + G_{l,\text{cross}} + G_{l,\text{within}}. \tag{5}$$

Exact expressions of these terms are in Appendix C.2, along with a proof of the following theorem.

**Theorem 2.** *Consider the deep linear UFM, under the same assumptions as Theorem 1. Then, at any global optimum of the loss, the components of the decomposition in equation 5 satisfy*

*(a)* $G_{l,within} = 0$.

*(b)* $G_{l,cross}$ *has rank* $K(K-1)$. *One choice of spanning eigenvectors for its nonzero eigenspace is* $\left( \mu_1^{(l+1)} - \mu_{c'}^{(l+1)} \right) \otimes \mu_c^{(l)}$, *for* $c' \in \{2,\ldots,K\}$, $c \in \{1,\ldots,K\}$.

*(c)* $G_{l,class}$ *has rank* $K$, *with eigenvectors for its nonzero eigenspace given by* $\mu_G^{(l+1)} \otimes \mu_c^{(l)}$, *for* $c \in \{1,\ldots,K\}$.

*All nonzero eigenvalues are equal to the value reported in Theorem 1. Furthermore, the nonzero eigenspaces of* $G_{l,class}$ *and* $G_{l,cross}$ *are orthogonal.*

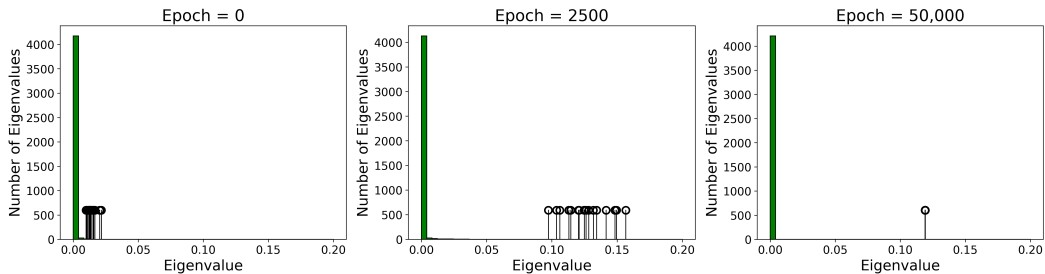

Figure 2: Histograms of the spectrum of Hess$_l$ for a deep linear UFM at an intermediate layer $l$ over a range of training epochs. The top $K^2 = 16$ outlier eigenvalues are plotted as spikes.

Thus, Papyan's decomposition is recoverable in this model: the two spectral components have the correct eigenvalue counts. Moreover, since their eigenspaces are orthogonal, subtracting either component from the layer-wise Hessian eliminates that portion of the image, reducing the rank exactly as predicted. This explains why, in Papyan's subtraction experiments, only one set of outliers shifts at a time. As before, the structure is driven by the class feature means, though it again lacks the noise and the scale separation of the mini-bulk and $K$ outliers. This is again due to the UFM model and MSE loss.

### 3.2 GRADIENT ALIGNMENT WITH OUTLIER EIGENSPACE

We now compare the layer-wise gradients to the eigenvectors of Hess$_l$. In Appendix C.3, we show that

$$g^{(l)} = \frac{\partial \mathcal{L}}{\partial w^{(l)}} = \lambda w^{(l)} + \underbrace{\mathrm{Av}_{ic}\{(A_{l+1}^T u_{ic}) \otimes h_{ic}^{(l)}\}}_{\tilde{g}^{(l)}}, \tag{6}$$

where $u_{ic} = W_L...W_1 h_{ic} - y_c$. To determine whether the gradient lies in the top eigenspace, we examine the term $\tilde{g}^{(l)}$. The following theorem describes $\tilde{g}^{(l)}$ at global optimum.

**Theorem 3.** *Consider the deep linear UFM described in equation 4 under the same assumptions as Theorem 1. At any global optimum of the loss, the quantity $\tilde{g}^{(l)}$, defined in equation 6, is given by*

$$\tilde{g}^{(l)} = \frac{1}{K}(\sqrt{n})^{\frac{-1}{L+1}} C^{\frac{L}{L+1}}(C-1) \sum_{c=1}^{K} \left( \hat{\mu}_c^{(l+1)} \otimes \hat{\mu}_c^{(l)} \right),$$

*where $C$ is the constant defined in Theorem 1. Hence, $\tilde{g}^{(l)}$ has exactly $K$ equal nonzero coefficients when expanded in the natural basis given in Theorem 1.*

The proof appears in Appendix C.3. Although the layer-wise Hessian possesses $K^2$ nonzero eigenvectors, the layer-wise gradient lies only in the span of $K$ of them when expressed in the natural feature-mean basis. This matches the empirical findings of Gur-Ari et al. (2018).

### 3.3 WEIGHT MATRICES

The next theorem characterizes the low-dimensional structure of the weight matrices at a DNC solution. It recovers the spectral results of Papyan et al. (2020) and shows that the weights are entirely determined by the feature means. A proof is provided in Appendix C.4.

**Theorem 4.** *Consider the deep linear UFM described in equation 4, under the same assumptions as Theorem 1. Then, at any global optimum of the loss, the weight matrices $W_l$ can be expressed as*

$$W_l = \frac{1}{K\lambda}(\sqrt{n})^{\frac{-1}{L+1}} C^{\frac{L}{L+1}}(1-C) \sum_{c=1}^{K} \hat{\mu}_c^{(l+1)} \hat{\mu}_c^{(l)\top}.$$

*Hence, the rank of $W_l$ is $K$, with all nonzero singular values equal to $\frac{1}{K\lambda}(\sqrt{n})^{\frac{-1}{L+1}} C^{\frac{L}{L+1}}(1-C)$. The corresponding left and right singular vectors are given by $\{\hat{\mu}_c^{(l+1)}\}_{c=1}^{K}$ and $\{\hat{\mu}_c^{(l)}\}_{c=1}^{K}$, respectively.*

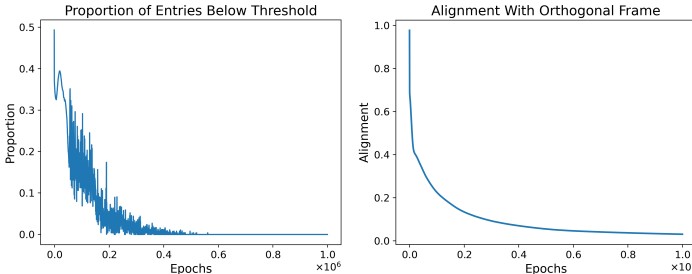

Figure 3: Early stages of training for a layer of a deep ReLU UFM. **Left**: Proportion of feature vector entries below $-10^{-6}$. **Right**: Frobenius distance of $\bar{H}_l^T \bar{H}_l$ from $I$ after normalization.

### 3.4 FULL HESSIAN SPECTRUM

We now justify why the full Hessian exhibits structure similar to that of the layer-wise Hessians. Returning to the full DNN in equation 1, suppose that the feature map $h(x; \bar{\theta})$ consists of $\bar{L}$ layers, and that the deep UFM provides a suitable approximation of this map. the entire network then has $\bar{L} + L$ many layers, and its Hessian can be expressed in block form

$$\text{Hess}(\theta) = \begin{bmatrix} \text{Hess}_{\bar{L},\bar{L}} & \text{Hess}_{\bar{L},L} \\ \text{Hess}_{\bar{L},L}^T & \text{Hess}_{L,L} \end{bmatrix},$$

where the blocks are separated by the parameters in the first $\bar{L}$ layers and last $L$ layers. We begin with the bottom-right block, which can be analyzed directly in the UFM setting.

**Theorem 5.** *Consider the deep linear UFM described in equation 4, under the same assumptions as Theorem 1. At any global optimum, the Hessian with respect to the weight layers $W_1, ..., W_L$, denoted $\text{Hess}_{L,L}$, can be written as the sum of two terms, one of which vanishes in the limit $\lambda \to 0$. The leading-order term has rank $K^2$, with all nonzero eigenvalues equal to $\frac{1}{K} n^{\frac{-1}{L+1}} C^{\frac{2L}{L+1}} L$.*

The proof appears in Appendix C.5. This result shows that, for small regularization, the bottom-right block of the Hessian exhibits the same spectral structure as the layer-wise Hessians. The next theorem extends this to the full Hessian at a DNC solution.

**Theorem 6.** *Consider the network described in equation 1, where the feature map $h(x; \bar{\theta})$ has $\bar{L}$ layers, and assume the UFM modeling assumptions hold for the last $L$ layers when $\bar{L}$ is sufficiently large. Let $\hat{\lambda}_i(M)$ denote the $i^{th}$ largest eigenvalue of the normalized matrix $M/\|M\|_F$. Further assume that all block Hessians are of the same scale. Then, in the limit $L, \bar{L} \to \infty$, such that $\bar{L}/L \to 0$, we have at any global optimum*

$$\hat{\lambda}_i(\text{Hess}(\theta)) - \hat{\lambda}_i(\text{Hess}_{L,L}) \to 0, \quad for \ i = 1, ..., K^2,$$

$$\hat{\lambda}_i(\text{Hess}(\theta)) \to 0, \quad for \ i > K^2.$$

The proof is given in Appendix C.6. This theorem shows that, under the assumptions of our model, when the network sufficiently overparameterizes the data, the full Hessian at a DNC solution approximates the spiked structure of the layer-wise Hessians. The modeling assumptions require the network to attain arbitrarily high levels of collapse in the depth limit. While this appears to hold approximately in experiments—explaining the correspondence with the observations of Papyan (2020)—it is clear from the observations of He & Su (2022) and the theoretical analysis of Súkeník et al. (2025) that regularization and optimization difficulty play a significant role. Our theorem assumes that regularization decays linearly with depth, as indicated in equation 17, but the decay rate required in real networks is likely architecture-dependent. Exploring these intricacies is an interesting direction for future work.

In real networks, noise arises from the approximations used in the theorem, but this behavior is consistent with the observations of Papyan (2020). We also emphasize that the theorem places minimal constraints on the Hessian blocks of the feature map. In practice, we expect approximate DNC in those layers as well, which further reinforces the spectral properties. Thus, DNC has implications for the local flatness of converged solutions that extend beyond purely layer-wise effects.

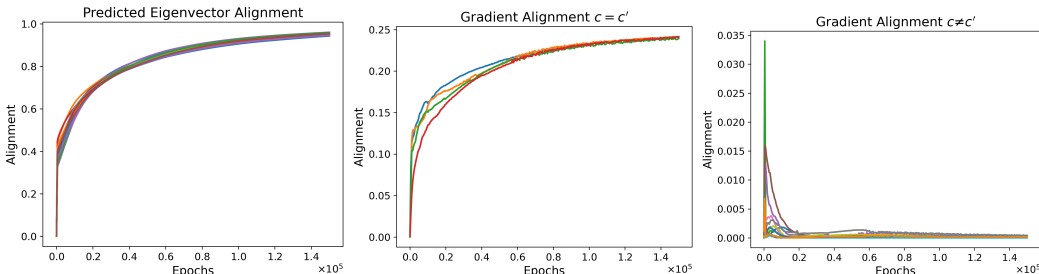

Figure 4: Early stages of training for a deep ReLU UFM. **Left**: Squared cosine similarity between $\mu_c^{(l+1)} \otimes \mu_{c'}^{(l)}$ and $\text{Hess}_l(\mu_c^{(l+1)} \otimes \mu_{c'}^{(l)})$. **Middle & Right**: Decomposition coefficients of $\tilde{g}^{(l)}$ in terms of the predicted eigenvectors $\mu_c^{(l+1)} \otimes \mu_{c'}^{(l)}$, measured by squared cosine similarity. The middle panel corresponds to $c = c'$, and the right panel to $c \neq c'$.

## 4 LOW DIMENSIONAL STRUCTURE IN THE DEEP ReLU UFM

We now turn to the deep UFM with ReLU activations. Súkeník et al. (2024) recently showed that DNC is not a global optimum in this case. Nevertheless, we analyze its implications due to the prevalence of DNC in real networks. We will show that the previous theorems continue to hold for the best-performing DNC solutions, which we find emerge over long timescales.

The loss function remains as in equation 4, but with $\sigma(x) = x1(x \geq 0)$. For this model, it is helpful to define the matrices $\tilde{W}_{l,ic}$ and $\tilde{A}_{l,ic}$ as

$$(\tilde{W}_{l,ic})_{xy} = (W_l)_{xy}\sigma'(h_{ic}^{(l)})_y, \quad \tilde{A}_{l,ic} = (\tilde{W}_{L,ic})(\tilde{W}_{L-1,ic})...(\tilde{W}_{l,ic}),$$

where $\sigma'(x) = 1(x \geq 0)$ is the derivative of the ReLU function, applied element-wise. Using these definitions, Appendix D.1 shows that for $1 \leq l < L$, the layer-wise Hessians take the form

$$\text{Hess}_l = \text{Av}_{ic}\Big\{ (\tilde{A}_{l+1,ic}^T \tilde{A}_{l+1,ic}) \otimes \sigma(h_{ic}^{(l)})\sigma(h_{ic}^{(l)})^T \Big\}.$$

We now examine this object under DNC-structured solutions. Numerically, we find that when DNC solutions arise in the ReLU case, the pre-ReLU feature matrices evolve rapidly to have nonnegative entries. This is intuitive: applying ReLU reduces the Frobenius norm of matrices with negative entries, and a lower Frobenius norm requires the scales of the weight matrices to increase to maintain the same fit loss, which increases the overall loss due to the regularization.

At early stages, these DNC solutions resemble global minima of the linear case, modified by a rank-one update to enforce non-negativity. As training continues, this update decays slowly to zero, and the final solution corresponds to a global minimum of the linear model, but with all feature matrices nonnegative. Motivated by this, we define DNC in the deep ReLU UFM as follows.

**Definition 2** (DNC Structure in the Deep ReLU UFM). *A DNC solution in this setting is any solution* $(W_L^*, \ldots, W_1^*, H_1^*)$ *that is a global minimum of the corresponding linear model, and additionally satisfies the following non-negativity condition*

$$\sigma\left(h_{ic}^{(l)}\right) = h_{ic}^{(l)}, \quad \forall l = 2, \ldots, L, \quad \forall i = 1, \ldots, n, \quad \forall c = 1, \ldots, K.$$

Using this definition, we have $(\tilde{W}_{l,ic}) = W_l$, and consequently $\tilde{A}_{l+1,ic} = A_{l+1}$. Moreover, since the activation functions have no effect at a DNC solution, the feature vectors $h_{ic}^{(l)}$ coincide with those of the linear model. This allows Theorems 2–4 to extend to the ReLU setting.

**Theorem 7.** *Consider the deep ReLU UFM described in equation 4. Suppose the network width satisfies $d \geq K$, and the regularization parameter $\lambda$ satisfies equation 17. Let the parameter values $(W_L^*, \ldots, W_1^*, H_1^*)$ have DNC structure as in Definition 2. Then the conclusions of Theorems 1–4 regarding the layer-wise Hessian, gradients, and weight matrices also hold in this nonlinear setting.*

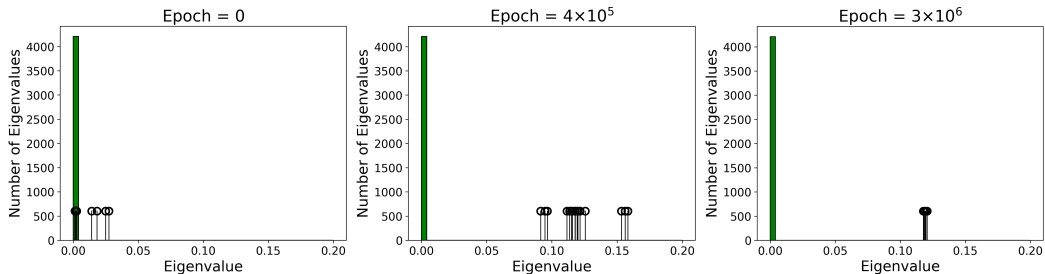

Figure 5: Histograms of the spectrum of $\text{Hess}_l$ for a deep ReLU UFM at an intermediate layer $l$ over a range of training epochs. The top $K^2 = 16$ outlier eigenvalues are plotted as spikes.

The proof appears in Appendix D.1. This result requires defining $\sigma'(0) = 1$. Without this, the minimum DNC solutions can still be expressed with linear networks, but the Hessians are not well defined due to non-differentiability at zero. We set $\sigma'(0) = 1$ to show that this issue merely obscures the underlying structure, and the same Hessians can be recovered with this minimal convention.

To support our DNC definition, we show that among the set of solutions satisfying the DNC properties identified by Papyan — and adopted as the definition by Súkeník et al. (2024) — the ones that minimize the loss are exactly those satisfying Definition 2. The proof is in Appendix D.2.

**Theorem 8.** *Consider the deep ReLU UFM described in equation 4 under the same assumptions as Theorem 7. Define the matrices $\Lambda_l = \sigma(H_l)$, for $l = 2, \ldots, L$. Consider parameter matrices $(W_L, \ldots, W_1, H_1)$ satisfying the following properties for $l = 1, \ldots L$*

*DNC1:* $H_l = \bar{H}_l \otimes 1_n^\top, \quad \Lambda_l = \bar{\Lambda}_l \otimes 1_n^\top.$

*DNC2:* $\bar{H}_l^\top \bar{H}_l \propto I_K, \quad \bar{\Lambda}_l^\top \bar{\Lambda}_l \propto I_K.$

*DNC3: The rows of the weight matrix $W_l$ are linear combinations of the columns of $\bar{H}_l$.*

*In addition, assume the network output $Z = W_L \sigma (\ldots W_2 \sigma(W_1 H_1) \ldots)$ aligns with an orthogonal frame, meaning $Z \propto I_K \otimes 1_n^\top$. Then the loss-minimizing solutions among this class are precisely the global minima of the linear model that also satisfy $H_l = \sigma(H_l)$ for all $l = 2, \ldots, L$.*

## 5 NUMERICAL EXPERIMENTS

We now empirically validate our results. Full training details, as well as additional experiments evaluating our theory on deep UFMs and real networks trained on canonical datasets are in Appendix G.

**Linear Layers:** We train a deep linear UFM using gradient descent with hyperparameters $d = 65$, $L = 4$, $K = 4$, $n = 40$, and $\lambda = 5 \times 10^{-5}$, focusing on layer $l = 3$. To measure how the vectors $\mu_c^{(l+1)} \otimes \mu_{c'}^{(l)}$ align as eigenvectors of $\text{Hess}_l$, we compute their squared cosine similarity with their images under $\text{Hess}_l$. Figure 1 shows this for each of the $K^2$ predicted top eigenvectors, showing the alignment rapidly approaches 1. Figure 2 displays the spectrum of $\text{Hess}_l$. The $K^2$ outliers converge to a common value, confirming Theorem 1. We also analyze the decomposition of the gradient $\tilde{g}^{(l)}$, defined in equation 6, in the natural basis $\{\mu_c^{(l+1)} \otimes \mu_{c'}^{(l)}\}_{c,c'=1}^K$, again using squared cosine similarity. Figure 1 shows these coefficients during training. The $c = c'$ cases converge uniformly to $1/K$, while all others decay to zero, exactly as predicted by Theorem 3.

**ReLU Layers:** We use the same hyperparameters as the linear case, and test Definition 2 by tracking the fraction of entries below $-10^{-6}$ in the features from layer $l = 3$ over the first $10^6$ epochs. Figure 3 shows that this diminishes to zero. For subsequent experiments, we zero out parameters with magnitude below $10^{-6}$ and set $\sigma'(0) = 1$. We test whether the feature means form an orthogonal frame by measuring the Frobenius distance between $\bar{H}_l^T \bar{H}_l$ and $I_K$. Figure 3 shows this decreases to zero. Together, these results support Definition 2: intermediate features become non-negative while converging to an orthogonal frame. We next analyze the same spectral properties as in the linear case. These are shown in Figures 4 and 5. They show the conclusions are unchanged.

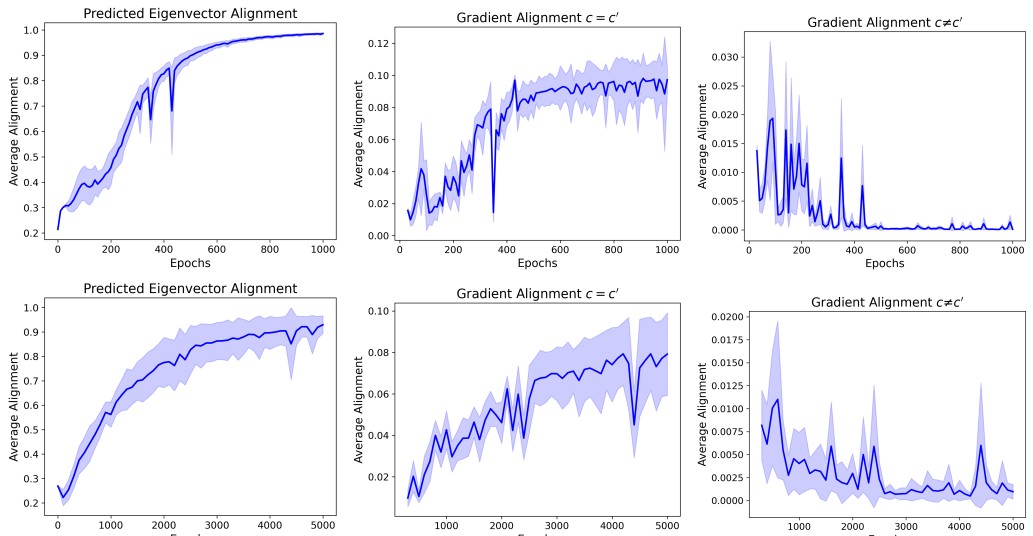

Figure 6: training dynamics of a DNN at an intermediate separated layer $l$ (top: MNIST, bottom: CIFAR-10). Each plot shows quantities averaged over $c, c'$ with one–standard-deviation error bars. **Left**: Squared cosine similarity between $\mu_c^{(l+1)} \otimes \mu_{c'}^{(l)}$ and $\mathrm{Hess}_l(\mu_c^{(l+1)} \otimes \mu_{c'}^{(l)})$. **Middle & Right**: Decomposition coefficients of $\tilde{g}^{(l)}$ in terms of the predicted eigenvectors $\mu_c^{(l+1)} \otimes \mu_{c'}^{(l)}$, measured by squared cosine similarity. The middle panel corresponds to $c = c'$, and the right panel to $c \neq c'$.

**Real networks:** We consider our theory for DNNs trained on the MNIST (Lecun et al., 1998) and CIFAR-10 (Krizhevsky & Hinton, 2009) datasets. The feature map $h(x; \bar{\theta})$ is given by a ResNet-20 architecture, followed by four linear layers of width $d = 64$. We apply UFM style regularization to the outputs of the feature map and to each layers in the linear head. The regularization parameter is set to $\lambda = 5 \times 10^{-4}$, except for the feature layer, where it is set to $\lambda_H = 1 \times 10^{-7}$. We focus on the layer $l = 3$.

Figure 6 shows the alignment of the predicted eigenvectors as true eigenvectors of the layer-wise Hessian. As in the model, this alignment quickly approaches one, indicating that the vectors evolve into eigenvectors over the course of training. The same figure also reports the gradient decomposition results, showing that for $c = c'$ we get approximately $1/K = 0.1$, whereas for $c \neq c'$ we get zero, as predicted by our theory. CIFAR-10 displays greater noise and weaker convergence than MNIST. This reflects the higher complexity; the network overparameterizes CIFAR-10 to a lesser extent, making the modeling assumptions more approximate.

## 6 CONCLUDING REMARKS

In this work, we analytically recovered the low-dimensional structure of Hessian spectra, gradients, and weight matrices in the deep UFM. We showed that DNC underlies these phenomena, deriving explicit eigenvalue and eigenvector expressions in terms of class feature means. We further provided theoretical and numerical evidence that these results extend to ReLU UFMs and to full Hessians, suggesting that linear analyses with unconstrained features offer insights into more realistic settings. Our theoretical results are then verified with experiments on canonical datasets.

Taken together, our work provides the first analytic explanation that simultaneously reproduces all low-dimensional spectral phenomena described by Papyan and others. We also crucially demonstrate that DNC causes this wide range of the low-dimensional spectra and gradient observations, including newly discovered fine-grained details such as eigenvectors. More broadly, our work establishes DNC as a unifying mechanism shaping curvature, gradient alignment, and weight structure. This perspective opens new avenues for understanding optimization landscapes, diagnosing training dynamics, and designing architectures or regularization schemes that exploit low-dimensional structure for improved performance and generalization.

## REPRODUCIBILITY STATEMENT

All experimental details required to replicate the experimental results in the main text, including explicit definitions of our metrics, are provided in Section 5 and Appendix G.1. The experiments on DNNs applied to standard datasets are fully described in Appendix G.2. Proofs of the theoretical results in the main text are contained in Appendices C and D. Theorems stated in Appendix B are proved in Appendix F, while the theoretical results in Appendix E include their proofs within the same appendix. All modeling assumptions are described in Section 2 and Appendix A.

## LARGE LANGUAGE MODEL USAGE

Large Language Models were used exclusively to improve the grammar and style of the main text. They were not used for any mathematical content.

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

## A  MOTIVATING THE DEEP UNCONSTRAINED FEATURE MODEL

In this section, we justify the use of UFMs as a modeling tool by drawing on prior empirical evaluations and examining the plausibility of the underlying modeling assumptions.

### A.1  EXPERIMENTAL EVALUATIONS OF UFMs

**The UFM reproduces previously documented phenomena:** A fundamental criterion for the usefulness of any abstract model is its ability to capture known empirical behavior. The UFM has been the focus of an extensive body of work and has repeatedly matched empirical observations across a variety of settings—including neural collapse (Mixon et al., 2020; Zhu et al., 2021; Ji et al., 2022; Lu & Steinerberger, 2022), its variants under different loss functions (Zhou et al., 2022a;b; Han et al., 2022; Behnia & Thrampoulidis, 2024), DNC (Súkeník et al., 2023; Dang et al., 2023; Tirer & Bruna, 2022; Tirer et al., 2023), and geometric structures arising in high–class-count regimes (Jiang et al., 2023).

Our paper extends this line of evidence by showing that UFM predictions align with spectral and gradient-related phenomena observed in modern deep networks, as summarized in the introduction.

**The UFM generates new predictions that are subsequently validated:** A more compelling justification for a model is its ability to produce novel predictions that are later confirmed empirically. The literature provides many such examples for UFMs, including predictions of NC behavior under class imbalance (Fang et al., 2021; Thrampoulidis et al., 2022; Hong & Ling, 2023; Dang et al., 2023), the emergence of new low-rank solutions (Súkeník et al., 2024; Garrod & Keating, 2024), extensions to regression (Andriopoulos et al., 2024) and graph neural networks (Kothapalli et al., 2023), and analogues in language models (Thrampoulidis, 2024; Zhao et al., 2024).

Our work provides another such instance: the eigenvector structure of deep-network Hessians and other deep learning matrices, as described in our paper, had not been predicted previously. The UFM yields explicit analytic formulas, and our experiments show that real networks closely follow these predictions.

These two considerations—agreement with established empirical phenomena and the ability to forecast new ones—carry more weight than direct arguments about the plausibility of modeling assumptions in isolation. Without empirical alignment, such assumptions would be of limited relevance; with alignment, they become scientifically meaningful.

### A.2  THE THEORETICAL BASIS OF THE UFM

**The unconstrained feature assumption:** This assumption corresponds to the hypothesis that, in the overparameterized limit, the learned feature representations at global minima coincide with choosing the optimal feature representations of the training data directly. Under mild assumptions on the data, universal approximation results ensure that in the overparameterization limit the feature map can express the function that places the feature vectors at an optimal location. Thus, at global minima and in the high-parameter regime, the optimal features and the features of the actual network can coincide. This is the mathematical foundation underlying the assumption. For certain architectures, a direct correspondence between UFMs and real networks has been rigorously established by Súkeník et al. (2025).

This foundation, of course, idealizes several aspects of real training. Deep networks do not typically reach exact global minima, and the overparameterized limit is never literally achieved in practice. Our use of the model takes these limitations into account: we apply the UFM only to describe the behavior of highly overparameterized networks at convergence, not to model the trajectory of training. Ultimately, the validity of such assumptions is justified not in isolation but through the empirical validations discussed above.

Crucially, the UFM is not unusual in relying on idealizations. Many influential theoretical frameworks for deep learning employ assumptions known to hold only approximately: NTK theory (Jacot et al., 2018) invokes the infinite-width limit; spin-glass analogies (Choromanska et al., 2015) predict Hessian spectra that differ markedly from those observed in practice; and linearized models (Saxe et al., 2013) sacrifice expressiveness by design. Nonetheless, these models have been valu-

able precisely because, despite their idealizations, they capture essential empirical behaviors and have yielded significant insights into learning theory. We view the UFM similarly: its scientific utility derives not from the literal correctness of its assumptions but from its consistent empirical alignment with phenomena observed in practical overparameterized networks.

**The regularization term:** For simplicity, the UFM applies regularization directly to the features produced by the feature map rather than to the network parameters themselves. Recent work has questioned this assumption (Súkeník et al., 2024; Garrod & Keating, 2024), as feature-level regularization can in some settings yield solutions that differ from DNC. Nevertheless, when training with hyperparameter choices typical in modern deep learning practice, the outcomes of UFMs and real networks continue to align closely (Súkeník et al., 2024; Garrod & Keating, 2024; Papyan et al., 2020).

## A.3 WHY THE UFM CAN EXACTLY FIT THE DATA

A defining feature of the UFM is that its feature map is assumed to be arbitrarily expressive. Consequently, the model can, in principle, map each input directly to its corresponding label and achieve zero training error. At first glance this might seem problematic. Below, we explain why this property is intentional and foundational to the design of the UFM.

**Modeling the Overparameterized Regime:** The central purpose of the UFM is to isolate and study the geometric structures that emerge in the limit of extreme overparameterization. In such regimes, real neural networks possess enough capacity to fit the training data exactly, and additional layers or parameters contribute progressively less to reducing the loss once interpolation is achievable.

The UFM captures this limiting behavior by assuming a feature map flexible enough to represent the data perfectly. This is not a flaw but an explicit modeling choice: it reflects the empirical fact that modern networks often operate far beyond the interpolation threshold. By doing so, the UFM allows us to analyze the organization of learned representations when capacity is not a limiting factor, which is precisely the setting emphasized in the original Neural Collapse (NC) work of Papyan et al. (2020) and numerous other empirical studies (Papyan, 2018; 2019; 2020). This controlled abstraction enables a clean examination of the geometric consequences of overparameterization without confounding details.

**Exact Fitting as a Standard Setting for Studying Implicit Bias:** The assumption that the model can fit the data exactly is also firmly rooted in a long tradition of theoretical work on implicit bias in optimization and deep learning. Many influential papers deliberately study training dynamics in interpolation settings—settings where the loss can be driven to zero—to reveal how gradient-based methods or depth implicitly favor certain solutions.

To highlight just a few canonical examples:

- Saxe et al. (2013) analyze how depth and gradient flow induce the sequential emergence of features in linear networks trained with MSE.

- Soudry et al. (2018) and Lyu & Li (2019) show that gradient descent on losses with exponential tails implicitly maximizes the margin in linearly separable classification.

- Arora et al. (2019) study deep matrix factorization and demonstrate that depth induces low-rank biases that improve generalization.

- Gunasekar et al. (2018) characterize how different optimization methods lead to different implicit biases in separable linear classification problems.

For a survey of this line of work, see the review by Vardi (2023).

The UFM should be understood as operating squarely within this tradition. NC and DNC manifest across many architectures and datasets, yet common loss functions contain no explicit term encouraging simplex equiangular tight frames or orthogonal frames. Their emergence instead arises as a consequence of implicit bias induced by overparameterization. The UFM provides an analytically tractable model for studying this bias in the limit of high overparameterization, and its correspondence is supported by extensive empirical evidence, as summarized previously in this appendix.

## B  THE CROSS-ENTROPY CASE

The cross-entropy version of the deep UFM was analyzed in detail by Garrod & Keating (2024). They showed that although DNC is not a global optimum of this model, it remains a critical point with a positive semi-definite Hessian, and under certain hyperparameter choices it is preferentially selected during training. Here, we examine the implications of DNC in such a model for the layer-wise Hessians of the network. The corresponding unconstrained feature model is defined as

$$\mathcal{L}(W_L, ..., W_1, H_1) = g(W_L...W_1 H_1) + \sum_{l=1}^{L} \frac{1}{2}\lambda\|W_l\|_F^2 + \frac{1}{2}\lambda\|H_1\|_F^2, \tag{7}$$

where $g$ implements the CE loss

$$g(Z) = -\frac{1}{Kn} \sum_{c=1}^{K} \sum_{i=1}^{n} \log\left(\frac{\exp((z_{ic})_c)}{\sum_{c'=1}^{K} \exp((z_{ic})_{c'})}\right), \tag{8}$$

and $z_{ic} = W_L...W_1 h_{ic}$ are the logit vectors forming the columns of the matrix $Z \in \mathbb{R}^{K \times Kn}$, using the same ordering as for the feature matrices. As before, $\lambda > 0$ is a regularization coefficient applied equally to all parameters. We also define the class feature means to be the same as in Section 2.

Within this model, Zhu et al. (2021) and Garrod & Keating (2024) clarify that the appropriate definition of DNC is the following:

**Definition 3** (DNC in the Deep CE UFM). *A parameter set $(W_L, ..., W_1, H_1)$ of the CE deep UFM exhibits DNC structure if the following hold*

$$Z = \alpha S \otimes 1_n^T, \quad \text{where } S = I_K - \frac{1}{K}1_K 1_K^T,$$

*and the constant $\alpha$ is given by the larger solution to*

$$\lambda n^{\frac{1}{L+1}} = \frac{1}{(K-1) + e^{\alpha}}\alpha^{\frac{L-1}{L+1}}. \tag{9}$$

*Furthermore, the parameter matrices admit singular value decompositions of the following form*

$$W_l = U_l \Sigma U_{l-1}^T, \quad \text{for } l = 1, ..., L-1; \quad W_L = U_L \tilde{\Sigma} U_{L-1}^T; \quad H_1 = U_0 \bar{\Sigma} V_0^T, \tag{10}$$

*where $U_L \in \mathbb{R}^{K \times K}$, $V_0 \in \mathbb{R}^{Kn \times Kn}$, $U_{L-1}, ..., U_0 \in \mathbb{R}^{d \times d}$ are all orthogonal matrices. The matrices $\Sigma \in \mathbb{R}^{d \times d}$, $\tilde{\Sigma} \in \mathbb{R}^{K \times d}$, and $\bar{\Sigma} \in \mathbb{R}^{d \times Kn}$ have their top $K \times K$ block equal to $(\alpha\sqrt{n})^{\frac{1}{L+1}} diag(1, ..., 1, 0)$, and all other entries zero.*

Garrod & Keating (2024) showed that such a solution exists and is a critical point with a positive semi-definite Hessian whenever $\lambda$ is sufficiently small and $d \geq K$. While this definition may appear detached from the original notion of DNC, they demonstrate that it naturally implies all the DNC properties. Moreover, Zhu et al. (2021) show that in the $L = 1$ case it coincides with the global minimum. From equation 9, it is also clear that as $\lambda \to 0$, the logit scale $\alpha$ diverges.

Defining the layer-wise Hessian as in the main text, we obtain the following theorem concerning its spectrum at convergence to a DNC structure.

**Theorem 9.** *Consider the deep UFM with CE loss given in equation 7, with network width $d \geq K$ and regularization parameter $\lambda$ small enough that the DNC structure of Definition 3 is a critical point. For a layer $l$ with $1 \leq l < L$, the layer-wise Hessian decomposes as the sum of two terms, one of which vanishes exponentially faster in the logit scale $\alpha \to \infty$ limit associated with small regularization. Writing $\hat{\nu}_{cc'}^{(l)} = \hat{\mu}_c^{(l+1)} \otimes \hat{\mu}_{c'}^{(l)}$, the leading-order term at DNC can be written as*

$$Hess_l = \beta\left(\frac{K-1}{K}\right)^2 \left[\sum_c \hat{\nu}_{cc}^{(l)}\hat{\nu}_{cc}^{(l)T} + \frac{1}{K}\sum_{c,c'} \hat{\nu}_{cc'}^{(l)}\hat{\nu}_{cc'}^{(l)T}\right],$$

*where*

$$\beta = \frac{e^\alpha(\alpha\sqrt{n})^{\frac{2L}{L+1}}}{n((K-1)+e^\alpha)^2}.$$

*The nonzero spectrum consists of the following $(K-1)^2$ eigenvalues:*

- *Eigenvalue $\beta$ with multiplicity $1$, eigenvector $\sum_c \hat{\nu}_{cc}^{(l)}$.*

- *Eigenvalue $\frac{K-1}{K}\beta$, multiplicity $K-1$, eigenvectors of the form $\hat{\nu}_{cc}^{(l)} - \hat{\nu}_{c'c'}^{(l)}$, $c \neq c'$, $c, c' = 1, ..., K$.*

- *Eigenvalue $\frac{1}{K}\beta$, multiplicity $K^2 - 3K + 1$, eigenvectors of the form $\hat{\nu}_{cc'}^{(l)} - \hat{\nu}_{c'c}^{(l)}$, $c \neq c'$, $c, c' = 1, ..., K$.*

This result reveals a clear separation into bulk, mini-bulk, and outliers, as observed in Papyan (2020). Specifically, the bulk corresponds to the $d^2 - (K-1)^2$ zero eigenvalues, the mini-bulk to the $K^2 - 3K + 1$ smaller but nonzero eigenvalues, and the $K$ spikes to the large eigenvalues, including a single dominant one. Thus, the separation reported by Papyan (2020) arises from the choice of loss function, and its absence in the main text is not a consequence of the UFM assumption. Apart from this distinction, the interpretation parallels that of the MSE case. The difference between $K^2$ and $(K-1)^2$ arises from the nonlinearity in Papyan's work: with ReLU activations, the network exploits the global-centering degree of freedom in the DNC definition, producing the additional $2K - 1$ spikes.

We include this section to emphasize that the absence of a mini-bulk in the main text stems from the loss function rather than from the unconstrained feature assumption. We further conjecture that analogues of Theorems 2–5 extend to this setting, though we leave this for future work.

## C  DEEP LINEAR UFM PROOFS

In this appendix, we provide proofs of the theoretical results for the deep linear UFM. Supporting lemmas appear in Appendix C.7, while the model definition is given in equation 4 and additional notation in Section 2. Most results are stated and proved for layers $1 \leq l < L$. The case $l = L$ also holds, but requires additional care with matrix dimensions since $W_L \in \mathbb{R}^{K \times d}$ rather than $\mathbb{R}^{d \times d}$. For clarity of exposition, we omit this case.

### C.1  PROOF OF THEOREM 1

We consider the Hessian with respect to the parameters of a given layer $W_l$ for $1 \leq l < L$, ignoring the regularization terms. It is convenient to express the objective function in terms of the individual training points. Defining $u_{ic} = W_L...W_1 h_{ic} - y_c$, we can write

$$\mathcal{L}(H_1, W_1, ..., W_L) = \text{Av}_{ic}\left\{\frac{1}{2}\|W_L...W_1 h_{ic} - y_c\|_2^2\right\} = \text{Av}_{ic}\left\{\frac{1}{2}u_{ic}^T u_{ic}\right\}.$$

Taking second derivatives gives

$$\frac{\partial^2 \mathcal{L}}{\partial(W_l)_{ab}\partial(W_l)_{ef}} = \text{Av}_{ic}\left\{\frac{\partial u_{ic}}{\partial(W_l)_{ab}} \cdot \frac{\partial u_{ic}}{\partial(W_l)_{ef}}\right\} + \text{Av}_{ic}\left\{\frac{\partial^2 u_{ic}}{\partial(W_l)_{ab}(W_l)_{ef}} \cdot u_{ic}\right\}.$$

The second term clearly vanishes in the linear case. We can then calculate the relevant first derivative

$$\frac{\partial(u_{ic})_x}{\partial(W_l)_{ab}} = (A_{l+1})_{xa}(h_{ic}^{(l)})_b, \quad \text{where } A_{l+1} = W_L...W_{l+1}. \tag{11}$$

Substituting this into the expression above gives

$$\frac{\partial^2 \mathcal{L}}{\partial(W_l)_{ab}\partial(W_l)_{ef}} = (A_{l+1}^T A_{l+1})_{ae}\text{Av}_{ic}\left\{(h_{ic}^{(l)})_b(h_{ic}^{(l)})_f\right\}.$$

To write this as a $d^2 \times d^2$ matrix, we flatten $W_l$ using the standard convention $(w_l)_{d(x-1)+y} = (W_l)_{xy}$. This induces the following Kronecker structure in the layer-wise Hessian

$$\text{Hess}_l = \frac{\partial^2 \mathcal{L}}{\partial^2 w_l} = A_{l+1}^T A_{l+1} \otimes \left[ \text{Av}_{ic}\left\{ h_{ic}^{(l)} h_{ic}^{(l)T} \right\} \right]. \tag{12}$$

We now assume we are at a global optimum, and so can use the properties of deep neural collapse as detailed in Lemma 1. Specifically, assume the network width obeys $d \geq K$, and assume the level of regularization obeys the condition stated in equation 17. Then we have by the DNC1 property of Lemma 1 that $h_{ic} = \mu_c$, where $\mu_c = \text{Av}_i\{h_{ic}\}$. In addition, recalling that $\mu_c^{(l)} = W_{l-1}...W_1\mu_c$, our Hessian is now

$$\text{Hess}_l = A_{l+1}^T A_{l+1} \otimes \left[ \text{Av}_c\left\{ \mu_c^{(l)} \mu_c^{(l)T} \right\} \right].$$

Also recall the matrix of class means at the $l$th layer, denoted $\bar{H}_l = [\mu_1^{(l)}, ..., \mu_K^{(l)}] \in \mathbb{R}^{d \times K}$. By Lemma 2 this matrix also forms an orthogonal frame in the sense that $\bar{H}_l^T \bar{H}_l \propto I_K$.

We begin by considering the right side of our Kronecker product. Since the class means are orthogonal, the matrix $\frac{1}{K} \sum_{c=1}^K \mu_c^{(l)} \mu_c^{(l)T}$ is already in the form of an eigen-decomposition. This matrix has rank $K$ and is simply a scaled projection onto $\text{Span}\{\mu_c^{(l)}\}_{c=1}^K$ with eigenvectors $\hat{\mu}_c^{(l)}$. In addition, since the class means have the same norm, the nonzero eigenvalues are all equal, with value $\|\mu^{(l)}\|_2^2/K$.

For the left side of our Kronecker product, note that from the DNC3 property of Lemma 1, we have $A_{l+1}^T \propto \bar{H}_{l+1}$, and so the rows $a_c^{(l+1)}$ of $A_{l+1}$ are such that $a_c^{(l+1)} = \alpha^{(l+1)}\mu_c^{(l+1)}$, for some constant $\alpha^{(l+1)}$. Hence

$$A_{l+1}^T A_{l+1} = (\alpha^{(l+1)})^2 \bar{H}_{l+1} \bar{H}_{l+1}^T = (\alpha^{(l+1)})^2 \sum_{c=1}^K \mu_c^{(l+1)} \mu_c^{(l+1)T},$$

and again this is the eigen-decomposition of this matrix, showing this matrix is also rank $K$, and has eigenvectors $\hat{\mu}_c^{(l+1)}$ with all eigenvalues equal to $(\alpha^{(l+1)})^2 \|\mu^{(l+1)}\|_2^2$.

As a consequence, our layer-wise Hessian has the following eigen-decomposition

$$\text{Hess}_l = \frac{(\alpha^{(l+1)})^2}{K} \|\mu^{(l+1)}\|_2^2 \|\mu^{(l)}\|_2^2 \sum_{c,c'=1}^K \left[ \hat{\mu}_c^{(l+1)} \otimes \hat{\mu}_{c'}^{(l)} \right] \left[ \hat{\mu}_c^{(l+1)} \otimes \hat{\mu}_{c'}^{(l)} \right]^T.$$

It remains to re-express the coefficient in terms of a constant $C$ that solves an algebraic equation in the hyper-parameters. First note, as a consequence of Lemma 1, we have that $W_L...W_1\bar{H}_1 \propto I_K$, call the constant of proportionality $C$. We also have that

$$W_L...W_1\bar{H}_1 = A_{l+1}\bar{H}_{l+1} = \alpha^{(l+1)}\bar{H}_{l+1}^T \bar{H}_{l+1} = \alpha^{(l+1)}\|\mu^{(l+1)}\|_2^2 I_K,$$

and so

$$C = \alpha^{(l+1)}\|\mu^{(l+1)}\|_2^2. \tag{13}$$

Using this to replace $\alpha^{(l+1)}$ in the eigen-decomposition of $\text{Hess}_l$, along with the result of Lemma 5 that gives the feature mean norms in terms of the constant $C$, gives the form stated in the theorem. From this the eigenvalues and eigenvectors stated in the theorem can be read off.

We only need now show that $C$ is the solution of an algebraic equation in the hyper-parameters. By Lemma 4, we have that at any optimal point the loss can be written as

$$\mathcal{L} = \frac{1}{2Kn}\|Z - Y\|_F^2 + \frac{1}{2}\lambda(L+1)\|Z\|_{S_{\frac{2}{L+1}}}^{\frac{2}{L+1}},$$

where $Z = W_L...W_1 H_1$ and $\|\cdot\|_{S_p}$ is a Schatten quasi-norm with parameter $p$. Using that $Z = CI_K \otimes 1_n^T$, and hence has singular values $C\sqrt{n}$, with multiplicity $K$, this gives that the loss at a DNC solution is given by

$$\mathcal{L} = \frac{1}{2}(1 - C)^2 + \frac{1}{2}K\lambda(L + 1)n^{\frac{1}{L+1}}C^{\frac{2}{L+1}}.$$

Since DNC represents a global minimum, the value of $C$ must minimize this loss, and so $\partial_C \mathcal{L} = 0$. This gives

$$1 = \underbrace{C + K\lambda n^{\frac{1}{L+1}}C^{-\frac{L-1}{L+1}}}_{f(C)}.$$

Wherever DNC is globally optimal this equation has a solution, and so this algebraic equation specifies the value of $C$ subject to our regularization condition. Note this equation has two solutions when $\lambda$ is small enough, this can be seen by considering the derivative of $f(C)$, which shows $f(C)$ only has one turning point and diverges as $C \to 0, \infty$. The larger solution clearly performs better on the loss, since when $C$ small $\mathcal{L} \approx 1$ whereas when $C \approx 1$, $\mathcal{L} \approx 0$, and hence the larger solution is the global minimum.

### C.2 PROOF OF THEOREM 2

Using our unconstrained feature approximation, the mapping of a data point is $f(x_{ic}; \theta) = W_L...W_1 h_{ic}$. Ignoring regularization terms, the Hessian at layer $l$ can be decomposed using the Gauss-Newton decomposition, which follows from the chain and product rules

$$= \underbrace{\mathrm{Av}_{ic}\left\{ \frac{\partial f(x_{ic};\theta)^T}{\partial w_l} \frac{\partial^2 l(z, y_c)}{\partial z^2}\bigg|_{z=z_{ic}} \frac{\partial f(x_{ic};\theta)}{\partial w_l} \right\}}_{G_l} + \underbrace{\mathrm{Av}_{ic}\left\{ \sum_{c'} \frac{\partial l(z, y_c)}{\partial z_{c'}}\bigg|_{z_{ic}} \frac{\partial^2 f_{c'}(x_{ic};\theta)}{\partial w_l^2} \right\}}_{E_l},$$

where $z_{ic} = f(x_{ic}; \theta)$. Denote the first of these by $G_l$ and the second by $E_l$. In our case, since $\partial^2_{w_l} f_{c'}(x_{ic}; \theta) = 0$ we have that $E_l = 0$, and so $\mathrm{Hess}_l = G_l$.

Noting that $\frac{\partial^2 l(z, y_c)}{\partial z^2} = I$, we can write $\mathrm{Hess}_l$ in the following form

$$\mathrm{Hess}_l = G_l = \sum_{i,c,c'} \frac{1}{Kn} v_{icc'}v_{icc'}^T, \quad \text{where } v_{icc'} = \frac{\partial f_{c'}(x_{ic};\theta)}{\partial w_l}.$$

Calculating the derivative gives $v_{icc'} = a_{c'}^{(l+1)} \otimes h_{ic}^{(l)}$, with $(a_{c'}^{(l+1)})_x = (A_{l+1})_{c'x}$ being the rows of $A_{l+1}$. Note this is a weighted second moment matrix over the objects $v_{icc'}$.

The analogy to Papyan's decomposition for our MSE case is the following: first defining the quantities

$$v_{cc'} = \sum_i \frac{1}{n} v_{icc'}, \qquad v_c = \sum_{c'} \frac{1}{K} v_{cc'}.$$

We decompose as

$$G_l = G_{l,\text{class}} + G_{l,\text{cross}} + G_{l,\text{within}},$$

where the components of the decomposition are given by

$$G_{l,\text{class}} = \sum_c v_c v_c^T,$$

$$G_{l,\text{cross}} = \sum_c G_{l,\text{cross},c}, \quad \text{where} \quad G_{l,\text{cross},c} = \sum_{c'} \frac{1}{K}(v_{cc'} - v_c)(v_{cc'} - v_c)^T,$$

$$G_{l,\text{within}} = \sum_{c,c'} \frac{1}{K}G_{l,\text{within},c,c'}, \quad \text{where} \quad G_{l,\text{within},c,c'} = \sum_i \frac{1}{n}(v_{icc'} - v_{cc'})(v_{icc'} - v_{cc'})^T.$$

We can now again assume we are at a global optimum, that $d \geq K$, and $\lambda$ obeys equation 17. We can then use the properties of DNC outlined in Lemma 1. The DNC1 property of Lemma 1 gives us

that $h_{ic} = \mu_c$, and hence $v_{icc'} = v_{cc'}$. As a consequence $G_{l,\text{within}} = 0$ and does not contribute to the outlier eigenvalues of the spectrum.

In addition using Lemma 2, and the DNC3 property of Lemma 1, gives us that $a_{c'}^{(l+1)} = \alpha^{(l+1)}\mu_{c'}^{(l+1)}$, for some constant $\alpha^{(l+1)}$. So at the optima we have

$$v_{cc'} = \alpha^{(l+1)}\mu_{c'}^{(l+1)} \otimes \mu_c^{(l)},$$

as well as

$$v_c = \alpha^{(l+1)}\mu_G^{(l+1)} \otimes \mu_c^{(l)}, \qquad v_{cc'} - v_c = \alpha^{(l+1)}(\mu_{c'}^{(l+1)} - \mu_G^{(l+1)}) \otimes \mu_c^{(l)}.$$

So, the forms of our decomposition components are given by

$$G_{l,\text{class}} = \alpha^{(l+1)2} \sum_{c=1}^{K} (\mu_G^{(l+1)} \otimes \mu_c^{(l)})(\mu_G^{(l+1)} \otimes \mu_c^{(l)})^T,$$

$$G_{l,\text{cross}} = \frac{\alpha^{(l+1)2}}{K} \sum_{c,c'=1}^{K} ([\mu_{c'}^{(l+1)} - \mu_G^{(l+1)}] \otimes \mu_c^{(l)})([\mu_{c'}^{(l+1)} - \mu_G^{(l+1)}] \otimes \mu_c^{(l)})^T.$$

This provides an eigen-decomposition of $G_{l,\text{class}}$, it is simply a scaled projection operator onto $\text{Span}\{\mu_G^{(l+1)} \otimes \mu_c^{(l)}\}_{c=1}^{K}$, and so has rank $K$. The eigenvectors are clearly given by $\hat{\mu}_G^{(l+1)} \otimes \hat{\mu}_c^{(l)}$ for $c \in \{1, ..., K\}$. Also note the eigenvalues are equal to what was reported in Theorem 1, since $\|\mu_G^{(l+1)}\|_2^2 = \frac{1}{K}\|\mu^{(l+1)}\|_2^2$, and using this with the expressions for $C$ in equation 13 and equation 18 recovers the previously stated value.

We also see $G_{l,\text{cross}}$ has image $\text{Span}\{(\mu_{c'}^{(l+1)} - \mu_G^{(l+1)}) \otimes \mu_c^{(l)}\}_{c,c'=1}^{K}$. The globally centered means $\mu_{c'}^{(l+1)} - \mu_G^{(l+1)}$ are clearly not linearly independent, since they sum to zero, and so span a $K-1$ dimensional space. They in fact form a simplex ETF, as a consequence the rank of $G_{l,\text{cross}}$ is $K(K-1)$. It is easy to verify that one choice of spanning eigenvectors for the nonzero eigenvalues is given by $(\mu_1^{(l+1)} - \mu_{c'}^{(l+1)}) \otimes \mu_c^{(l)}$, for $c' \in \{2, ..., K\}$, $c \in \{1, ..., K\}$, with eigenvalues equal to what was reported in Theorem 1.

It is also easy to see that the vectors in the two projections are orthogonal, and so $G_{l,\text{class}}$ and $G_{l,\text{cross}}$ knockout different parts of the projection in $G_l$, or equivalently $\text{Hess}_l$.

## C.3   Proof of Theorem 3

we now consider the layer-wise gradient, denoted $g_l$, for $1 \leq l < L$, at a global optimum, and show how it relates to the top eigenspace of the corresponding layer-wise Hessian. As before we have

$$\mathcal{L} = \text{Av}_{ic}\left\{\frac{1}{2}\|W_L...W_1h_{ic} - y_c\|_2^2\right\} + \frac{1}{2}\lambda\|W_l\|_F^2 + ....$$

Writing $u_{ic} = W_L...W_1h_{ic} - y_c$, and using the expression for $\partial_{W_l}u_{ic}$ from equation 11, we obtain the derivative

$$\frac{\partial\mathcal{L}}{\partial(W_l)_{ab}} = \lambda(W_l)_{ab} + \text{Av}_{ic}\left\{(A_{l+1}^T u_{ic})_a (h_{ic}^{(l)})_b\right\}.$$

After flattening $W_l$, as described in Section 3.1, we obtain

$$g^{(l)} = \frac{\partial\mathcal{L}}{\partial w^{(l)}} = \lambda w^{(l)} + \underbrace{\text{Av}_{ic}\left\{(A_{l+1}^T u_{ic}) \otimes h_{ic}^{(l)}\right\}}_{\tilde{g}^{(l)}}. \tag{14}$$

We now focus on the term $\tilde{g}^{(l)}$. We again assume we are at a global optimum, that the network width satisfies $d \geq K$, and the level of regularization $\lambda$ obeys the condition detailed in equation 17. the DNC1 property of Lemma 1 gives $h_{ic} = \mu_c$, which simplifies our quantity at this global optimum to

$$\tilde{g}_l = \text{Av}_{ic}\left\{(A_{l+1}^T u_{ic}) \otimes h_{ic}^{(l)}\right\} = \text{Av}_c\left\{(A_{l+1}^T u_c) \otimes \mu_c^{(l)}\right\},$$

where we have defined the quantity $u_c = W_L...W_1\mu_c - y_c$.

We consider the left term in our Kronecker product separately. Recall from Lemma 1 and Corollary 2 that the rows $a_c^{(l+1)}$ of $A_{l+1}$ have the form $a_c^{(l+1)} = \alpha^{(l+1)}\mu_c^{(l+1)}$. Hence

$$A_{l+1}^T = \alpha^{(l+1)}[\mu_1^{(l+1)}...\mu_K^{(l+1)}],$$

and so

$$A_{l+1}^T y_c = \alpha^{(l+1)}\mu_c^{(l+1)}.$$

Also,

$$A_{l+1}^T W_L...W_1\mu_c = A_{l+1}^T A_{l+1}\mu_c^{(l+1)}.$$

Using the form of $A_{l+1}$ We find that

$$A_{l+1}^T A_{l+1}\mu_c^{(l+1)} = \alpha^{(l+1)2}\|\mu^{(l+1)}\|_2^2\mu_c^{(l+1)},$$

and thus,

$$A_{l+1}^T u_c = \alpha^{(l+1)}\left[\alpha^{(l+1)}\|\mu^{(l+1)}\|_2^2 - 1\right]\mu_c^{(l+1)}.$$

Using the expression for the constant $\alpha^{(l+1)}$ in terms of the constant $C$ stated in equation 13, this simplifies to

$$A_{l+1}^T u_c = \frac{C}{\|\mu^{(l+1)}\|_2^2}\left[C - 1\right]\mu_c^{(l+1)}. \tag{15}$$

Substituting this into the expression of $\tilde{g}^{(l)}$ then gives, at optima,

$$\tilde{g}^{(l)} = \frac{C(C-1)}{K}\frac{\|\mu^{(l)}\|_2}{\|\mu^{(l+1)}\|_2}\sum_{c=1}^K \hat{\mu}_c^{(l+1)} \otimes \hat{\mu}_c^{(l)}.$$

Then using equation 18 to express the feature mean norms in terms of $C$ gives the final result.

Recall that the set of eigenvectors of the corresponding layer-wise Hessian at a global optima are given by $\{\hat{\mu}_{c'}^{(l+1)} \otimes \hat{\mu}_c^{(l)}\}_{c,c'=1}^K$. We see that with this natural basis the gradient lies in a $K$ dimensional subspace with equal coefficients for each of these $K$ eigenvectors.

## C.4 PROOF OF THEOREM 4

We now consider each weight matrix $W_l$ at a global optimum. Again, we assume the network width satisfies $d \geq K$ and the level of regularization $\lambda$ obeys the condition detailed in equation 17.

First note that since, at any optimum, the first order derivatives are zero, we have

$$\frac{\partial \mathcal{L}}{\partial W_l} = \lambda W_l + \frac{1}{Kn}A_{l+1}^T[W_L...W_1H_1 - Y]H_l^T = 0 \tag{16}$$

$$\implies W_l = -\frac{1}{Kn\lambda} A_{l+1}^T [W_L...W_1 H_1 - Y] H_l^T.$$

Using the DNC2 and DNC1 properties from Lemma 1, we have that, at a global optimum, $W_L...W_1 H_1 = C(I_K \otimes 1_n^T)$, and $H_l = \bar{H}_l \otimes 1_n^T$. Substituting these into the expression for $W_l$, we obtain

$$W_l = \frac{1-C}{Kn\lambda} A_{l+1}^T [I_K \otimes 1_n^T](\bar{H}_l \otimes 1_n^T)^T$$

$$= \frac{1-C}{K\lambda} A_{l+1}^T \bar{H}_l^T.$$

Using the DNC3 property from Lemma 1, which states that $A_{l+1}^T = \alpha^{(l+1)}\bar{H}^{(l+1)}$, we further simplify this expression to

$$W_l = \frac{1-C}{K\lambda} \alpha^{(l+1)} \bar{H}^{(l+1)} \bar{H}^{(l)T}.$$

Next, using the fact that $\bar{H}_{ij}^{(l)} = (\mu_j^{(l)})_i$ we can rewrite the product of the $H$ matrices as

$$W_l = \frac{1-C}{K\lambda} \alpha^{(l+1)} \sum_c \mu_c^{(l+1)} \mu_c^{(l)T}.$$

Finally, substituting the expression for $\alpha^{(l+1)}$ in terms of $C$ in equation 13 we obtain

$$W_l = \frac{C(1-C)}{K\lambda} \frac{\|\mu^{(l)}\|_2}{\|\mu^{(l+1)}\|_2} \sum_c \hat{\mu}_c^{(l+1)} \hat{\mu}_c^{(l)T}.$$

Then using equation 18 to express the feature mean norms in terms of $C$ gives the final result.

This form is precisely a singular value decomposition of $W_l$. As a consequence, $W_l$ clearly has rank $K$, and the nonzero singular values, as well as the left and right singular vectors, can be immediately identified.

### C.5    PROOF OF THEOREM 5

We now wish to consider the Hessian with respects to the parameters $(W_L, ..., W_1)$. First note, using the notation $u_{ic} = W_L...W_1 h_{ic} - y_c$ as before, we have:

$$\frac{\partial^2 \mathcal{L}}{\partial (W_l)_{xy} \partial (W_r)_{ab}} = \text{Av}_{ic} \left\{ \frac{\partial u_{ic}}{\partial (W_l)_{xy}} \cdot \frac{\partial u_{ic}}{\partial (W_r)_{ab}} + \frac{\partial^2 u_{ic}}{\partial (W_l)_{xy} \partial (W_r)_{ab}} \cdot u_{ic} \right\}.$$

The second term did not contribute to the diagonal terms of the Hessian, which we computed earlier. However for off-diagonal terms it is now nonzero. The first of these terms has a similar form to the diagonal case:

$$\text{Av}_{ic} \left\{ \frac{\partial u_{ic}}{\partial (W_l)_{xy}} \cdot \frac{\partial u_{ic}}{\partial (W_r)_{ab}} \right\} = (A_{l+1}^T A_{r+1})_{xa} \text{Av}_{ic} \left\{ (h_{ic}^{(l)})_y (h_{ic}^{(r)})_b \right\}.$$

Whilst for off-diagonal, taking wlog $r > l$, we have the second term given by

$$\text{Av}_{ic} \left\{ \frac{\partial^2 u_{ic}}{\partial (W_l)_{xy} \partial (W_r)_{ab}} \cdot u_{ic} \right\} = \text{Av}_{ic} \left\{ (A_{r+1}^T u_{ic})_a (W_{r-1}...W_{l+1})_{bx} (h_{ic}^{(l)})_y \right\}.$$

Intuitively, one expects this term to be small. This is for the same reason as argued by Papyan (2020): for well-performing networks $u_{ic} \approx 0$, and since $u_{ic}$ appears explicitly in this term, it should also be approximately zero. To verify this one can consider the second terms scale and compare it to the other term. First we assume we are at a global minimum and that the relevant conditions hold. The exact same steps used in the proof of Theorem 1 give that the first term, post flattening, is given by

$$\frac{1}{K} n^{\frac{-1}{L+1}} C^{\frac{2L}{L+1}} \sum_{c,c'} \left[ \hat{\mu}_c^{(l+1)} \otimes \hat{\mu}_{c'}^{(l)} \right] \left[ \hat{\mu}_c^{(r+1)} \otimes \hat{\mu}_{c'}^{(r)} \right]^T,$$

and note since $C \to 1$ as $\lambda \to 0$, this term does not decay for small regularization. The second term can be written, using equation 15, as (here we do not flatten it, since it does not have a nice Kronecker expression):

$$C(C-1) \frac{\|\mu^{(l)}\|_2}{\|\mu^{(r+1)}\|_2} (W_{r-1}...W_{l+1})_{bx} \mathrm{Av}_{ic} \left\{ (\hat{\mu}_c^{(r+1)})_a (\hat{\mu}_c^{(l)})_y \right\}.$$

Next using Lemma 3 to seperate the scales of the weight matrices out, and equation 18 to write the feature mean norms in terms of $C$, this becomes

$$= n^{\frac{-1}{L+1}} C^{\frac{L-1}{L+1}} (C-1)(U_{r-1} D U_l^T) \mathrm{Av}_{ic} \left\{ (\hat{\mu}_c^{(r+1)})_a (\hat{\mu}_c^{(l)})_y \right\},$$

where $D \in \mathbb{R}^{d \times d}$ has its top $K \times K$ block be $\mathrm{diag}(1, ..., 1, 0)$, and all other entries zero. It is clear to see that as $C \to 1$, which occurs as $\lambda \to 0$, this term has norm tending to zero, and so can be seen as a perturbative term. We now drop it and only work with the terms coming from the leading order component of the Hessian.

Hence, to leading order, our top $L$ layer Hessian is given by

$$\mathrm{Hess}_{1:L} = \begin{bmatrix} \mathrm{Hess}^{(1,1)} & ... & \mathrm{Hess}^{(1,L)} \\ ... & ... & ... \\ \mathrm{Hess}^{(L,1)} & ... & \mathrm{Hess}^{(L,L)} \end{bmatrix},$$

where

$$\mathrm{Hess}^{(l,r)} = \frac{1}{K} n^{\frac{-1}{L+1}} C^{\frac{2L}{L+1}} \sum_{c,c'} \left[ \hat{\mu}_c^{(l+1)} \otimes \hat{\mu}_{c'}^{(l)} \right] \left[ \hat{\mu}_c^{(r+1)} \otimes \hat{\mu}_{c'}^{(r)} \right]^T.$$

Now using that $\hat{\mu}_c^{(l+1)} = U_l D U_L^T e_c$ (this comes from Lemma 3 and $A_{l+1}^T \propto \bar{H}_{l+1}$), we have:

$$\sum_c \hat{\mu}_c^{(l+1)} \hat{\mu}_c^{(r+1)T} = U_l D U_r^T,$$

and hence

$$\mathrm{Hess}^{(l,r)} = \frac{1}{K} n^{\frac{-1}{L+1}} C^{\frac{2L}{L+1}} (U_l \otimes U_{l-1})(D \otimes D)(U_r \otimes U_{r-1})^T.$$

Consequently, defining $Q_l = U_l \otimes U_{l-1}$, we can write:

$$\mathrm{Hess}_{1:L} = \frac{1}{K} n^{\frac{-1}{L+1}} C^{\frac{2L}{L+1}} \begin{bmatrix} Q_1(D \otimes D)Q_1^T & ... & Q_1(D \otimes D)Q_L^T \\ ... & ... & ... \\ Q_L(D \otimes D)Q_1^T & ... & Q_L(D \otimes D)Q_L^T \end{bmatrix}$$

$$= \frac{1}{K} n^{\frac{-1}{L+1}} C^{\frac{2L}{L+1}} \mathrm{diag}(Q_1, ..., Q_L)(D \otimes D \otimes 1_L 1_L^T)\mathrm{diag}(Q_1, ..., Q_L)^T,$$

where the matrix $\mathrm{diag}(Q_1, ..., Q_L)$ is understood to be block diagonal, and inherits orthogonality from the orthogonal matrices $Q_l$.

Hence the matrix $\mathrm{Hess}_{1:L}$ has the same eigenvalues as the matrix $\frac{1}{K} n^{\frac{-1}{L+1}} C^{\frac{2L}{L+1}} (D \otimes D \otimes 1_L 1_L^T)$, consequently it has $K^2$ nonzero eigenvalues, each taking the value $\frac{1}{K} n^{\frac{-1}{L+1}} C^{\frac{2L}{L+1}} L$.

### C.6 PROOF OF THEOREM 6

Recall that we assume the feature map $h(x; \bar{\theta})$ has $\bar{L}$ many layers, and then denote the Hessian of the full network by

$$\mathrm{Hess}(\theta) = \begin{bmatrix} \mathrm{Hess}_{\bar{L}, \bar{L}} & \mathrm{Hess}_{\bar{L}, L} \\ \mathrm{Hess}_{L, \bar{L}} & \mathrm{Hess}_{L, L} \end{bmatrix}.$$

We can write this as

$$\mathrm{Hess}(\theta) = \underbrace{\begin{bmatrix} 0 & 0 \\ 0 & \mathrm{Hess}_{L, L} \end{bmatrix}}_{\widetilde{\mathrm{Hess}}} + \underbrace{\begin{bmatrix} \mathrm{Hess}_{\bar{L}, \bar{L}} & \mathrm{Hess}_{\bar{L}, L} \\ \mathrm{Hess}_{L, \bar{L}} & 0 \end{bmatrix}}_{\tilde{P}}.$$

We shall show, subject to the assumptions, that the second term can be viewed as a perturbation to the first matrix.

Assuming that all layers have equal scale Hessian blocks, labeling this scale $\nu$, we have:

$$\|\mathrm{Hess}_{\bar{L}, \bar{L}}\|_F^2 = \bar{L}^2 \nu^2,$$
$$\|\mathrm{Hess}_{\bar{L}, L}\|_F^2 = \bar{L} L \nu^2,$$
$$\|\mathrm{Hess}_{L, L}\|_F^2 = L^2 \nu^2,$$

and hence

$$\left\|\tilde{P}\right\|_F^2 = \left(\bar{L}^2 + 2\bar{L}L\right) \nu^2,$$

$$\left\|\widetilde{\mathrm{Hess}}\right\|_F^2 = L^2 \nu^2.$$

Recalling that $\bar{L} << L$, We see that the ratio of their Frobenius norms is

$$\frac{\left\|\tilde{P}\right\|_F^2}{\left\|\widetilde{\mathrm{Hess}}\right\|_F^2} \sim \frac{1}{L}.$$

We now apply Weyl's inequality: this states that for $A, B$ two symmetric matrices of the same dimension, and $\lambda_i(M)$ denoting the $i^{\mathrm{th}}$ largest singular value of the matrix $M$, we have

$$|\lambda_i(A + B) - \lambda_i(B)| \leq \|A\|_F.$$

Applying this with $B = \widetilde{\mathrm{Hess}}$, $A = \tilde{P}$ gives

$$\left|\lambda_i\left(\mathrm{Hess}(\theta)\right) - \lambda_i\left(\widetilde{\mathrm{Hess}}\right)\right| \leq \|\tilde{P}\|_F.$$

Now use the normalized matrices, using $\hat{\lambda}_i(M)$ to denote the eigenvalues of $M/\|M\|_F$, we have

$$\left| \left( \bar{L} + L \right) \nu \hat{\lambda}_i \left( \text{Hess}(\theta) \right) - L\nu\hat{\lambda}_i \left( \widetilde{\text{Hess}} \right) \right| \le \sqrt{\bar{L}^2 + 2\bar{L}L}\nu,$$

which reduces to

$$\left| \left( 1 + \frac{\bar{L}}{L} \right) \hat{\lambda}_i \left( \text{Hess}(\theta) \right) - \hat{\lambda}_i \left( \widetilde{\text{Hess}} \right) \right| \le \sqrt{\frac{\bar{L}^2}{L^2} + 2\frac{\bar{L}}{L}}.$$

It is then simple to see that as $L \to \infty$

$$\hat{\lambda}_i \left( \text{Hess}(\theta) \right) - \hat{\lambda}_i \left( \widetilde{\text{Hess}} \right) \to 0.$$

Since $\widetilde{\text{Hess}}$ clearly has the same top $K^2$ eigenvalues as $\text{Hess}_{L,L}$, with the remaining being zero, this gives:

$$\hat{\lambda}_i \left( \text{Hess}(\theta) \right) - \hat{\lambda}_i \left( \text{Hess}_{L,L} \right) \to 0, \quad \text{for } i = 1, ..., K^2,$$

$$\hat{\lambda}_i \left( \text{Hess}(\theta) \right) \to 0, \quad \text{for } i > K^2.$$

### C.7 SUPPORTING LEMMAS

The supporting lemmas necessary for the proofs in Appendix C are provided here.

**Lemma 1** (from Theorem 3.1 in Dang et al. (2023)). *Consider the deep linear UFM described in equation 4. Let the network width satisfy $d \ge K$, and assume the level of regularization $\lambda$ is such that*

$$0 < \sqrt[l]{Kn\lambda^{L+1}} < \frac{1}{KL^2}(L-1)^{\frac{L-1}{L}}. \tag{17}$$

*Let the set of parameter values $(W_L^*, W_{L-1}^*, \ldots, W_1^*, H_1^*)$ be a global optimizer. Then, the following properties hold for all $l = 1, ..., L$*

***DNC1:*** $H_1^* = \bar{H}^* \otimes 1_n^\top$, *where* $\bar{H}^* = [\mu_1, \ldots, \mu_K] \in \mathbb{R}^{d \times K}$.

***DNC2:*** $\bar{H}^{*\top}\bar{H}^* \propto W_L^* W_{L-1}^* \cdots \bar{H}^* \propto (W_L^* W_{L-1}^* \cdots W_l^*)(W_L^* W_{L-1}^* \cdots W_l^*)^\top \propto I_K$.

***DNC3:*** $W_L^* W_{L-1}^* \cdots W_1^* \propto \bar{H}^{*\top}, \quad W_L^* W_{L-1}^* \cdots W_l^* \propto (W_{l-1}^* \cdots W_1^* \bar{H}^*)^\top$.

**Lemma 2.** *Under the context of Lemma 1, the columns of $\bar{H}_l = W_{l-1} \cdots W_1 \bar{H}_1$ form an orthogonal frame in the sense that $\bar{H}_l^\top \bar{H}_l \propto I_K$.*

**Proof**: Although this is not explicitly stated in Lemma 1, it follows immediately from the DNC2 and DNC3 properties in the lemma statement.

**Lemma 3** (from Lemma 1 in Garrod & Keating (2024)). *Let $d \ge K$. At any optimal point of the deep linear UFM described in equation 4, there exists a singular value decomposition (SVD) of the parameter matrices of the following form*

$$W_l = U_l \Sigma U_{l-1}^\top, \quad \text{for } l = 1, \ldots, L-1,$$

$$W_L = U_L \tilde{\Sigma} U_{L-1}^\top, \quad H_1 = U_0 \bar{\Sigma} V_0^\top,$$

*where $U_L \in \mathbb{R}^{K \times K}$, $V_0 \in \mathbb{R}^{Kn \times Kn}$, $U_{L-1}, \ldots, U_0 \in \mathbb{R}^{d \times d}$ are all orthogonal matrices, $\Sigma \in \mathbb{R}^{d \times d}$, $\tilde{\Sigma} \in \mathbb{R}^{K \times d}$, and $\bar{\Sigma} \in \mathbb{R}^{d \times Kn}$ are diagonal or block-diagonal matrices whose top $K \times K$ blocks are given by $\text{diag}(\sigma_1, \ldots, \sigma_K)$, with all other entries being zero.*

Note whilst Garrod & Keating (2024) consider a slightly different model to ours, the proof hinges only on the use of $L_2$ regularization, and the exact same steps in their proof can be used for our model.

**Lemma 4.** *The loss $\mathcal{L}$ at an optimal point can be expressed entirely in terms of the matrix $Z = W_L \cdots W_1 H_1$ as*

$$\mathcal{L} = \frac{1}{2Kn}\|Z - Y\|_F^2 + \frac{1}{2}\lambda(L+1)\|Z\|_{S_{\frac{2}{L+1}}}^{\frac{2}{L+1}},$$

*where $\|\cdot\|_{S_p}$ denotes the Schatten quasi-norm with parameter $p$.*

**Proof:** consider an optimal point of the loss. Using Lemma 3, We have that if $Z = W_L...W_1 H_1$ has SVD given by $Z = U_L \text{diag}[(\sigma_1, ..., \sigma_K), 0_{K \times K(n-1)}]V_0^T$, then each parameter matrix has at most $K$ nonzero singular values given by $\sigma_1^{\frac{1}{L+1}}, ..., \sigma_K^{\frac{1}{L+1}}$. Using that the Frobenius norm is the sum of the squares of the singular values, we have

$$\mathcal{L} = \frac{1}{2Kn}\|Z - Y\|_F^2 + \frac{1}{2}\lambda(L+1)\sum_{c=1}^{K} \sigma_c^{\frac{2}{L+1}}.$$

Using the definition of the Schatten quasi norm then immediately gives the result.

**Lemma 5.** *Suppose we are at a global minimum of the deep UFM, $d \geq K$, and the regularization condition of equation 17 holds. Then the norms of the feature means can be written as*

$$\|\mu^{(l)}\|_2 = \frac{1}{\sqrt{n}}(C\sqrt{n})^{\frac{l}{L+1}}, \tag{18}$$

*where $C$ is given by the network output as $W_L...W_1 H_1 = CI \otimes 1_n^T$.*

**Proof:** Note $H_l = W_{l-1}...W_1 H_1$. By Lemma 3, we have

$$H_l = U_l \bar{\Sigma}^l V_0^T$$

Where for a DNC solution the nonzero singular values in $\bar{\Sigma}^l$ are $(C\sqrt{n})^{\frac{l}{L+1}}$ with multiplicity $K$. Taking Frobenius norm squared of both sides then gives

$$Kn\|\mu^{(l)}\|_2^2 = K(C\sqrt{n})^{\frac{2l}{L+1}}$$

Which gives

$$\|\mu^{(l)}\|_2 = \frac{1}{\sqrt{n}}(C\sqrt{n})^{\frac{l}{L+1}}$$

# D    DEEP RELU UFM PROOFS

Here we detail the forms of the layer-wise Hessians, gradients, and weights for the deep ReLU UFM. The deep ReLU UFM loss function is provided in equation 4.

## D.1    PROOF OF THEOREM 7

Here we show the implications of the first four theorems carry over to the ReLU case when DNC solutions arise. We will assume $2 \leq l < L$ for the duration of this proof. The result also holds for $l = 1$, but requires a slightly different description due to the convention of not using a nonlinearity after the matrix $H_1$. As before, when examining the Hessian, we drop the regularization terms. For this subsection we define the vectors $u_{ic}$ as

$$u_{ic} = W_L \sigma(...W_2 \sigma(W_1 h_{ic})...) - y_c,$$

so that we can express our MSE loss as

$$\mathcal{L} = \text{Av}_{ic}\left\{\frac{1}{2}u_{ic}^T u_{ic}\right\}.$$

Consequently, our layer-wise Hessian, before flattening the weights, is given by

$$\frac{\partial^2 \mathcal{L}}{\partial(W_l)_{ab}\partial(W_l)_{ef}} = \text{Av}_{ic}\left\{\frac{\partial u_{ic}}{\partial(W_l)_{ab}} \cdot \frac{\partial u_{ic}}{\partial(W_l)_{ef}} + u_{ic} \cdot \frac{\partial^2 u_{ic}}{\partial(W_l)_{ab}\partial(W_l)_{ef}}\right\}.$$

The second term is zero here since we take $\sigma$ to be ReLU, which is a piece-wise linear function. We note that

$$\frac{\partial(u_{ic})_d}{\partial(W_l)_{ab}} = \sum_{ef}(W_L)_{de}1((h_{ic}^{(L)})_e \geq 0)(W_{L-1})_{ef}\frac{\partial}{\partial(W_l)_{ab}}\sigma(h_{ic}^{(L-1)})_f.$$

Now, define the following matrices for $2 \leq l \leq L$

$$(\tilde{W}_{l,ic})_{xy} = (W_l)_{xy}1((h_{ic}^{(l)})_y \geq 0).$$

Continuing,

$$= \sum_{ef}(\tilde{W}_{L,ic})_{de}(W_{L-1})_{ef}\frac{\partial}{\partial(W_l)_{ab}}\sigma(h_{ic}^{(L-1)})_f.$$

Repeating this process leads to

$$= \sum_{ef}(\tilde{W}_{L,ic}...\tilde{W}_{l+1,ic})_{de}\frac{\partial}{\partial(W_l)_{ab}}[(W_l)_{ef}\sigma(h_{ic}^{(l)})_f]$$

$$= \sum_{ef}(\tilde{W}_{L,ic}...\tilde{W}_{l+1,ic})_{de}\delta_{ae}\delta_{bf}\sigma(h_{ic}^{(l)})_f.$$

Define the matrix

$$\tilde{A}_{l+1,ic} = \tilde{W}_{L,ic}...\tilde{W}_{l+1,ic}.$$

Thus, we obtain

$$\frac{\partial(u_{ic})_d}{\partial(W_l)_{ab}} = (\tilde{A}_{l+1,ic})_{da}\sigma(h_{ic}^{(l)})_b,$$

and

$$\frac{\partial \mathcal{L}}{\partial(W_l)_{ab}\partial(W_l)_{ef}} = \text{Av}_{ic}\left\{(\tilde{A}_{l+1,ic}^T\tilde{A}_{l+1,ic})_{ae}\sigma(h_{ic}^{(l)})_b\sigma(h_{ic}^{(l)})_f\right\}.$$

Finally, after flattening, we have

$$\text{Hess}_l = \frac{\partial^2 \mathcal{L}}{\partial w_l^2} = \text{Av}_{ic}\left\{(\tilde{A}_{l+1,ic}^T\tilde{A}_{l+1,ic}) \otimes \sigma(h_{ic}^{(l)})\sigma(h_{ic}^{(l)})^T\right\}.$$

Using the definition of DNC in the ReLU case from Definition 2, We have that $\sigma(h_{ic}^{(l)}) = h_{ic}^{(l)}$. Thus, $1((h_{ic}^{(l)})_y \geq 0) = 1$ for all $y \in \{1,..,d\}$ and for all $l \in \{2,...,L\}$. This implies $\tilde{W}_{l,ic} = W_l$, and so $\tilde{A}_{l+1,ic} = A_{l+1}$. Consequently, the Hessian simplifies to

$$\text{Hess}_l = A_{l+1}^T A_{l+1} \otimes \left[ \text{Av}_{ic} \left\{ h_{ic}^{(l)} h_{ic}^{(l)T} \right\} \right].$$

Since each feature matrix passes through the nonlinearity with no changes, the features $h_{ic}^{(l)}$ are the same as those which occur in the linear network with the same parameter matrices.

The layer-wise Hessian of a solution with DNC structure has the same form as the linear case, and the DNC solutions by definition are global minimum of the linear model and thus obeys the properties detailed in Lemmas 1, 2, 3 and 4. Hence, the proofs of Theorem 1 and Theorem 2 in the linear case, detailed in Appendix C, follow through identically.

We can now consider the gradients. Using the working from above it is simple to show the gradients in the deep UFM have the following form

$$\frac{\partial \mathcal{L}}{\partial (W_l)_{ab}} = \lambda (W_l)_{ab} + \text{Av}_{ic} \left\{ (\tilde{A}_{l+1,ic}^T u_{ic})_a \sigma(h_{ic}^{(l)})_b \right\}.$$

After flattening, and using that $\sigma(h_{ic}) = h_{ic}$ we then have

$$g^{(l)} = \lambda w^{(l)} + \text{Av}_{ic} \left\{ (A_{l+1}^T u_{ic}) \otimes h_{ic}^{(l)} \right\}.$$

We also have for our DNC structure that $u_{ic}$ match the values of the corresponding linear model, and hence this again reduces to the linear gradient, and the proof of Theorem 3, detailed in Appendix C.3, follows through exactly the same.

In the case of the weights, since the DNC structure obeys the same properties as in the linear case, they will necessarily have the same singular value spectrum and singular vectors, and so Theorem 4 also carries over trivially.

## D.2 PROOF OF THEOREM 8

Here, we demonstrate that among solutions satisfying the DNC properties described by Súkeník et al. (2024), the ones that minimize the loss are precisely those from Lemma 1 that additionally have all their intermediate representations with non-negative entries. We establish this by showing that such solutions attain a lower bound on the loss within the class of considered solutions. We shall assume the same regularization condition, stated in equation 17, as in the linear case. This ensures that the best-performing DNC solution outperforms the zero solution, allowing us to restrict our analysis to cases where no parameter matrix is zero.

We express the parameter matrices with their scales separated

$$W_l = \alpha_l \hat{W}_l, \quad H_1 = \alpha_0 \hat{H}_1, \quad \text{where } \alpha_i > 0, \ i = 0, ..., L.$$

The matrices $\hat{W}_l$ for $l = 1, ..., L$ and $\hat{H}_1$ have unit Frobenius norm. Using the homogeneity of ReLU, we can write the output matrix $Z$ as

$$Z = W_L \sigma(...W_2 \sigma(W_1 H_1)...) = \left( \prod_{l=0}^{L} \alpha_l \right) \hat{W}_L \sigma(...\hat{W}_2 \sigma(\hat{W}_1 \hat{H}_1)...).$$

Since we assume that $Z$ aligns with the matrix $I_K \otimes 1_n^T$, we write

$$\hat{W}_L \sigma(...\hat{W}_2 \sigma(\hat{W}_1 \hat{H}_1)...) = \beta I_K \otimes 1_n^T.$$

Substituting this into the loss, and using the parameter matrices decompositions, the loss for a DNC solution is

$$\mathcal{L}_{\text{DNC}} = \frac{1}{2Kn} \|Z - Y\|_F^2 + \frac{1}{2} \lambda \|H_1\|_F^2 + \frac{1}{2} \lambda \sum_{l=1}^{L} \|W_l\|_F^2$$

$$= \frac{1}{2} \left( \beta \prod_{l=1}^{L} \alpha_l - 1 \right)^2 + \frac{1}{2} \lambda \sum_{l=0}^{L} \alpha_l^2 := f(\beta, \alpha_0, ..., \alpha_L).$$

We now argue that for a fixed $\beta$, the function $f(\beta, \alpha_0, ..., \alpha_L)$ is minimized when all $\alpha_l$ are equal. First, since $f \to \infty$ if any $\alpha_l \to \infty$, the minimum must occur at a finite turning point. Taking a derivative,

$$\frac{\partial f}{\partial \alpha_l} = \lambda \alpha_l + \beta \left( \prod_{l' \neq l}^{L} \alpha_{l'} \right) \left( \beta \prod_{l'=0}^{L} \alpha_{l'} - 1 \right).$$

Setting this to zero, and using that $\alpha_l \neq 0$, gives

$$\lambda \alpha_l^2 = \beta \left( \prod_{l'=0}^{L} \alpha_{l'} \right) \left( \beta \prod_{l'=0}^{L} \alpha_{l'} - 1 \right).$$

Since the right-hand side is the same for all $l$, it follows that each $\alpha_l$ must be equal at the minimum. Thus, defining $\alpha$ as the common value,

$$\mathcal{L}_{\text{DNC}} \geq \min_{\alpha} \underbrace{\left\{ \frac{1}{2} (\beta \alpha^{L+1} - 1)^2 + \frac{1}{2} \lambda (L+1) \alpha^2 \right\}}_{f(\beta, \alpha)} := g(\beta),$$

where we have defined the function $f(\beta, \alpha)$ to acknowledge we are focused on the $\alpha_l = \alpha$ case, as well as the function $g(\beta)$ where the minimum over $\alpha$ of $f(\beta, \alpha)$ is attained.

We now claim $g(\beta)$ is monotonically decreasing for $\beta > 0$. Let $0 < \beta_1 < \beta_2$, and let $\alpha' = \text{argmin}_{\alpha} \{ f(\beta_1, \alpha) \}$. Defining

$$\alpha^* = \alpha' \left( \frac{\beta_1}{\beta_2} \right)^{\frac{1}{L+1}},$$

we obtain

$$g(\beta_1) = \frac{1}{2} (\beta_1 (\alpha')^{L+1} - 1)^2 + \frac{1}{2} (L+1)(\alpha')^2$$

$$> \frac{1}{2} (\beta_1 (\alpha')^{L+1} - 1)^2 + \frac{1}{2} (L+1)(\alpha')^2 \left( \frac{\beta_1}{\beta_2} \right)^{\frac{2}{L+1}} = f(\beta_2, \alpha^*) \geq g(\beta_2),$$

which gives $g(\beta_1) > g(\beta_2)$, confirming $g(\beta)$ is monotonically decreasing.

To summarize what we have shown so far, the loss of any DNC solution is lower bounded by a solution where the frames are chosen so as to maximize $\beta$, and the scales $\alpha_l$ are equal and set to the value that minimizes $f(\beta_{\max}, \alpha)$. We now show that the solutions that maximize $\beta$ are precisely the ones with positive intermediate representations.

Recall we define $\Lambda_l = \sigma(H_l)$ for $l > 1$. By assumption we have that $H_l$ and $\Lambda_l$ have the following forms

$$H_l = \bar{H}_l \otimes 1_n^T, \quad \Lambda_l = \bar{\Lambda}_l \otimes 1_n^T,$$

where the matrices $\bar{H}_l, \bar{\Lambda}_l$ align with an orthogonal frame. This implies that their SVDs can be written as

$$\bar{H}_l = \gamma_l U_l^H \tilde{D}^T Q^T, \quad \bar{\Lambda}_l = \rho_l U_l^\Lambda \tilde{D}^T Q^T,$$

where $Q \in \mathbb{R}^{K \times K}$ and $U_l^H, U_l^\Lambda \in \mathbb{R}^{d \times d}$ are orthogonal matrices, $\tilde{D} = [I_K, 0_{K \times (d-K)}] \in \mathbb{R}^{K \times d}$, and $\gamma_l, \rho_l$ are scales that give the nonzero singular values of $\bar{H}_l$ and $\bar{\Lambda}_l$ respectively.

We now aim to derive an upper bound for $\beta$ in terms of the singular values of the normalized features and weight matrices. For each $l$, we have the equation $H_l = W_{l-1} \Lambda_{l-1}$. Since these matrices have

repeated columns, it follows that $\bar{H}_l = W_{l-1}\bar{\Lambda}_{l-1}$. Denote the first $K$ singular values of $W_{l-1}$ by $\omega_i^{(l-1)}$ for $i = 1, ..., K$, arranged in decreasing order. All other singular values are zero by the assumed third DNC property. Given that the singular values of $\bar{H}_l$ are $\gamma_l$ with multiplicity $K$, and a similar statement holds for $\bar{\Lambda}_{l-1}$ with $\rho_{l-1}$ being the nonzero singular values, Lemma 6 gives us the following inequality for $2 \leq l \leq L + 1$

$$\gamma_l \leq \omega_K^{(l-1)}\rho_{l-1}. \tag{19}$$

This gives us an inequality relating $\gamma_l$ to $\rho_{l-1}$. We will now get an inequality that relates $\rho_{l-1}$ to $\gamma_{l-1}$. We have $\Lambda_l = \sigma(H_l)$, and using that there are repeated columns this gives $\bar{\Lambda}_l = \sigma(\bar{H}_l)$. Taking Frobenius norm squared and using the property $\|\sigma(M)\|_F^2 \leq \|M\|_F^2$, with equality only if $\sigma(M) = M$, we obtain

$$\|\bar{\Lambda}_l\|_F^2 \leq \|\bar{H}_l\|_F^2,$$

which, using the forms of these matrices, gives

$$\rho_l \leq \gamma_l. \tag{20}$$

This inequality also extends to $l = 1$, where it is trivially satisfied since we do not apply a nonlinearity at this layer, and so $H_1 = \Lambda_1$. Using the two inequalities in equation 19 and equation 20 as a recurrence relation in $\gamma_l$ and $\rho_l$, we derive

$$\gamma_{L+1} \leq \gamma_1 \omega_K^{(L)}...\omega_K^{(1)}.$$

Since $H_{L+1} = Z$, and hence $\gamma_{L+1} = \beta \prod_{l=0}^{L} \alpha_l$, we have

$$\beta \prod_{l=0}^{L} \alpha_l \leq \gamma_1 \omega_K^{(L)}...\omega_K^{(1)}.$$

Using $\omega_K^{(l)}/\alpha_l = \hat{\omega}_K^{(l)}$, where $\hat{\omega}_K^{(l)}$ are the smallest potentially nonzero singular values of the normalized weight matrices, and noting that $\gamma_1/\alpha_0 = 1/\sqrt{nK}$, we arrive at

$$\beta \leq \frac{1}{\sqrt{nK}}\hat{\omega}_K^{(L)}...\hat{\omega}_K^{(1)}.$$

Since $\hat{W}_l$ are rank $K$ matrices with unit Frobenius norm, the maximum value of their smallest nonzero singular value occurs when all singular values are equal, giving $\omega_K^{(l)} = 1/\sqrt{K}$. Thus,

$$\beta \leq \frac{1}{\sqrt{n}}K^{-\frac{L+1}{2}}.$$

Going back through the conditions for this inequality to be attained, we have the following

1. Each $W_l$ has $K$ equal nonzero singular values.

2. Intermediate representations are positive for $l = 2...., L$, meaning $H_l = \sigma(H_l)$.

3. The singular values of $\Lambda_l$ and $H_l$ obey $\gamma_l = \omega_K^{(l-1)}\rho_{l-1}$.

These conditions are only on the frames of the parameter matrices, we also require the following of their scales

4. For $l_1, l_2 \in \{0, ..., L\}$ we have $\alpha_{l_1} = \alpha_{l_2} = \alpha$.

5. The value $\alpha$ attains the minimum of the function $f(\beta_{\max}, \alpha)$, where $\beta_{\max} = n^{-\frac{1}{2}} K^{-\frac{L+1}{2}}$.

Condition (2) implies that the nonlinearity has no effect for the best-performing DNC solutions. As a result, the network function can be expressed equivalently by a linear network, and has a loss that is attainable in the linear case. We know the global minimum of our linear loss is given by solutions with the structure described in Lemma 1, and a subset of these global minima obey condition (2). Since these are the best performing linear solutions and can be expressed by a nonlinear network, they are the only candidates for the minimal DNC solutions in the nonlinear case. It follows immediately from Lemma 1, as well as properties detailed in Lemmas 6 and 3, that they obey all of the other conditions, and so the best performing DNC solutions in the nonlinear model are precisely the set of global minimum solutions of the linear model, with the extra condition that the intermediate representations are non-negative.

### D.3 Supporting Lemmas

The supporting lemma necessary for the proofs in Appendix D is provided here.

**Lemma 6** (from Horn & Johnson (1985)). *Let $A \in \mathbb{R}^{m \times k}$ and $B \in \mathbb{R}^{k \times n}$. Denoting the $i^{th}$ singular value of a matrix by $\sigma_i(\cdot)$, ordered in descending magnitude, we have*

$$\sigma_i(AB) \leq \sigma_i(A)\,\sigma_1(B).$$

## E  Other Deep Learning Matrices

Here, we examine the covariance of gradients and backpropagation errors, describing the structure at global minima for the deep linear UFM.

### E.1 Covariance of Gradients

The covariance of gradients was considered empirically by Jastrzebski et al. (2020). Specifically, we examine the matrix

$$C^{(l)} = \mathrm{Av}_{ic}\big\{(g_{ic}^{(l)} - g^{(l)})(g_{ic}^{(l)} - g^{(l)})^T\big\}, \tag{21}$$

where

$$g_{ic}^{(l)} = \nabla_{w^{(l)}} l(f(x_{ic}; \theta), y_c), \quad g^{(l)} = \mathrm{Av}_{ic}\big\{g_{ic}^{(l)}\big\}.$$

Note that we only consider the gradient with respect to the MSE loss, not the full loss including the regularizes, and work at a layer-wise level. From equation 14 in Appendix C.3, we recall that

$$\nabla_{w^{(l)}} l(f(x_{ic}; \theta), y_c) = A_{l+1}^T u_{ic} \otimes h_{ic}^{(l)}.$$

Under the regularization condition in equation 17 and assuming $d \geq K$, equation 15 in Appendix C.3 gives $A_{l+1}^T u_{ic}$, in terms of the feature means and the constant $C$, leading to

$$g_{ic}^{(l)} = C(C-1)\frac{\|\mu^{(l)}\|_2}{\|\mu^{(l+1)}\|_2}\hat{\mu}_c^{(l+1)} \otimes \hat{\mu}_c^{(l)}, \quad g^{(l)} = \frac{C(C-1)}{K}\frac{\|\mu^{(l)}\|_2}{\|\mu^{(l+1)}\|_2}\sum_{c=1}^{K}\hat{\mu}_c^{(l+1)} \otimes \hat{\mu}_c^{(l)}.$$

Writing $\hat{\nu}_c^{(l)} = \hat{\mu}_c^{(l+1)} \otimes \hat{\mu}_c^{(l)}$ to compress the notation, we find that at the optimum, $C^{(l)}$ can be expressed as

$$C^{(l)} = \frac{C^2(C-1)^2}{K}\frac{\|\mu^{(l)}\|_2^2}{\|\mu^{(l+1)}\|_2^2}\sum_{c=1}^{K}\left[\hat{\nu}_c^{(l)} - \mathrm{Av}_{c'}\big\{\hat{\nu}_{c'}^{(l)}\big\}\right]\left[\hat{\nu}_c^{(l)} - \mathrm{Av}_{c''}\big\{\hat{\nu}_{c''}^{(l)}\big\}\right]^T.$$

The vectors $\hat{\nu}_c^{(l)}$, for $c \in \{1, ..., K\}$, are orthogonal, and so the centered vectors

$$\hat{\nu}_c^{(l)} - \mathrm{Av}_{c'}\{\hat{\nu}_{c'}^{(l)}\}, \quad \text{for } c \in \{1, ..., K\},$$

form a simplex ETF and span a $K-1$ dimensional subspace. Additionally, it follows that $C^{(l)}$ has a spanning set of eigenvectors with nonzero eigenvalue given by

$$\hat{\nu}_1^{(l)} - \hat{\nu}_c^{(l)}, \quad \text{for } c \in \{2, ..., K\},$$

each with eigenvalue

$$\frac{1}{K} n^{\frac{-1}{L+1}} C^{\frac{2L}{L+1}} (C-1)^2,$$

where we used equation 18 to express the feature mean norms in terms of $C$.

Thus, we state the following theorem for the second moment of gradients.

**Theorem 10.** *Consider the deep linear UFM described in equation 4. Let the network width satisfy $d \geq K$, and consider a layer $l$ satisfying $1 \leq l < L$. Additionally, assume that the regularization parameter $\lambda$ satisfies the condition in equation 17. Then, at any global optimum of the loss, the layer-wise covariance of gradients matrix $C^{(l)}$, defined in equation 21, takes the form*

$$C^{(l)} = \frac{1}{K} n^{\frac{-1}{L+1}} C^{\frac{2L}{L+1}} (C-1)^2 \sum_{c=1}^{K} \left[\hat{\nu}_c^{(l)} - Av_{c'}\left\{\hat{\nu}_{c'}^{(l)}\right\}\right] \left[\hat{\nu}_c^{(l)} - Av_{c''}\left\{\hat{\nu}_{c''}^{(l)}\right\}\right]^T,$$

*where $\hat{\nu}_c^{(l)} = \hat{\mu}_c^{(l+1)} \otimes \hat{\mu}_c^{(l)}$. As a consequence, $C^{(l)}$ has rank $K-1$, with a spanning set of nonzero eigenvectors given by*

$$\hat{\nu}_1^{(l)} - \hat{\nu}_c^{(l)}, \quad \text{for } c \in \{2, \ldots, K\}.$$

*Furthermore, all the nonzero eigenvalues are equal and given by*

$$\frac{1}{K} n^{\frac{-1}{L+1}} C^{\frac{2L}{L+1}} (C-1)^2,$$

*where $C$ is the constant referred to in Theorem 1.*

### E.2 BACKPROPAGATION ERRORS

The backpropagation errors have been studied by Oymak et al. (2019) and Papyan (2020). Here we consider the extended backpropagation errors, similar to the analysis by Papyan (2020), but adapted to the MSE loss. For a given sample $x_{ic}$, the extended backpropagation error at layer $l$ in our model is defined as

$$\delta_{icc'}^{(l)} = \frac{\partial l(f(x_{ic}; \theta), y_{c'})}{\partial h_{ic}^{(l)}}.$$

We then consider the weighted second moment matrix of the extended backpropagation errors, using the same weights we had in our Hessian matrix, similar to the approach by Papyan

$$\Delta^{(l)} = \sum_{i,c,c'} \frac{1}{Kn} \delta_{icc'}^{(l)} \delta_{icc'}^{(l)T}. \tag{22}$$

Using the fact that

$$\frac{\partial l(f(x_{ic}; \theta), y_{c'})}{\partial h_{ic}^{(l)}} = A_l^T u_{icc'},$$

where $u_{icc'} = W_L...W_1 h_{ic} - y_{c'}$, we obtain

$$\Delta^{(l)} = \frac{1}{Kn} \sum_{i,c,c'} (A_l^T u_{icc'})(A_l^T u_{icc'})^T.$$

Applying the regularization condition in equation 17, assuming $d \geq K$, and considering a global optimum, the DNC1 property from Lemma 1 gives

$$\Delta^{(l)} = \frac{1}{K} \sum_{c,c'} (A_l^T u_{cc'})(A_l^T u_{cc'})^T,$$

where $u_{cc'} = W_L...W_1 \mu_c - y_{c'}$. The DNC2 and DNC3 properties of Lemma 1 then yield

$$A_l^T u_{cc'} = A_l^T A_l \mu_c^{(l)} - A_l^T y_{c'}$$

$$= \alpha^{(l)2} \|\mu^{(l)}\|_2^2 \mu_c^{(l)} - \alpha^{(l)} \mu_{c'}^{(l)}.$$

Using the expression for $C$ in terms of $\alpha^{(l)}$ and $\|\mu^{(l)}\|_2$ from equation 13, we obtain

$$A_l^T u_{cc'} = \alpha^{(l)} \left[ C \mu_c^{(l)} - \mu_{c'}^{(l)} \right].$$

Thus, the extended backpropagated error matrix takes the form

$$\Delta^{(l)} = \frac{\alpha^{(l)2}}{K} \sum_{c,c'} \left[ C \mu_c^{(l)} - \mu_{c'}^{(l)} \right] \left[ C \mu_c^{(l)} - \mu_{c'}^{(l)} \right]^T.$$

Expanding the summation and simplifying gives

$$= \alpha^{(l)2} \left[ (C^2 + 1) \sum_{c=1}^{K} \mu_c^{(l)} \mu_c^{(l)T} - 2CK \mu_G^{(l)} \mu_G^{(l)T} \right].$$

Clearly, the image of $\Delta^{(l)}$ lies in $\text{Span}\{\mu_c^{(l)}\}_{c=1}^K$, implying that $\Delta^{(l)}$ has at most rank $K$. Considering the action of $\Delta^{(l)}$ on specific vectors, we find

$$\Delta^{(l)} \left[ \mu_c^{(l)} - \mu_G^{(l)} \right] = \alpha^{(l)} C(C^2 + 1) \left[ \mu_c^{(l)} - \mu_G^{(l)} \right], \qquad \Delta^{(l)} \mu_G^{(l)} = \alpha^{(l)} C(C-1)^2 \mu_G^{(l)}.$$

Thus, the eigenspace corresponding to the eigenvalue $\alpha^{(l)} C(C^2 + 1)$ has dimension $K - 1$, while the eigenspace corresponding to $\alpha^{(l)} C(C-1)^2$ has dimension 1. Notably, all but one nonzero eigenvalue are equal, with the distinct eigenvalue being smaller, since $C \approx 1$ for well-performing models. Replacing $\alpha^{(l)}$ using equation 13, and writing the norm of the feature means in terms of $C$ using equation 18, we arrive at the following theorem.

**Theorem 11.** *Consider the deep linear UFM described in equation 4. Let the network width satisfy $d \geq K$, and consider a layer $l$ satisfying $1 \leq l < L$. Additionally, assume that the regularization parameter $\lambda$ satisfies the condition in equation 17. Then, at any global optimum of the loss, the extended backpropagation error matrix $\Delta^{(l)}$, defined in equation 22, has rank $K$. Moreover, the top eigenspace has dimension $K - 1$, with eigenvalue*

$$(C\sqrt{n})^{\frac{2L-2l+2}{L+1}} (C^2 + 1),$$

*and a set of corresponding spanning eigenvectors given by*

$$\mu_1^{(l)} - \mu_c^{(l)}, \quad for\ c \in \{2, \dots, K\}.$$

*The lower one-dimensional eigenspace has eigenvalue*

$$(C\sqrt{n})^{\frac{2L-2l+2}{L+1}} (C - 1)^2,$$

*with corresponding eigenvector $\mu_G^{(l)}$. Here, $C$ is the same constant as referred to in Theorem 1.*

Notably, our result appears to differ from Papyan et al. (2020), which reported $K^2$ outliers in the weighted second moment of backpropagation errors. This discrepancy may arise from differences in the choice of loss function.

## F  CE CASE PROOFS

recall in this case our loss function is

$$\mathcal{L}(H_1, W_1, ..., W_L) = \frac{1}{Kn} g(W_L...W_1 H_1, Y) + \frac{1}{2}\lambda\|H_1\|_F^2 + \frac{1}{2}\lambda\sum_{l=1}^{L} \|W_l\|_F^2,$$

where, if we define the matrix $Z = W_L...W_1 H_1$, which has columns $z_{ic} = W_L...W_1 h_{ic}$, we have

$$g(Z, Y) = -\sum_{i=1}^{n}\sum_{c=1}^{K} \log\left(\frac{\exp((z_{ic})_c)}{\sum_{c'=1}^{K}\exp((z_{ic})_{c'})}\right).$$

Additionally, define $A_{l+1} = W_L...W_{l+1}$ and $H_l = W_{l-1}...W_1 H_1$ as before.

### F.1  PROOF OF THEOREM 9

First note that

$$\frac{\partial Z_{ab}}{\partial (W_l)_{uv}} = (A_{l+1})_{au}(H_l)_{vb},$$

$$\frac{\partial g(Z, Y)}{\partial Z_{ab}} = (P - Y)_{ab}, \quad \text{where} \quad P_{ab} = \frac{\exp(Z_{ab})}{\sum_{a'=1}^{K}\exp(Z_{a'b})}.$$

Taking derivatives of $\mathcal{L}$ with respects to $W_l$ Then gives

$$\frac{\partial \mathcal{L}}{\partial (W_l)_{uv}} = \frac{1}{Kn}\frac{\partial Z_{ab}}{\partial (W_l)_{uv}}\frac{\partial g(Z, Y)}{\partial Z_{ab}} + \lambda(W_l)_{uv} \implies$$

$$\frac{\partial \mathcal{L}}{\partial W_l} = \frac{1}{Kn}A_{l+1}^T(P - Y)H_l^T + \lambda W_l.$$

We now drop the regularization term, since again it just translates the eigenvalues and leaves eigenvectors unchanged.

Now also note that

$$\frac{\partial P_{ab}}{\partial Z_{uv}} = \delta_{vb}[\text{diag}(p_b) - p_b p_b^T]_{au},$$

where $p_b$ is the $b^{\text{th}}$ column of $P$, meaning $(p_b)_i = P_{ib}$. We also define the matrix $\rho_b = \text{diag}(p_b) - p_b p_b^T$.

Using this, the second derivative is

$$\frac{\partial^2 \mathcal{L}}{\partial (W_l)_{uv}(W_l)_{xy}} = \frac{1}{Kn}(A_{l+1}^T)_{ua}\frac{\partial P_{ab}}{\partial (W_l)_{xy}}(H_l^T)_{bv}$$

$$= \frac{1}{Kn}(A_{l+1}^T)_{ua}\frac{\partial Z_{rs}}{\partial (W_l)_{xy}}\frac{\partial P_{ab}}{\partial Z_{rs}}(H_l^T)_{bv}.$$

Now being explicit over which variables are summed over:

$$= \frac{1}{Kn}\sum_{abrs}(A_{l+1}^T)_{ua}(A_{l+1})_{rx}(H_l)_{ys}(H_l^T)_{bv}\delta_{sb}(\rho_b)_{ar}.$$

Using our notation of the columns of $H_l$ being $h_b^{(l)}$ in the sense that $(h_b^{(l)})_v = (H_l)_{vb}$, this reduces to

$$\frac{\partial^2 \mathcal{L}}{\partial (W_l)_{uv}(W_l)_{xy}} = \frac{1}{Kn} \sum_b (A_{l+1}^T \rho_b A_{l+1})_{ux} (h_b^{(l)} h_b^{(l)T})_{yv}.$$

Now note that $b$ sums over the number of data points, we can hence choose to replace it with a sum over $i$ and $c$, meaning a sum over the samples, giving

$$\frac{\partial^2 \mathcal{L}}{\partial (W_l)_{uv}(W_l)_{xy}} = \mathrm{Av}_{ic} \left\{ (A_{l+1}^T \rho_{ic} A_{l+1})_{ux} (h_{ic}^{(l)} h_{ic}^{(l)T})_{yv} \right\}.$$

Now flattening the weights as we have done previously gives us that

$$\mathrm{Hess}_l = \mathrm{Av}_{ic} \left\{ \left( A_{l+1}^T \rho_{ic} A_{l+1} \right) \otimes \left( h_{ic}^{(l)} h_{ic}^{(l)T} \right) \right\}.$$

Note this is different from the MSE case, since the left side of the Kronecker product does depend on the considered data-point, and so we cannot just consider the spectrum of each side individually.

We now specialize to our DNC solution. First note using the properties of the definition we have that for $1 \le l < L$: $A_{l+1} = U_L \tilde{\Sigma}^{L-l} U_l^T$, $H_l = U_{l-1} \bar{\Sigma}^l V_0^T$, and hence $h_{ic}^{(l)} = U_{l-1} \bar{\Sigma}^l V_0^T e_{ic}$, where $e_{ic}$ are the standard basis vectors ordered to match the columns of $H_1$.

Plugging this into our layer-wise Hessian and using properties of the Kronecker product this gives

$$\mathrm{Hess}_l = (U_l \otimes U_{l-1})[\mathrm{Av}_{ic}\{((\tilde{\Sigma}^{L-l})^T U_L^T \rho_{ic} U_L \tilde{\Sigma}^{L-l}) \otimes (\bar{\Sigma}^l V_0^T e_{ic} e_{ic}^T V_0 (\bar{\Sigma}^l)^T)\}](U_l \otimes U_{l-1})^T.$$

We now drop the orthogonal transformation for ease of exposition, and will return to it later. Define the resulting matrix as $\mathrm{Hess}_l'$. We can also seperate the scales from the matrices $\tilde{\Sigma}, \bar{\Sigma}$ since all nonzero singular values are equal, hence write $\bar{\Sigma} = (\alpha\sqrt{n})^{\frac{1}{L+1}} \bar{D}$ and $\tilde{\Sigma} = (\alpha\sqrt{n})^{\frac{1}{L+1}} \tilde{D}$, where $\bar{D} \in \mathbb{R}^{d \times Kn}$, $\tilde{D} \in \mathbb{R}^{K \times d}$ both have their top $K \times K$ block being $\mathrm{diag}(1, 1, ..., 1, 0)$, with all other entries being zero.

This gives

$$\mathrm{Hess}_l' = (\alpha\sqrt{n})^{\frac{2L}{L+1}} \mathrm{Av}_{ic}\{(\tilde{D}^T U_L^T \rho_{ic} U_L \tilde{D}) \otimes (\bar{D} V_0^T e_{ic} e_{ic}^T V_0 \bar{D}^T\}.$$

We then drop this constant out front for now and will return to it later. Also perform the orthogonal transformation given by the following matrix

$$Q = \begin{bmatrix} U_L & 0_{K \times (d-K)} \\ 0_{(d-K) \times K} & I_{d-K} \end{bmatrix} \otimes \begin{bmatrix} U_L & 0_{K \times (d-K)} \\ 0_{(d-K) \times K} & I_{d-K} \end{bmatrix} \in \mathbb{R}^{d^2 \times d^2}.$$

call the resulting matrix $\mathrm{Hess}_l''$. This matrix is an average of cross products. From here we shall suppress the matrix dimensions on the identity matrix and zero matrix, which should be clear from context. The left side of the Kronecker product within the average is given by

$$\begin{bmatrix} U_L & 0 \\ 0 & I \end{bmatrix} \tilde{D}^T U_L^T \rho_{ic} U_L \tilde{D} \begin{bmatrix} U_L^T & 0 \\ 0 & I \end{bmatrix}$$

$$= \begin{bmatrix} U_L D U_L^T \rho_{ic} U_L D U_L^T & 0 \\ 0 & 0 \end{bmatrix},$$

where $D = \mathrm{diag}(1, 1, ..., 1, 0) \in \mathbb{R}^{K \times K}$. Note that the standard simplex $S = I_K - \frac{1}{K} 1_K 1_K^T$ can be diagonalizes as $S = U_L D U_L^T$, hence the left side of the Kronecker product finally reduces to

$$= \begin{bmatrix} S\rho_{ic}S & 0 \\ 0 & 0 \end{bmatrix}.$$

the right side of the Kronecker product within the average is given by

$$\begin{bmatrix} U_L & 0 \\ 0 & I \end{bmatrix} \bar{D}V_0^T e_{ic}e_{ic}^T V_0 \bar{D}^T \begin{bmatrix} U_L^T & 0 \\ 0 & I \end{bmatrix}$$

$$= \begin{bmatrix} U_L D'V_0^T e_{ic}e_{ic}^T V_0 D'U_L^T & 0 \\ 0 & 0 \end{bmatrix},$$

where $D' \in \mathbf{R}^{K \times Kn}$ is a singular value matrix with the nonzero values being 1 with multiplicity $K - 1$. Now use the fact that $S \otimes 1_n^T = \sqrt{n}\, U_L D'V_0^T$, this reduces to

$$\frac{1}{n} \begin{bmatrix} (S \otimes 1_n^T)e_{ic}e_{ic}^T(S \otimes 1_n^T) & 0 \\ 0 & 0 \end{bmatrix}.$$

The layer-wise Hessian is then

$$\text{Hess}_l'' = \frac{1}{n}\text{Av}_{ic}\left\{ \begin{bmatrix} S\rho_{ic}S & 0 \\ 0 & 0 \end{bmatrix} \otimes \begin{bmatrix} (S \otimes 1_n^T)e_{ic}e_{ic}^T(S \otimes 1_n^T) & 0 \\ 0 & 0 \end{bmatrix} \right\}.$$

Now note that this matrix only has nonzero entries in its top $K^2 \times K^2$ block. If we are interested in the nonzero eigenvalues and eigenvectors we can focus on this top block matrix. We also drop the factor of $1/n$ for now. Call the resulting $K^2 \times K^2$ matrix $\text{Hess}_l'''$.

Now note that, by definition, for a DNC solution we have $\rho_{ic} = \rho_c$, i.e. they are independent of the $i$ index. In addition if we define $s_{ic} = (S \otimes 1_n^T)e_{ic}$, this is also independent of the $i$ index, so $s_{ic} = s_c$, where $s_c$ are the columns of the standard simplex $S$. This further reduces our Hessian to

$$\text{Hess}_l''' = \text{Av}_c\left\{ S\rho_c S \otimes s_c s_c^T \right\}.$$

We now demonstrate that $\rho_c$ has two terms, one of which is exponentially smaller than the other. We show this for $c = 1$, for simplicity, though it should be clear that this calculation is the same for all $c = 1, ..., K$. First note using the form of $Z$ in the definition of DNC

$$p_1^T = \frac{1}{(K-1) + e^\alpha}(e^\alpha, 1..., 1).$$

Hence

$$\text{diag}(p_1) = \frac{1}{(K-1) + e^\alpha}\text{diag}(e^\alpha, 1, ..., 1),$$

$$p_1 p_1^T = \left( \frac{1}{(K-1) + e^\alpha} \right)^2 \begin{bmatrix} e^{2\alpha} & e^\alpha & e^\alpha & ... & e^\alpha \\ e^\alpha & 1 & 1 & ... & 1 \\ e^\alpha & 1 & 1 & ... & 1 \\ ... & ... & ... & ... & ... \\ e^\alpha & 1 & 1 & ... & 1 \end{bmatrix},$$

and so

$$\rho_1 = \text{diag}(p_1) - p_1 p_1^T$$

$$= \left(\frac{1}{(K-1)+e^\alpha}\right)^2 \begin{bmatrix} (K-1)e^\alpha & -e^\alpha & -e^\alpha & ... & -e^\alpha \\ -e^\alpha & (K-2)+e^\alpha & -1 & ... & -1 \\ -e^\alpha & -1 & (K-2)+e^\alpha & ... & -1 \\ ... & ... & ... & ... & ... \\ -e^\alpha & -1 & -1 & ... & (K-2)+e^\alpha \end{bmatrix}.$$

$$= \frac{e^\alpha}{((K-1)+e^\alpha)^2} \underbrace{\begin{bmatrix} (K-1) & -1 & -1 & ... & -1 \\ -1 & 1 & 0 & ... & 0 \\ -1 & 0 & 1 & ... & 0 \\ ... & ... & ... & ... & ... \\ -1 & 0 & 0 & ... & 1 \end{bmatrix}}_{\rho'_1} + O(e^{-2\alpha}). \tag{23}$$

We refer to this leading order term as $\rho'_c$, and from here drop the higher order term. This is reasonable since $\alpha$ is large when the level of regularization is small. We also drop the constant out front for now and will bring it back with the previously dropped constant later.

We now use Lemma 7, which states that

$$\rho'_c = Ks_c s_c^T + S.$$

Also noting that $Ss_c = s_c$, we get that $S\rho'_c S = \rho'_c$. This further reduces us to the leading order term being

$$\text{Hess}'''_l = \text{Av}_c \left\{ \rho'_c \otimes s_c s_c^T \right\},$$

and hence that

$$\text{Hess}'''_l = \frac{1}{K} \sum_{c=1}^{K} \left( \left[ Ks_c s_c^T + \sum_{b=1}^{K} s_b s_b^T \right] \otimes s_c s_c^T \right)$$

$$= \left[ \sum_{c=1}^{K} s_c s_c^T \otimes s_c s_c^T \right] + \frac{1}{K} \left( \sum_{c=1}^{K} s_c s_c^T \right) \otimes \left( \sum_{c=1}^{K} s_c s_c^T \right)$$

$$= \left[ \sum_{c=1}^{K} s_c s_c^T \otimes s_c s_c^T \right] + \frac{1}{K} S \otimes S. \tag{24}$$

Note this matrix has the following eigenvectors and eigenvalues:

- eigenvalue 1 with multiplicity 1 and eigenvector $\sum_c s_c \otimes s_c$.
- eigenvalue $\frac{K-1}{K}$, multiplicity $K-1$, eigenvectors of the form $s_a \otimes s_a - s_b \otimes s_b$, $a \neq b$, $a, b = 1, ..., K$.
- eigenvalue $\frac{1}{K}$, multiplicity $K^2 - 3K + 1$, eigenvectors of the form $s_a \otimes s_b - s_b \otimes s_a$, $a \neq b$, $a, b = 1, ..., K$.

This is the separation of the spectrum into $K$ spikes, with one being larger than the others, and a mini-bulk of size $O(K(K-1))$. The bulk itself represents the remaining zero singular values of the original larger matrix.

We can now reverse the orthogonal transformations and scaling to get back to $\text{Hess}_l$. The combination of the constants that we dropped is equal to

$$\beta = \frac{e^\alpha (\alpha \sqrt{n})^{\frac{2L}{L+1}}}{n((K-1)+e^\alpha)^2}.$$

Reversing the orthogonal transformations, the component from the second of the two matrices in equation 24 gives

$$= \frac{1}{K} \beta \left( U_l D U_l^T \otimes U_{l-1} D U_{l-1}^T \right),$$

whilst the first gives

$$= \beta \sum_c U_l \tilde{D}^T U_L^T e_c e_c^T U_L \tilde{D} U_l^T \otimes U_{l-1} \tilde{D}^T U_L^T e_c e_c^T U_L \tilde{D} U_{l-1}^T.$$

Using that $\bar{H}_{l+1} = (\alpha\sqrt{n})^{\frac{l}{L+1}} U_l \tilde{D}^T U_L^T$ and $\bar{H}_{l+1} \bar{H}_{l+1}^T = (\alpha\sqrt{n})^{\frac{2l}{L+1}} U_l D U_l^T$, as well as $\|\bar{H}_{l+1}\|_F^2 = K\|\mu^{(l+1)}\|_2^2$ gives

$$\sum_c \hat{\mu}_c^{(l+1)} \hat{\mu}_c^{(l+1)T} = \frac{K}{K-1} U_l D U_l^T, \quad \hat{\mu}_c^{(l+1)} = \sqrt{\frac{K}{K-1}} U_l \tilde{D}^T U_L^T e_c,$$

Writing $\hat{\nu}_{cc'}^{(l)} = \hat{\mu}_c^{(l+1)} \otimes \hat{\mu}_{c'}^{(l)}$, the leading order term of the layer-wise Hessian is then given by:

$$\text{Hess}_l = \beta \left( \frac{K-1}{K} \right)^2 \left[ \sum_c \hat{\nu}_{cc}^{(l)} \hat{\nu}_{cc}^{(l)T} + \frac{1}{K} \sum_{c,c'} \hat{\nu}_{cc'}^{(l)} \hat{\nu}_{cc'}^{(l)T} \right],$$

Since the class means $\hat{\mu}_c^{(l)}$ satisfy the same dot product relationships as the columns of the matrix $S$, and the eigenvalues are simply scaled by $\beta$, we arrive at the following eigenvalues and eigenvectors:

- Eigenvalue $\beta$ with multiplicity 1 and eigenvector $\sum_c \hat{\nu}_{cc}^{(l)}$.

- Eigenvalue $\frac{K-1}{K}\beta$, multiplicity $K-1$, eigenvectors of the form $\hat{\nu}_{cc}^{(l)} - \hat{\nu}_{c'c'}^{(l)}$, $c \neq c'$, $c, c' = 1, ..., K$.

- Eigenvalue $\frac{1}{K}\beta$, multiplicity $K^2 - 3K + 1$, eigenvectors of the form $\hat{\nu}_{cc'}^{(l)} - \hat{\nu}_{c'c}^{(l)}$, $c \neq c'$, $c, c' = 1, ..., K$.

- the remaining $d^2 - (K-1)^2$ eigenvalues are zero.

### F.2 SUPPORTING LEMMAS

The supporting lemma necessary for the proof in Appendix F is provided here.

**Lemma 7.** *The quantity $\rho_c'$, defined in equation 23, can be written as:*

$$\rho_c' = K s_c s_c^T + S.$$

**Proof:** Note that the entries of $\rho_c'$ are given by

$$(\rho_c')_{ij} = \begin{cases} K-1, & \text{if } i = j = c \\ 1, & \text{if } i = j \neq c \\ -1, & \text{if } i = c, \text{ or } j = c, \text{ but } i \neq j \\ 0, & \text{otherwise} \end{cases}$$

Similarly the entries of $K s_c s_c^T$ and $S$ are given by

$$(Ks_c s_c^T)_{ij} = \begin{cases} \frac{(K-1)^2}{K}, & \text{if } i = j = c \\ -\frac{K-1}{K}, & \text{if } i = c, \text{ or } j = c, \text{ but } i \neq j \\ \frac{1}{K}, & \text{otherwise} \end{cases}$$

$$S_{ij} = \begin{cases} \frac{K-1}{K}, & \text{if } i = j \\ -\frac{1}{K}, & \text{otherwise} \end{cases}$$

Looking at this case by case, it is clear that $\rho_c' = Ks_c s_c^T + S$.

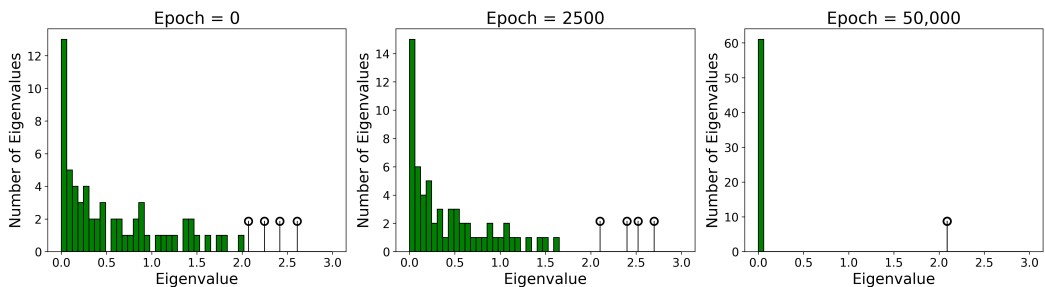

Figure 7: Histogram of the eigenvalues of $W_l^T W_l$ for the deep linear UFM at an intermediate layer, shown across a range of training epochs. The top $K = 4$ outliers are plotted separately as spikes.

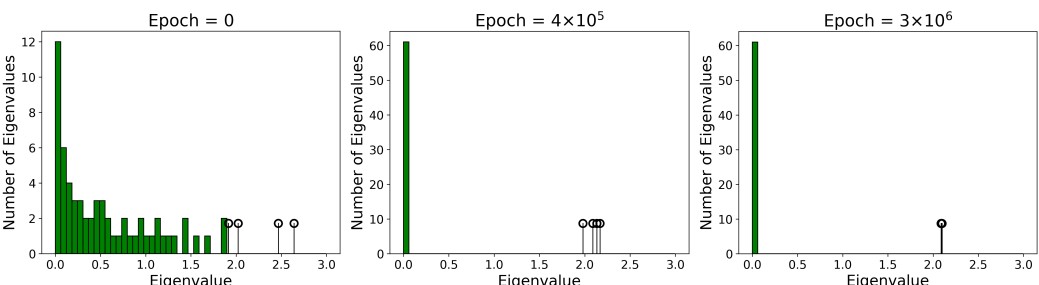

Figure 8: Histogram of the eigenvalues of $W_l^T W_l$ for the deep ReLU UFM at an intermediate layer, shown across a range of training epochs. The top $K = 4$ outliers are plotted separately as spikes.

## G  FURTHER NUMERICAL EXPERIMENTS

In this section, we provide additional details about the experiments in the main text, as well as further experiments involving the weight matrices and Papyan's decomposition. We also present further experiments on standard DNNs applied to canonical datasets.

### G.1  FURTHER EXPERIMENTS IN THE DEEP UFM

**More Details about the Main Text Experiments:** We begin by elaborating on the metrics used for the figures in Section 5 of the main text.

To measure the extent to which $\{\mu_c^{(l+1)} \otimes \mu_{c'}^{(l)}\}_{c,c'=1}^K$ form eigenvectors of $\mathrm{Hess}_l$, as shown in Figures 1 and 4, we used the cosine similarity squared between $\mu_c^{(l+1)} \otimes \mu_{c'}^{(l)}$ and $\mathrm{Hess}_l(\mu_c^{(l+1)} \otimes \mu_{c'}^{(l)})$, given by

$$f_{cc'} = \frac{|(\mu_c^{(l+1)} \otimes \mu_{c'}^{(l)})^T \mathrm{Hess}_l(\mu_c^{(l+1)} \otimes \mu_{c'}^{(l)})|^2}{\|(\mu_c^{(l+1)} \otimes \mu_{c'}^{(l)})\|_2^2 \|\mathrm{Hess}_l(\mu_c^{(l+1)} \otimes \mu_{c'}^{(l)})\|_2^2}.$$

This metric equals one precisely when $\mu_c^{(l+1)} \otimes \mu_{c'}^{(l)}$ is an eigenvector of $\mathrm{Hess}_l$.

To analyze the decomposition of the gradient $\tilde{g}^{(l)}$ in the natural basis $\{\mu_c^{(l+1)} \otimes \mu_{c'}^{(l)}\}_{c,c'=1}^K$, also shown in Figures 1 and 4, we used the cosine similarity squared between $\tilde{g}^{(l)}$ and $\mu_c^{(l+1)} \otimes \mu_{c'}^{(l)}$, given by

$$\tilde{f}_{cc'} = \frac{|(\mu_c^{(l+1)} \otimes \mu_{c'}^{(l)})^T \tilde{g}^{(l)}|^2}{\|\mu_c^{(l+1)} \otimes \mu_{c'}^{(l)}\|_2^2 \|\tilde{g}^{(l)}\|_2^2}.$$

When the vectors $\mu_c^{(l+1)} \otimes \mu_{c'}^{(l)}$ form an orthogonal basis, these coefficients sum to 1.

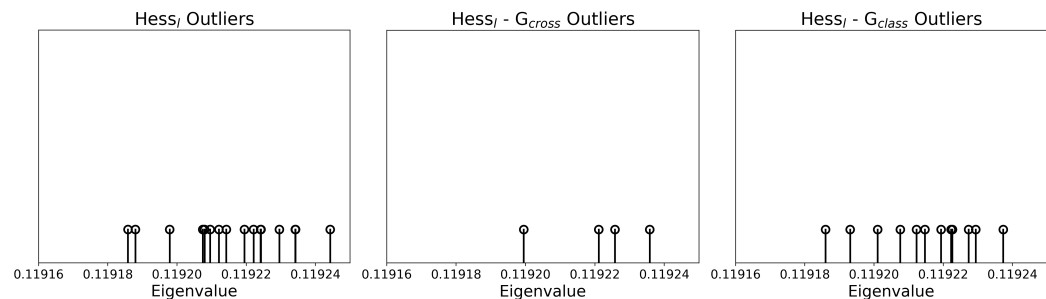

Figure 9: Plot of the top $K^2 = 16$ eigenvalues of $\mathrm{Hess}_l$ for the deep linear UFM after 50,000 epochs, together with knockouts of each component from the decomposition in equation 5.

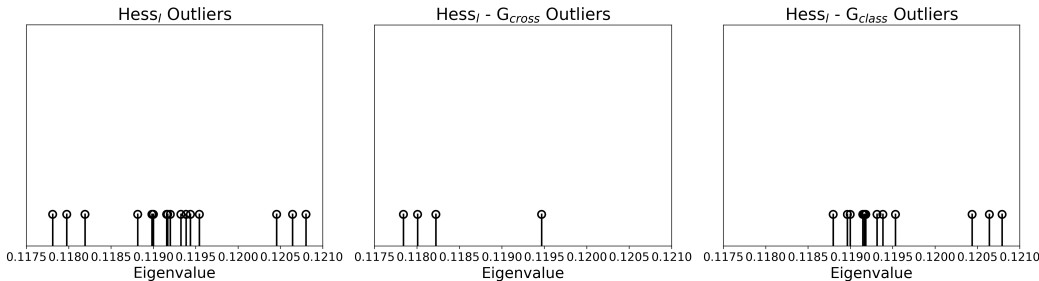

Figure 10: Plot of the top $K^2 = 16$ eigenvalues of $\mathrm{Hess}_l$ for the deep ReLU UFM after $10^6$ epochs, together with knockouts of each component from the decomposition in equation 5.

To assess the extent to which the matrix $\bar{H}_l^T \bar{H}_l$ forms an orthogonal frame, as shown in Figure 3, we considered the Frobenius distance after normalization between $\bar{H}_l^T \bar{H}_l$ and $I$, defined as

$$M_l = \left\| \frac{\bar{H}_l^T \bar{H}_l}{\|\bar{H}_l^T \bar{H}_l\|_F} - \frac{I_K}{\sqrt{K}} \right\|_F .$$

This metric is non-negative, taking the value zero precisely when $\bar{H}_l^T \bar{H}_l$ forms an orthogonal frame.

**More Experiments in deep UFMs:**

We now provide experimental evidence for the remaining theorems in the main text. We begin with plots of the spectrum of the Gram matrix $W_l^T W_l$ at various points in training, corresponding to the squared singular values. These spectra are shown in Figure 7 for the linear case and in Figure 8 for the ReLU case. As with the Hessian, the top eigenvalues initially blend into the bulk but progressively separate during training, eventually converging uniformly to an identical value. Simultaneously, the bulk eigenvalues collapse to a single atom at zero, in agreement with Theorem 4.

To investigate the decomposition results described in Theorem 2, we display the Hessian's outlier eigenvalues at a late stage of training, along with their behavior when each component of the decomposition from equation 5 is individually removed. These results are shown in Figure 9 for the linear case and Figure 10 for the ReLU case. We omit the case where $G_{\mathrm{within}}$ is subtracted, since the spectrum is identical to that of the full Hessian, as predicted by our theory. We observe that removing $G_{l,\mathrm{cross}}$ and $G_{l,\mathrm{class}}$ eliminates exactly $K(K-1) = 12$ and $K = 4$ outlier eigenvalues, respectively, as predicted by Theorem 2.

Finally, we examine the eigenvalues of the feature Gram matrix $\bar{H}_l^T \bar{H}_l$ in the ReLU case, shown in Figure 11. In Section 4, we claimed that the network quickly develops non-negative entries and forms an orthogonal frame with an additional rank-one perturbation to enforce non-negativity. This spike then decays over the course of training. This can be seen in the figure: at intermediate times

there is one dominant spike relative to the others, but over a longer time horizon it converges to the same value as the remaining spikes.

## G.2 FULL NETWORKS WITH UFM STYLE REGULARIZATION

Here we provide further experimental details and evaluations of how the theorems in the main text manifest in DNNs trained on the MNIST (Lecun et al., 1998) and CIFAR-10 (Krizhevsky & Hinton, 2009) datasets. For the MNIST experiments, we subsample 5,000 examples per class to match the class balance of CIFAR-10. Input data is preprocessed by subtracting the mean and dividing by the standard deviation. We use the ResNet-20 architecture as the feature map $h(x; \bar{\theta})$, followed by four linear layers of width $d = 64$. We use UFM style regularization, regularizing the outputs of the feature map and the layers in the linear head. The regularization parameter is set to $\lambda = 5 \times 10^{-4}$, except for the feature layer, where it is set to $\lambda_H = 1 \times 10^{-7}$. This lower value accounts for the impact of the number of data points on the overall regularization strength. As in the UFM experiments, we focus on the layer $l = 3$.

For MNIST, we train for 4,000 epochs, starting with a learning rate of 0.04, which is halved after 2,000 epochs. For CIFAR-10, we train for 5,000 epochs, starting with a learning rate of 0.05, halved after 2,500 epochs. We use batch gradient descent with a batch size of 10,000 to approximate full gradient descent, consistent with the model. We present detailed results from individual runs, noting that the conclusions are robust provided the regularization is not so large as to enforce the trivial zero solution.

Figures 12 and 14 show the eigenvalues of Hess$_l$, together with the knockouts of Papyan's decomposition described in equation 5. The $G_{l,\text{within}}$ case is omitted, as its spectrum is identical to that of Hess$_l$. We observe that while the predicted number of eigenvalues associated with each component is preserved, exact equality does not hold. This discrepancy arises because the unconstrained feature assumption is only approximate in practice, introducing noise. We also show the weights in Figures 13 and 15, with the same conclusions as in the model results presented in Appendix G.1.

Across all plots, CIFAR-10 displays greater noise and weaker convergence compared to MNIST. This is intuitive: CIFAR-10 is a more complex dataset, requiring more sophisticated DNNs to achieve the same level of overparameterization. Consequently, the unconstrained feature assumption holds less closely, introducing additional noise into the results.

In Figure 16, we show how, for the MNIST network, the predicted left singular vectors $\hat{\mu}_c^{(l+1)}$ and predicted right singular vectors $\hat{\mu}_c^{(l)}$, for $c = 1, \dots, K$, align with the true singular vectors. This alignment is quantified using the cosine similarity computed before and after applying the appropriate Gram matrix of the weights. We observe that our theoretical predictions for the singular vectors emerges rapidly during training.

The effective rank, denoted $f_{\text{ER}}$, of a matrix $M$ with singular values $\sigma_1, \dots, \sigma_m$, is given by

$$f_{\text{ER}} = \exp\left(-\sum_{i=1}^{m} \frac{\sigma_i}{\sum_j \sigma_j} \log\left(\frac{\sigma_i}{\sum_j \sigma_j}\right)\right) \tag{25}$$

with the convention that $0 \log(0) = 0$. In Figure 17, we show the effective ranks of the layerwise Hessian and weight matrix throughout training. In both cases, we see that the effective rank converges to the value predicted by of our theory for the true rank. This indicates that both matrices exhibit the correct set of large outliers together with very small bulk values. The weight matrix approach its convergence value from above, whereas the Hessian approaches its convergence value from below. This behavior is due to the initialization and architecture, rather than any specific aspect of our theory, and understanding these effects represent an interesting direction for future work.

We also plot the mean and standard deviation of the outlier eigenvalues of the weights and the Hessian for the MNIST network in Figure 18. In both cases, we again see that they converge to a single value with very little variability in their distribution.

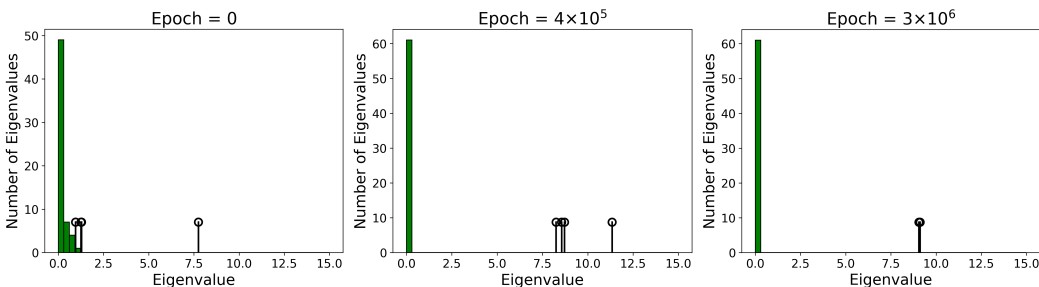

Figure 11: Histogram of the eigenvalues of $\bar{H}_l^T \bar{H}_l$ for the deep ReLU UFM at an intermediate layer, shown across a range of training epochs. The top $K$ outliers are plotted separately as spikes.

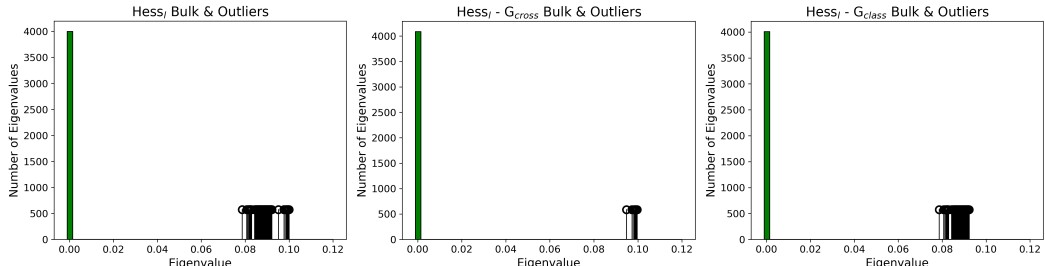

Figure 12: Training of a DNN applied to MNIST after 4000 epochs. **Left:** Histogram of the eigenvalues of $\text{Hess}_l$, with the top $K^2 = 100$ plotted as spikes. **Middle:** eigenvalues of $\text{Hess}_l - G_{l,\text{cross}}$, with the top $K = 10$ plotted as spikes. **Right:** eigenvalues of $\text{Hess}_l - G_{l,\text{class}}$, with the top $K(K-1) = 90$ plotted as spikes.

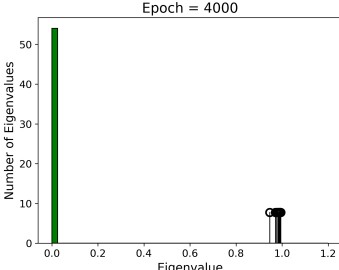

Figure 13: Histogram of the eigenvalues of $W_l^T W_l$ for a DNN applied to MNIST at an intermediate layer. The top $K$ outliers are plotted separately as spikes.

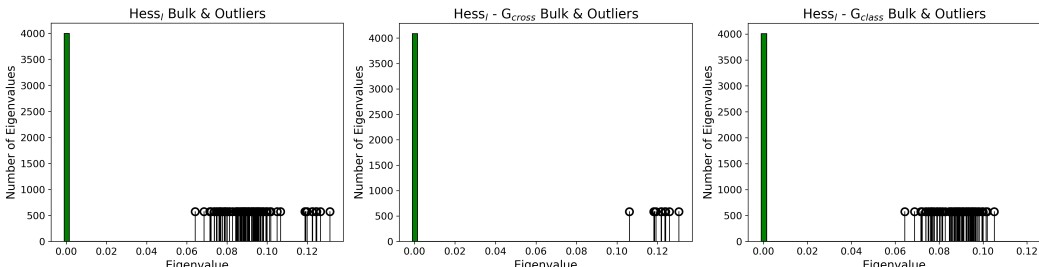

Figure 14: Training of a DNN applied to CIFAR-10 after 5000 epochs. **Left:** Histogram of the eigenvalues of $\text{Hess}_l$, with the top $K^2 = 100$ plotted as spikes. **Middle:** eigenvalues of $\text{Hess}_l - G_{l,\text{cross}}$, with the top $K = 10$ plotted as spikes. **Right:** eigenvalues of $\text{Hess}_l - G_{l,\text{class}}$, with the top $K(K-1) = 90$ plotted as spikes.

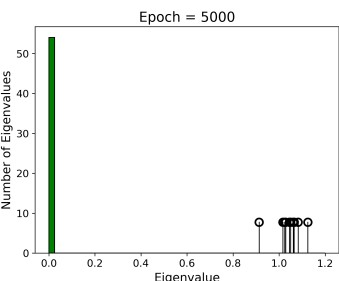

Figure 15: Histogram of the eigenvalues of $W_l^T W_l$ for a DNN applied to CIFAR-10 at an intermediate layer. The top $K$ outliers are plotted separately as spikes.

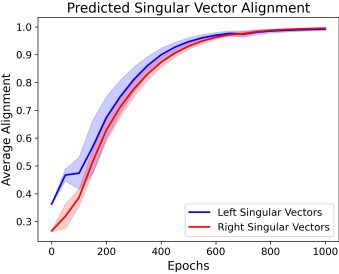

Figure 16: Extent to which the predicted left singular vectors $\hat{\mu}_c^{(l+1)}$ and predicted right singular vectors $\hat{\mu}_c^{(l)}$ align with the true singular vectors of the weight matrix $W_l$ for a DNN at an intermediate separated layer $l$ trained on MNIST. Alignment is measured using the cosine similarity before and after applying the appropriate Gram matrix. In both cases, the cosine similarity averaged over $c = 1, \ldots, K$ is shown, with one–standard-deviation error bars.

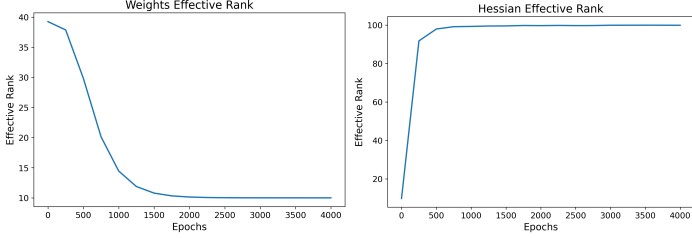

Figure 17: plots of the effective ranks, given in equation 25, of the layerwise Hessian and weight matrix throughout training for an intermediate separated layer of a DNN trained on MNIST.

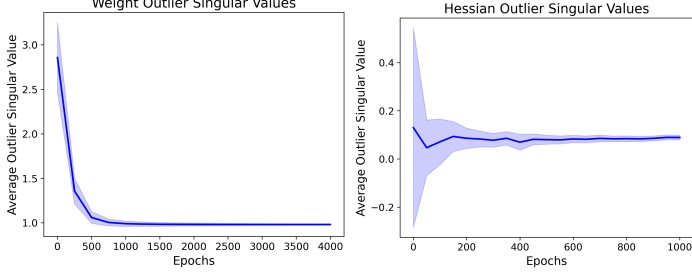

Figure 18: Outlier singular values of the weight matrix and layerwise Hessian fro an intermediate layer of a DNN trained on MNIST. For the weights, these correspond to the top $K$ singular values; for the Hessian, the top $K^2$ eigenvalues. In both cases, the average outlier value is shown, with one-standard-deviation error bars.

