# OpenReview forum: "Unifying Low Dimensional Observations in Deep Learning"
_ICLR.cc/2026/Conference — Submitted to ICLR 2026_

### Official Review · Reviewer_BQ76 · 2025-10-20

**Soundness:** 4
**Presentation:** 4
**Contribution:** 2
**Rating:** 4
**Confidence:** 4

**Summary:**

The paper analyzes Hessian spectra in deep learning. The authors formally connect the deep neural collapse phenomenon with the empirically observed eigenvalue bulks of the Hessian in well-trained DNNs. The authors mostly operate in the deep unconstrained feature regime, although some of their experiments are done with real networks. The authors first focus on the deep linear case and analyze per-layer Hessian spectra. They express the Hessian of the UFM-optimal solutions and compute its eigendecomposition, showing that they can be expressed in the form of class-means and the eigenvalues form bulks similar to those observed in the related work. Then they interpret the Fisher Information Matrix of UFM-optima and its decomposition into within-class, cross-class and class component in terms of the DNC structure. They also show that gradients align with the dominant eigenspace. Within the deep linear setting, they also provide generalizations into the full Hessian. In the non-linear deep unconstrained feature model regime, they show that DNC solutions’ Hessians exhibit the same structure and show that global optima among DNC solutions are those that can be expressed linearly as well. They support their theory with experiments done either directly on deep unconstrained features model or on real networks.

**Strengths:**

-	S1: The paper establishes a formal connection between deep neural collapse and outlier eigenvalues / bulks in the Hessians. This is an important (and rather surprisingly) missing piece of the puzzle.
-	S2: The paper is very well written and sound.

**Weaknesses:**

-   W1: At times, the authors slightly overstate their contribution. In particular, the related work already did make the connection between the neural collapse and Hessian spectra. Already [1], as also described by the authors, talk about the within-class, between-class and class contributions to the eigenvalue distributions. The link is later made explicit in [2] by the same author(s). It is also mentioned by some other works, such as [3] (although not explicitly as neural collapse). This goes against some of the sentences of the paper such as the third paragraph of introduction or line 63. While I agree that this paper certainly makes the connection most explicit and shows it formally, it is unfair to say that the entire idea is novel. I also think the title of the paper could be a bit less generic, as there are many types of low-dimensional phenomena in deep learning and this is only a very small subset of them.
-	W2: The attempts the authors make to generalize their linear UFM findings on per-layer Hessian to either full Hessian or non-linear setting are not very strong. As for the per-layer to full Hessian generalization, I find the regime of Theorem 6 to be very unrealistic and thus the connection with full network is unconvincing (W2A). As for the non-linear case, there are even two issues. The first one is that the analysis is only done for DNC solutions, however we know from [4, 5] that different structures also often emerge both in DUFM as well as in real networks (W2B). In general, the authors should unify (to live up to the title’s standard) the analysis of neural collapse with other low-rank phenomena, such as the bottleneck rank [5]. The second issue is the acknowledged non-differentiability of the loss at global minima. While the authors say that limit of smoothed ReLUs would solve the issue, I disagree as the limit should be independent of the way how the ReLU is smoothed, yet different smoothings would lead to different reasonable derivative definitions.
-	W3: The authors only analyze the perfect scenario. However, in real networks, the collapse is not only not achieved perfectly (smaller problem which would only require a perturbation analysis (still would be nice, though)), but, more importantly, it is achieved gradually. This means that at a constant fraction of the layers, the within-class variability is expected to only be removed to an extent that depends on that fraction. This relationship might be linear or log-linear or more complex (depending on the architecture), but one should not expect full collapse already at a constant fraction of layers. Authors should, therefore, discuss and potentially explain how the Hessian eigenvalue laws can be explained in this progressive collapse regime. In this regard, I would also recommend the authors to rephrase lines 310-312 as the Theorem 6 certainly doesn’t cover the “worst-case” scenario.
-	W4: A minor issue – the definition of DNC3 in the paper seems to be rather weak. This condition is almost trivially satisfied in most reasonable convergence assumptions. [4, 6] and other works on deep neural collapse work with stronger definitions.

[1] Papyan, Vardan. "Traces of class/cross-class structure pervade deep learning spectra." Journal of Machine Learning Research 21.252 (2020): 1-64.

[2] Papyan, Vardan, X. Y. Han, and David L. Donoho. "Prevalence of neural collapse during the terminal phase of deep learning training." Proceedings of the National Academy of Sciences 117.40 (2020): 24652-24663.

[3] Granziol, Diego, Stefan Zohren, and Stephen Roberts. "Learning rates as a function of batch size: A random matrix theory approach to neural network training." Journal of Machine Learning Research 23.173 (2022): 1-65.

[4] Súkeník, Peter, Marco Mondelli, and Christoph H. Lampert. "Deep neural collapse is provably optimal for the deep unconstrained features model." Advances in Neural Information Processing Systems 36 (2023): 52991-53024.

[5] Jacot, Arthur. "Bottleneck structure in learned features: Low-dimension vs regularity tradeoff." Advances in Neural Information Processing Systems 36 (2023): 23607-23629.

[6] Rangamani, Akshay, et al. "Feature learning in deep classifiers through intermediate neural collapse." International conference on machine learning. PMLR, 2023.

**Questions:**

-	Q1: What is the difference between $\mu$ and $\hat \mu$?
-	Q2: The eigenvectors that the gradient is aligned with identified in Theorem 3 do not seem to correspond to the Theorem 2 (c) class eigenvectors. This is somewhat surprising because in the cross-entropy case where the gradient is aligned with the top K eigenvectors, these should be the class eigenvectors. What is the cause of this seemingly counterintuitive difference?
-	Q3 / comment: The proof of Theorem 8 could perhaps be simplified. To prove that global optima of the non-linear case must have non-negative activations, a simple single-layer contradiction and the use of Theorem 4 could be used where one would construct higher norm solution for the same cost that could then be compensated with appropriately smaller weight matrix in the subsequent layer.

**Summary:**
Although I do have some concerns and I don’t consider the contribution of the paper to be groundbreaking (in particular because the paper only formalizes in simplified setting what has already been voiced and empirically measured), I still think the paper fills in an important gap in theory of the optimization landscape and soundly formalizes the ideas from the related work. Therefore, should the authors satisfiably address my concerns, I would be open to recommend it for acceptance. However, I cannot do so yet, without a successful discussion with the authors.

---

> ### Author Response · Authors · 2025-11-16
>
> We thank the reviewer for their detailed assessment of our work, which will substantially improve the paper. We address each concerns below. We would be grateful if the reviewer could reconsider their score if they find our responses satisfactory.
>
> **The authors slightly overstate their contribution … While I agree that this paper certainly makes the connection most explicit and shows it formally, it is unfair to say that the entire idea is novel.**
>
> In preparing this work, we looked for prior references linking DNC to deep-learning spectra and found the comment that you mention. However, Papyan et al. discuss only NC, not DNC, and our results show that NC alone cannot account for the observed spectral phenomena—DNC is required. Moreover, their remark provides little mathematical explanation, which is likely why the role of DNC was previously overlooked. For these reasons, we interpreted it as a conjectural and incomplete link.
>
> Regarding the work of Granziol et al. and related papers: while these works note that structure in features induces structure in the Hessian, they model only subsets of the full behaviour and do not incorporate NC/DNC. We view this as an interesting previous exploration of the Hessian, but given the importance of NC in the literature, the absence of this connection—and the fundamental geometric component it introduces—makes such results only tangentially related to our own. We will add an expanded related-work section to the appendix where this context will be included.
>
> We agree, however, that our phrasing could be clearer. Our intention was to claim a unified mathematical explanation rather than a heuristic or empirical connection (which may appear partially in earlier work). We will revise the text accordingly. We nonetheless believe the contribution is substantively novel.
>
> **The title of the paper could be a bit less generic.**
>
> We appreciate this helpful suggestion. Our title aimed to highlight the significance of DNC and gradient alignment, which extend beyond spectral phenomena, but we agree that this choice introduced ambiguity. We will change the title to “Unifying Low-Dimensional Spectra in Deep Learning”.
>
> **I find the regime of Theorem 6 to be unrealistic and thus the connection with full network is unconvincing.**
>
> Theorem 6 is a limiting argument showing that, in the large-depth regime, the full Hessian converges (up to scale) to the block Hessian of the deep UFM described in Theorem 5. Concretely, the regime first takes the number of layers in the feature map $\bar{L}$ to be large (so that the UFM approximation is accurate), and then takes the number of separated layers $L$ to be large so that the DNC component dominates. A clearer description may be to frame this as a joint limit $\bar{L},L→∞$ with $\bar{L}/L \to 0$.
>
> It is reasonable to question whether such a limit informs the behavior of finite-depth networks—this is similar to questioning the UFM assumption itself. As in prior work, our modelling relies on an overparameterization limit that is not explicitly quantified and, in principle, could fail if finite networks never approach that regime. This is precisely why it is important to empirically test the model, as we do in Appendix F (and as others have done, see Related Work), and to check that the results correspond with the empirical phenomena we are studying.
>
> If we have misunderstood your concern about the realism of the regime, we would be grateful for clarification.
>
> **As for the non-linear case … the analysis is only for DNC, however … different structures also often emerge both in DUFM as well as in real networks … the authors should unify (to live up to the title’s standard) the analysis of NC with other low-rank phenomena.**
>
> The reference you cite from Sukenik et al. may be incorrect; we assume you are referring to their other work [1]. If not, please let us know. Their results show that with ReLU activations, DNC is not globally optimal due to a low-rank bias induced by regularization. Jacot et al. describe a similar low-rank effect. We mention these structures in Section 4 but do not focus on them because they rarely appear in real networks trained under standard hyperparameter regimes, as shown by Papyan et al. [2]. Sukenik et al. also observe that for hyperparameter choices closer to those used in practice, DNC is more likely to emerge (see their Figure 3).
>
> It is an open question why these alternative low-rank solutions are not commonly seen in real networks, though several explanations have been proposed [1,3,4]. Our goal in this work was to analyze the spectral phenomena that arise specifically in the same regime as DNC, which is why we restricted our attention to DNC. The structures identified by Sukenik et al. still satisfy DNC1, so we expect portions of our analysis to extend to them, but given the arguments we make here, we view this as an interesting direction for future work. As mentioned, the title will be changed.

---

> ### Author Response · Authors · 2025-11-16
>
> **The acknowledged non-differentiability of the loss at global minima. While the authors say that limit of smoothed ReLUs would solve the issue, I disagree as the limit should be independent of the way how the ReLU is smoothed…**
>
> We first clarify that we do not claim that a smoothed-ReLU limit resolves the issue; we only suggest it as something we intend to explore, based on a similar idea to Theorem 8 of Sukenik et al. [1], which uses a sequence of ReLU approximations with controlled derivatives. In Section 4, we show that the global minima among DNC solutions can be expressed with linear networks, at which point the Hessian and gradient are not well defined because of non-differentiability.
>
> In training, we do not reach exact convergence, so we instead obtain small entries approaching zero from either side of the axis, leading to a Bernoulli-like pattern in the Hessian. This is precisely what we observed empirically, which was initially surprising given that the weights behaved exactly as in the linear model; the non-differentiability merely obscures the underlying structure. We set $σ′(0)=1$ to make this explicit and to indicate that the same Hessians can be recovered with a minimal convention. Due to space constraints, we could not fully elaborate on this point, but we would be happy to expand the discussion in a revision.
>
> **In real networks, the collapse … is achieved gradually … at a constant fraction of the layers, the within-class variability is expected to only be removed to an extent that depends on that fraction…. One should not expect full collapse already at a constant fraction of layers. Authors should discuss and potentially explain how the Hessian eigenvalue laws can be explained in this progressive collapse regime.**
>
> We agree that collapse strengthens with depth. Intuitively, the degree of collapse depends on the level of overparameterization: a smaller subset of layers can overparameterize the data less, while a larger subset can achieve a stronger collapse. However, adding additional layers does not reduce the collapse achievable by earlier layers. Thus, if after $L′$ layers the network achieves NC up to some small threshold, all subsequent layers will satisfy NC to at least that threshold. In this sense, the proportion of layers exhibiting collapse up to a fixed $ε$-threshold tends to 1 (rather than remaining constant) as depth increases.
>
> We assume the feature map is expressive enough that we operate within a reasonable collapse threshold for the separated layers, and the discrepancy between the UFM and real networks diminishes as this threshold decreases. Our experiments confirm that the resulting approximation accurately captures deep-learning spectral behavior. We also note that He & Su [5] empirically study the rate of DNC1 across layers and identify a geometric decay, which likely contributes to the strong agreement between our model and practice.
>
> For these reasons, we do not view the progressive-collapse regime as problematic for our theory. Given the extent of the present work, a full perturbation analysis is beyond our scope; moreover, an entire paper is devoted to such an analysis [6], which further justifies leaving it for future work.
>
> **Theorem 6 certainly doesn’t cover the “worst-case” scenario**
>
> We wanted to highlight that the limiting argument assumes nothing about the blocks in the feature map other than their scales. Consequently, we expect the convergence to be stronger in reality, since the feature map will also display some amount of DNC. You are right that we make simplifying assumptions that do not represent the worst case, but those assumptions are deemed to hold in the limit, so it is a “worst-case” limiting argument, not a worst-case argument in the absolute sense. We apologize for the confusion and will reword this comment.
>
> **The definition of DNC3 in the paper seems to be weak. This condition is almost trivially satisfied in most reasonable convergence assumptions.**
>
> DNC3 does not admit a clean generalization of NC3, which has led to ambiguity in the literature. In our theoretical results, we do not rely on the specific form of DNC3. We will revise our definition to align with that of Sukenik et al. [1].
>
> **What is the difference between $μ$ and $\hat{μ}$**
>
> By $\hat{\mu}$ we mean the normalized vectors, scaled to have unit Euclidean norm. We apologize for not defining this explicitly and will correct it in the revised manuscript.
>
> **The eigenvectors that the gradient is aligned with identified in Theorem 3 do not correspond to the Theorem 2 class eigenvectors.**
>
> In MSE, the bulk and minibulk are not separated. We have not yet performed the corresponding CE gradient analysis, but we expect that the clearer bulk–minibulk structure in CE resolves this mismatch.
>
> **The proof of Theorem 8 could be simplified.**
>
> Thank you for this perceptive comment. We will look into this, but we are currently prioritizing the main paper text and additional experiments.

---

> > ### Author Response · Authors · 2025-11-16
> > **References**
> >
> > [1] Peter Sukenik, Marco Mondelli and Christoph H. Lampert. Neural Collapse versus Low-rank Bias: Is Deep Neural Collapse Really Optimal? In Advances in Neural Information Processing Systems, 2024
> >
> > [2]  Vardan Papyan, Xuemei Han, and David L. Donoho. Prevalence of neural collapse during the terminal phase of deep learning training. Proceedings of the National Academy of Sciences of the United States of America, 2020.
> >
> > [3] Connall Garrod and Jonathan P. Keating. The persistence of neural collapse despite low-rank bias: An analytic perspective through unconstrained features. ArXiv preprint, 2024.
> >
> > [4] Peter Sukenik, Christoph H. Lampert and Marco Mondelli. Neural Collapse is Globally Optimal in Deep Regularized ResNets and Transformers, ArXiv preprint, 2025.
> >
> > [5] Hangfeng He and Weijie Su. A law of data separation in deep learning. Proceedings of the National Academy of Sciences of the United States of America, 2022.
> >
> > [6] Tom Tirer, Haoxiang Huang and Jonathan Niles-Weed. Perturbation Analysis of Neural Collapse. Proceedings of the 40th International Conference on Machine Learning, 2023.

---

> ### Comment · Reviewer_BQ76 · 2025-11-17
> **Thank you for detailed explanations**
>
> Thank you for the detailed explanations. Some points seem to be resolved. I will only react to the points that require further discussion.
>
> - **Contextualization w.r.t. related work:**
>
> Thank you for clarification. Please include most of this discussion and changes in the main body (instead of the appendix). But regarding the NC vs. DNC -- If I understand it correctly, the per-layer hessian results would also hold with NC only, but only in the last layer, right?
>
> - **Regarding Theorem 6 and regarding the progressive/gradual deep neural collapse from the next comment:**
>
> I Address these two issues together, as they basically require the same argument. To further clarify my comment regarding Theorem 6, I think the main concern is taking the limits in this order. This would mean that we assume that DNC is reached in a fraction of the layers that converges to 0 as the number of layers goes to infinity. However, looking at related work, for instance He&Su 2022 which you mention yourself, we see that this seems not to be the case. In particular, the DNC1 metric seems to not progress at all in a vanishing fraction of the layers if the law is indeed geometric. From this, I think we can derive that the DNC assumption of theorem 6 cannot hold in this limiting regime. For the very same reason I think that progressive neural collapse is also relevant for you in our discussion point in the next comment.
>
> I can also explain it using regularized deep linear network: we know that in that case, the singular values of the layers in global optima split equally between them. Thus, the DNC2, for instance, can be shown to be geometrically progressive. Similar argument holds in ResNets -- if we assume they approximate a neural ODE, then again, collapse should be progressive and nothing should happen in a vanishing fraction of the layers.
>
> Regarding Theorem 6 still, one more condition that might not hold is the equal scale of the sub-hessians. Did you measure such a thing in practice?
>
> Regarding your comment on experiments in Appendix F -- if I understand it correctly, these experiments do not measure the conditions of Theorem 6. In particular, there are no measurements about the global Hessian, all the measurements are done for single-layer subhessians.
>
> [5] Hangfeng He and Weijie Su. A law of data separation in deep learning. Proceedings of the National Academy of Sciences of the United States of America, 2022.

---

> > ### Author Response · Authors · 2025-11-17
> >
> > Thank you again for your thoughtful and constructive questions. We sincerely appreciate the time and care you are devoting to evaluating our paper. Below, we respond to the remaining points of discussion.
> >
> > **related work**:
> >
> > We will be adding a related-work section to the appendix to address another reviewer’s request regarding empirical evaluations of UFMs, but we will ensure that the clarifications you suggested are incorporated into the main body.
> >
> > Regarding NC vs. DNC: yes, NC alone suffices to characterize the Hessian at one layer, but not layer-wise across the network or for global Hessian quantities. Since Papyan et al. do not study layer-wise Hessians in [1] and focus on global Hessian structure in [2–4], this is consistent with how we interpret their comment.
> >
> > **Theorem 6 and gradual collapse.**
> >
> > Thank you for flagging the concern about the order of limits. After revisiting the statement, we agree that our description was not sufficiently clear. Below, we clarify the setup. Let $\bar{L}$ denote the depth of the feature map and $L$ the depth of the separated component.
> >
> > **The main concern is taking the limits in this order**: Stated appropriately for the model, we are taking a joint limit (rather than one after the other): $\bar{L}, L \to \infty$, such that $\bar{L}/L \to 0$. For example, $\bar{L} = L^{1/2}$.
> >
> > **This would mean that we assume that DNC is reached in a fraction of the layers that converges to 0 as the number of layers goes to infinity**: Actually this induces the opposite, which the more concise statement of the limit makes clear: the fraction of layers that exhibit DNC is
> > $$\frac{L}{\bar{L} +L} \to 1,$$
> >
> > rather than zero.
> >
> > For real networks this boils down to making the depth large, since the separation is somewhat arbitrary. Increasing $\bar{L}$ leads to a better approximation of the separated layers by the UFM, whilst increasing $L$ leads to a better approximation of the full Hessian using the UFM. but if we define a threshold $\epsilon$ of collapse that we deem close enough to the UFM, and define the feature map to start at this point, then increasing depth increases the Hessian approximation part, and the result depends only on the UFM being accurate when the threshold is small enough, and sufficient network depth after the threshold is hit.
> >
> > We do not see it as an issue that within the layers after the feature map the level of collapse continues to get better (i.e. more collapse), we only need that it does not get worse (i.e. less collapse), since then each is below our threshold, which is capturing the approximation.
> >
> > **However, looking at related work… we see that this seems not to be the case… the DNC1 metric seems to not progress at all in a vanishing fraction of the layers if the law is indeed geometric. From this, I think we can derive that the DNC assumption of theorem 6 cannot hold in this limiting regime.**
> >
> > We are confused by the phrasing ‘ the DNC1 metric seems to not progress at all in a vanishing fraction of the layers if the law is indeed geometric’, for our theory to hold we want DNC to increase in each layer, which seems quite clear from the results of He and Su, and what we think is intended by your phrasing.
> >
> > We hope that clarifying the limit more succinctly has clarified why the assumption essentially boils down to the UFM assumption?
> >
> > **For the same reason I think that progressive neural collapse is also relevant for you in our discussion point in the next comment.**
> >
> > We hope the above then consequently resolves both of these issues in your mind. If not, please let us know.
> >
> > **Theorem 6 … one condition that might not hold is the equal scale of the sub-hessians. Did you measure such a thing in practice?**
> >
> > You are correct that we do not evaluate that condition. Our first reason for this is that ReLU networks are homogenous, so we anticipate the scales of the feature map being controlled as a consequence of this and the regularization term. The other reason relates to why we did not empirically test Theorem 6.
> >
> > **experiments in Appendix F … these experiments do not measure the conditions of Theorem 6… there are no measurements about the global Hessian**
> >
> > Theorem 6 is a claim about eigenvalues. Our other theoretical results go beyond previous observations by characterizing eigenvectors, and we believed it important to test such theory given the result is novel even on an empirical level. Theorem 6 corresponds to the regime empirically evaluated by Papyan and many others. Consequently, their results are exactly the ones we would have shown to justify the theory. In particular, there are examples where people note the level of noise in the spectrum decreases as you increase the number of parameters [5], which exactly corresponds with predictions based on the modelling assumptions.
> >
> > Once again, we greatly appreciate the care and time you are investing in your review, and we welcome any further questions or suggestions you may have.

---

> > > ### Author Response · Authors · 2025-11-17
> > > **more References**
> > >
> > > [1] Vardan Papyan, Xuemei Han, and David L. Donoho. Prevalence of neural collapse during the terminal phase of deep learning training. Proceedings of the National Academy of Sciences of the United States of America, 2020.
> > >
> > > [2] Vardan Papyan. Traces of class/cross-class structure pervade deep learning spectra. Journal of Machine Learning Research, 2020.
> > >
> > > [3] Vardan Papyan. Measurements of three-level hierarchical structure in the outliers in the spectrum of deep net Hessians. ArXiv preprint,2019
> > >
> > > [4] Vardan Papyan. The Full Spectrum of Deep Net Hessians At Scale: Dynamics with Sample Size, ArXiv preprint, 2018.
> > >
> > > [5] Levent Sagun, Utku Evci, V. U. Guney, Yann Dauphin and Leon Bottou. Empirical Analysis of the Hessian of Over-Parametrized Neural Networks. In the International Conference on Learning Representations, 2017.

---

> > > ### Comment · Reviewer_BQ76 · 2025-11-18
> > >
> > > Thank you for your prompt responses. You convinced me about the related work, the equal scale condition and the experiments in appendix F. Let me elaborate on the gradual collapse, where it seems there is a misunderstanding.
> > >
> > > - **How I meant my progressive collapse concern:**
> > >
> > > I am sorry for confusion it seems my sentence *DNC is reached in a fraction of the layers that converges to 0 as the number of layers goes to infinity* can be interpreted in both opposite ways. What I meant is *not* that the DNC is present in a fraction of layers that converges to 0, but exactly the opposite. Let $A$ be the input, $B$ be the output of the first $\bar{L}$ layers (the feature map) and $C$ the output of the network. What I meant is that since you assume that $B$ is already collapsed and the collapse persists all the way to $C$, this means that the fraction of the layers that *make the features collapsed*, i.e. the first $\bar{L}$ layers goes to 0 in your setting, which is consistent with your description.
> > >
> > > Now if we agree that this is true (which I hope we do now) then my claim is that this is a problem. Let $DNC(l)$ denote the extent of DNC after $l$ layers (zero is perfect collapse). Now, looking at the results of He&Su 2022 (Figure 5), we see that $DNC(\bar{L}+L)$ is more-or-less constant as a function of the total number of layers. Denote this as $C$ and assume it is fairly small. Assume that $DNC(0)=1.$ Then based on the observations from the paper we can write $DNC(l)=\exp(\frac{l}{L}\log(C)).$ Now assume that $\frac{\bar{L}}{\bar{L}+L}=p$ and this $p$ depends on the total number of layers and converges to 0 as the total number of layers goes to infinity. In this case we have $DNC(\bar{L})=DNC(p(\bar{L}+L))=\exp(p \log(C))$ and this quantity converges to 1, meaning that in the limit, $DNC(\bar{L})=1$ which means that no progress has been made with the collapse of the layers until we reached the end of the feature map. This violates your UFM condition.
> > >
> > > Following this argument, in particular, I disagree with your claim that the threshold $\epsilon$ can be reached after a number of layers that is not *at least* a constant fraction of the total number of layers. This, howerver, violates the assumptions of your theorem.
> > >
> > > To put it even differently - if you scale up the number of layers, the work gets split roughly equally between the layers. This means that even if you look at ever increasing total number of layers within a network, if this is a vanishing fraction of the total depth, than their contribution converges to identity.
> > >
> > > I hope this clarified my concern. Please let me know if there still is a misunderstanding.

---

> > > > ### Author Response · Authors · 2025-11-20
> > > >
> > > > Thank you for bearing with us, we believe we now understand your concern and it is touching on a subtle but important assumption of our work that is not explicitly stated. In Theorem 6 we assume $\bar{L}$ represents, in the limit, a vanishing fraction of the total network. This is deemed sufficient for our full Hessian results due to two assumptions:
> > > >
> > > > 1. **Monotonicity of DNC**: DNC increases across each layer (lets say geometrically).
> > > > 2. **Asymptotic collapse**: DNC$(L+\bar{L})→0$ as the total depth $→∞$.
> > > >
> > > > As you point out, the second assumption appears to contradict the empirical behavior observed in He & Su, Fig. 5, where DNC$(L+\bar{L})$ seems to level off at a threshold $α>0$. This is the crux of your concern: if DNC does not continue to decrease with depth—and if $\bar{L}/(L+\bar{L})→0$—then the first $\bar{L}$ layers cannot achieve a fixed amount of collapse, and thus the assumptions of Theorem 6 are violated.
> > > >
> > > > This is a key conceptual point that we did not adequately address. Our position is that assumptions (1)–(2) are sufficient for the theoretical results we derive, and that (2) is in fact necessary at global minima—though we agree it is not always observed in finite-depth experiments. Let us outline why.
> > > >
> > > > **Why we expect assumption (2) at global minima**
> > > >
> > > > At a global minimizer of the loss among all functions, the network’s output must align with the labels. Since the labels satisfy DNC1, any function achieving this minimal loss must also satisfy full DNC1 at the output. By universal function theorems, sufficiently expressive networks can approximate such functions as depth grows. Consequently, in the limit of increasing depth, the globally optimal feature representation at the output must be fully collapsed; in particular, DNC$(L+\bar{L})→0$.
> > > >
> > > > Thus, the existence of a nonzero threshold $α$ for DNC$(L+\bar{L})$ is incompatible with achieving the global minimum in the expressiveness limit. This suggests that the plateau seen in Fig. 5 is not an intrinsic representational limitation but something specific to the optimization process used in the experiment.
> > > >
> > > > **Why He & Su may observe a plateau**
> > > >
> > > > If one inspects the work of He & Su, two design choices stand out that impede collapse at large depth:
> > > >
> > > > 1. **No residual connections**: Training deep networks without residual connections is known to increase optimization difficulty. Recent work also suggests that residual connections play an important role in the development of DNC [1,2].
> > > > 2. **Fixed training budget across depths**: Deeper networks typically require more training epochs to reach comparable effective optimization. Under a fixed-step training budget, deeper models are disproportionately undertrained. Thus the late-stage plateau in DNC may simply reflect optimization difficulties rather than true saturation of representational capacity.
> > > >
> > > > Unfortunately, He & Su do not provide loss curves, so we cannot tell whether deeper networks continue to reduce the loss toward the global minimum or whether they stall at suboptimal values. This missing information is essential to interpreting their DNC plateaus.
> > > >
> > > > **Supporting evidence from other works**
> > > >
> > > > Other empirical studies [3,4] observe that NC improves with expressiveness. However, these works vary width rather than depth, avoiding the optimization burden associated with deep, non-residual networks. Their results are consistent with the regime before the plateau in He & Su’s Fig. 5, which further supports the interpretation that the plateau arises from optimization difficulty rather than a fundamental obstruction to collapse.
> > > >
> > > > **Implications for our theory and revisions to the manuscript**
> > > >
> > > > We fully agree that the manuscript must be updated (especially Sections 1 and 3.4) to reflect the nuance your comment highlights. We will:
> > > >
> > > > * Explicitly state the assumptions underlying our full-Hessian results, including that $\bar{L}/(L+\bar{L})→0$.
> > > > * Acknowledge that He & Su report behavior inconsistent with assumption (2) in finite-depth settings.
> > > > * Explain why we believe their threshold α is likely an optimization artifact rather than an intrinsic limitation—and why assumption (2) is necessary for any network approaching the global minimum.
> > > > * Discuss how our conclusions would be affected if the He & Su plateau were intrinsic. In particular, noise would not vanish asymptotically, yet empirically it appears sufficiently small to explain the phenomena our theory aims to capture, at least in deeper layers and approximately for the full Hessian.
> > > >
> > > > A resolution would require a broad empirical study varying depth, width, residual connections, and training budgets—far beyond the scope of our work. But this is a valuable direction for future investigation, especially given the centrality of the UFM in recent analyses.
> > > >
> > > > We will incorporate these clarifications and welcome your thoughts on the revised explanation above before integrating it into the manuscript. Once again we are immensely grateful for your insights.

---

> > > > > ### Author Response · Authors · 2025-11-20
> > > > > **References**
> > > > >
> > > > > [1] Jianing Li, Vardan Papyan. Residual Alignment: Uncovering the Mechanisms of Residual Networks. Neural Information Processing Systems, 2023
> > > > >
> > > > > [2] Peter Súkeník, Christoph H. Lampert, Marco Mondelli. Neural Collapse is Globally Optimal in Deep Regularized ResNets and Transformers. ArXiv preprint, 2025.
> > > > >
> > > > > [3] Can Yaras, Peng Wang, Zhihui Zhu, Laura Balzano, Qing Qu. Neural Collapse with Normalized Features: A Geometric Analysis over the Riemannian Manifold. Neural Information Processing Systems, 2022.
> > > > >
> > > > > [4] Zhihui Zhu, Tianyu Ding, Jinxin Zhou, Xiao Li, Chong You, Jeremias Sulam, Qing Qu. A Geometric Analysis of Neural Collapse with Unconstrained Features. Neural Information Processing Systems, 2021.

---

> ### Comment · Reviewer_BQ76 · 2025-11-21
>
> Thank you for engaging in a scientifically productive discussion.
>
> I understand your argument and agree that two conditions are necessary for the UFM assumption in Theorem 6 to hold. However, I disagree that the second condition is to be found in global minima, if those are regularized. In particular, in your paper you consider regularized parameters. Then, there is a trade-off between how close we are to the exact collapse and how much does that cost. For instance, for pure ResNets [1] has shown that collapse is approached, but only if we only have 1 linear layer per residual block. In practice, usually there are two linear layers in each block. The authors show that in this case, if the regularization doesn't vanish as the number of layers goes to infinity, the neural collapse is not approached. In the case of pure MLPs, the situation is even worse, because the representation cost scales as $\lambda L.$ In your paper, you consider a mix of ResNet (feature extractor) and MLP (the head). I tried to analyze a 1D network of this structure and came to conclusion that the residual layers want to be very small in this case, suppressing their ability to produce strong collapse. Even without showing my computations, it is intuitive that similar tradeoff between the cost of the network and the closeness to collapse should persist, since it is the case for both pure MLPs and pure ResNets.
>
> I think the reason why He&Su observed this plateau can be partially explained by your observations, but probably also due to the use of weight decay.
>
> Now, maybe your assumption 2 would hold if you allowed very aggressive scaling of the regularization term to 0 as the number of layers approaches infinity. The scaling might also change depending whether one uses MSE loss or CE loss.
>
> [1] Peter Súkeník, Christoph H. Lampert, Marco Mondelli. Neural Collapse is Globally Optimal in Deep Regularized ResNets and Transformers. ArXiv preprint, 2025.

---

> > ### Author Response · Authors · 2025-11-22
> >
> > Thank you for the further clarification — we agree that we are now homing in on the core issue. In fact, our work does assume that the regularization parameter $\lambda$ tends to zero as $L \to \infty$, and at least at a linear rate. This assumption is already built into the statements leading up to Theorem 6 via Theorem 5:
> >
> > * Theorem 5 is stated “under the same assumptions as Theorem 1,” and Theorem 1 includes the condition that $\lambda$ satisfies Equation (17).
> > * For large $L$, Equation (17) is asymptotic to $\lambda \lesssim 1/L$.
> > * This decay condition is necessary to prevent the trivial zero solution from becoming optimal under excessive regularization, and it is also required for the Hessian perturbation term in Theorem 5 to remain small.
> >
> > Thus, when you write that assumption (2) might hold only if we “allow very aggressive scaling of the regularization term to 0,” this is the regime our framework is designed to operate in. Indeed, after reviewing Súkeník et al. (2025), Theorem 4.6, it is clear that our condition on $\lambda$ is stronger than theirs. As you point out, this stricter scaling is expected: our model includes an MLP head rather than being purely residual, and the MLP component introduces a regularization–collapse trade-off that is more delicate than in deep regularized ResNets alone.
> >
> > We fully agree that in other architectures the true rate at which $\lambda$ decays may need to be more aggressive than the simplified $\lambda \lesssim 1/L$ condition that is sufficient in our model. This is where we partly lean on empirical validation to justify such abstractions. This also connects to our earlier discussion of the low-rank bias results of Súkeník et al. and Garrod & Keating: the probability of DNC appears to depend in a highly nontrivial way on the decay rate of $\lambda$.
> >
> > If you feel that we have now gotten to the bottom of this conceptual issue, we would be very happy to write a precise outline of the modifications we will make to the paper in light of this exchange, so that you can evaluate them efficiently. We will also prepare a revised version shortly. However, we are equally happy to continue the discussion — we very much appreciate the depth and care of your feedback and are aware that you have already gone above the typical expectations of a reviewer.

---

> > > ### Comment · Reviewer_BQ76 · 2025-11-24
> > >
> > > Thank you for further discussing this point.
> > >
> > > I understand your argument and indeed the scaling $1/L$ could be sufficient for pure ResNets (although the issue is very delicate because even if the scaling guarantees convergence of NC metrics to 0, you would then need to make sure that $L, \bar L$ are scaled such, that $L/ \bar L$ would diverge to infinity at least as slow as the rate of convergence of the NC metrics to 0). However, for this combination of ResNets and MLPs, it is not clear to me that even such a scaling would help (I am not claiming it would not, I am just claiming that it is non-trivial). I would believe this with certainty only if one would disentangle the regularization strength on the MLP part from that of the ResNet part and scale these two regularization strengths differently.
> > >
> > > In any case, I will be happy to see your outlined changes and the revision of the manuscript. I think if you incorporate all our discussed points in the revision, the paper will meet the acceptance threshold in my eyes, even if I don't consider our discussion on the Theorem 6 and the progressive neural collapse fully resolved.

---

> > > > ### Author Response · Authors · 2025-11-24
> > > >
> > > > We agree that architecture-specific claims are difficult to justify at present, and that fully disentangling the regularization effects of the residual and MLP components—as you note—likely requires new analytical tools, in the spirit of Súkeník et al. (2025). We will make these limitations explicit in our discussion of the full Hessian results. This exchange has been extremely illuminating, and we are grateful for the depth of insight you have brought to these issues. It has also highlighted a promising direction for future work on the interplay between regularization strength, architectural design, and progressive collapse.
> > > >
> > > > Below, we outline the full set of revisions we will incorporate into the manuscript to ensure that all of your concerns are addressed.
> > > >
> > > > **General Revisions**
> > > >
> > > > * **Clarify the historical context:** In the introduction (paragraph 3), explicitly note that Papyan et al. (2020) suggested a connection between neural collapse and Hessian spectra.
> > > > * **Clarify prior work on Hessian analysis:** In Related Work, emphasize that earlier random-matrix–based Hessian studies leverage properties of the feature matrices in their Hessian explorations.
> > > > * **Situate our contribution accurately:** Make clear that we provide a new mathematical explanation for the Hessian spectral phenomena, although some empirical connections were previously observed.
> > > > * **Title revision:** Update the title to: “Unifying Low-Dimensional Spectra in Deep Learning”.
> > > > * **Activation smoothness clarification:** Replace the previous statement about smoothed approximations to ReLU with an explicit explanation that we set $\sigma'(0)=1$ as a minimal convention to recover the exact linear-network Hessians, noting that the network’s behavior is ultimately governed by the fact that it corresponds to a linear map in our regime.
> > > > * **Remove ambiguous language:** Remove the phrase “worst-case” from the explanation of Theorem 6.
> > > > * **Update DNC3 definition:** Correct the definition of DNC3 to align with Súkeník et al. (2024).
> > > > * **Clarify notation:** Explicitly define the notation for normalized vectors in the Background section.
> > > >
> > > > **Revisions Specific to Theorem 6 and the Full Hessian Results**
> > > >
> > > > * **Joint limit clarification:** State clearly that Theorem 6 considers a joint limit in both $L$ and $\bar{L}$.
> > > > * **Collapse assumption:** Emphasize in the explanation of Theorem 6 that our modelling assumes full collapse at the network output ($DNC→0$) in the depth limit.
> > > > * **Empirical alignment and discrepancies:** Note explicitly that while the experiments of Papyan et al. suggest collapse becomes sufficiently strong for our approximations to apply, the observations of He & Su (2022) and the theoretical results of Súkeník et al. (2025) demonstrate that optimization difficulty and regularization strength play a significant role and may prevent full collapse in other realistic regimes.
> > > > * **Role of regularization decay:** Highlight that Theorem 6 assumes regularization decays at least linearly with depth (as in Eq. 17), but that the necessary decay rate in practice is likely architecture-dependent.
> > > > * **Future work:** Explicitly identify the disentangling of architecture-specific regularization effects, and their implications for progressive collapse, as an important and promising avenue for future research.
> > > >
> > > >
> > > > If we have omitted anything or if you would like to suggest further adjustments, please let us know. We will provide both the revised manuscript and a detailed change log—ideally by tomorrow—so that you can verify that all points from our discussion have been accurately incorporated.

---

> > > > > ### Comment · Reviewer_BQ76 · 2025-11-26
> > > > >
> > > > > Thank you for your great efforts in this discussions and in improving the manuscript. I agree with the proposed changes and have increased my score.

---

### Official Review · Reviewer_SQKm · 2025-10-21

**Soundness:** 2
**Presentation:** 2
**Contribution:** 2
**Rating:** 2
**Confidence:** 3

**Summary:**

The authors study the spectral dynamics of the Unconstrained Features Model (UFM) for deep linear networks and deep networks with RELU activations, which assume that you directly optimize the representations input to the neural network, along with the intermediate weight matrices. The authors find that the UFM models exhibit the emergence of bulk outlier values in the gradient and Hessian spectra theoretically and empirically. The authors argue that these experiments and theory unify low-dimensional learning under the deep neural collapse framework, as these outlier eigenvalues correspond to different classes.

**Strengths:**

The goal of the paper is nice. There are lots of observations across deep learning that can be reduced to low dimensionality. The authors ask if we precisely describe this dimensionality using an established framework – deep neural collapse.

**Weaknesses:**

-	I find it overall very confusing how sections 3 and 4 related to “unifying low dimensional structure”. I guess they claim to show that for UFM the gradients / Hessian are low rank and have eigenvalues corresponding to different classes, but low dimensional structures in deep learning are much broader. Even in the context of the paper it is not clear to me what the reader should take away from each section.
-	It is very unclear that UFM captures the dynamics of real neural networks. The authors claim that they established DNC as “they establish DNC as a unifying mechanism shaping curvature, gradient alignment, and weight structure”, yet it is essentially impossible to establish a connection between their results and the UFM model to generalization on real datasets. The connection to the spectral dynamics of real neural networks on real datasets is not even established empirically in the paper. For example, I find it hard to expect that the Hessian and gradients in practice always have these outlier eigenvalues exactly corresponding to different classes.
-	I am especially concerned that weights being low rank is too general of a property to say anything useful regarding the purpose and structure of large singular directions of the weights. This is true even if you can predict eigenvalues/eigenvectors in the UFM model and their ranks. It is not just the singular values, but also, mainly, the singular vectors that are important.
-	The work claims to unify low-dimensional observations in deep learning without addressing the many settings and analyses that prove NNs learn low-dimensional structure and that this structure improves generalization. In particular, there is a long line of work on neural networks learning staircase functions multi-index models that is not connected to their observations in UFM or deep neural collapse (e.g. [1,2,3,4]).

[1] “The staircase property: How hierarchical structure can guide deep learning “. Abbe et al., 2021.

[2] “Neural Networks can Learn Representations with Gradient Descent”. Damian et al., 2021.

[3] “How two-layer neural networks learn, one (giant) step at a time”. Dandi et al., 2024.

[4] “Repetita Iuvant: Data Repetition Allows SGD to Learn High-Dimensional Multi-Index Functions”. Arnaboldi et al., 2024.

**Questions:**

None

---

> ### Author Response · Authors · 2025-11-16
>
> We thank the reviewer for the thoughtful and constructive feedback. We have carefully addressed each concern in detail below. If our clarifications and revisions resolve the issues you raised, we kindly ask that you consider updating your evaluation accordingly. If any part of our response does not fully address your concerns, we would be happy to continue the discussion and provide further explanation or additional experiments as needed.
>
> **W1: I find it overall very confusing how sections 3 and 4 related to “unifying low dimensional structure”... low dimensional structures in deep learning are much broader… it is not clear to me what the reader should take away from each section.**
>
> Thank you for pointing this out. We agree that the term “low-dimensional structure” was too broad and could be misleading. To avoid overclaiming, we will revise the title to “Unifying Low-Dimensional Spectra in Deep Learning”, which more accurately reflects our scope—namely, the spectra of deep-learning matrices, gradients, and DNC-related quantities—rather than all notions of low dimensionality in deep learning.
>
> We also appreciate that the takeaway from each section may not have been sufficiently clear. We apologize for this and will revise the manuscript to include a more explicit summary. For clarity, we outline the intended contributions of each section below:
>
> * **Section 3.1-3.3**: We show that DNC causes the spectral structure reported in prior work (as detailed in the introduction), in the sense that they all arise directly from the geometry of the class means. Thus DNC provides a single unifying explanation.
> * **Section 3.4**: We demonstrate that the full Hessian inherits this low-dimensional structure from the layer-wise Hessians, explaining why DNC leads to similar spectral behavior at the full-network level.
> * **Section 4**: We demonstrate that the same results extend beyond linear layers to ReLU layers.
>
> Taken together, these sections support our central claim: DNC provides a unified explanation for the low-dimensional spectral phenomena observed across deep learning.
>
> **W4: The work claims to unify low-dimensional observations in deep learning without addressing the many settings and analyses that prove NNs learn low-dimensional structure and that this structure improves generalization.**
>
> Thank you for raising this concern. As noted above, we will revise the title to make clear that our focus is specifically on low-dimensional spectral phenomena, as described in the introduction and throughout the main text, rather than on the full range of low-dimensional behaviors studied in deep learning. We hope that, with this clarification and the updated title, the work can be evaluated with respect to the more focused goals stated in the introduction, which would mitigate the concern you raise here.

---

> ### Author Response · Authors · 2025-11-16
>
> **W2A: It is very unclear that UFM captures the dynamics of real neural networks.**
>
> Thank you for raising this central question. We justify the correspondence between the UFM and real networks in two ways:
>
> **Empirical correspondence:** A substantial body of prior work has shown that UFMs accurately capture the terminal-phase behavior of overparameterized deep networks (18+ works, as summarized in the first paragraph of the Related Work section). In Appendix F, we additionally evaluate our own predictions on MNIST and CIFAR-10. These experiments show strong agreement with our theoretical spectra, eigenvectors, and gradient–Hessian alignment, providing direct evidence that the UFM reproduces the observed behavior of real networks at convergence.
>
> **Modelling assumptions:** The UFM assumes that the feature map h(x) is expressive enough to map the dataset to nearly arbitrary locations in feature space. This assumption is consistent with the overparameterized regime in which neural collapse and low-dimensional spectral phenomena are observed in practice. Importantly, we require correspondence only at convergence, not throughout the entire optimization trajectory. While optimizing the features h(x) in real networks is certainly more challenging than optimizing them directly in the UFM, the empirical observation that networks rarely become trapped in poor local minima—and instead reach solutions that are close to globally optimal—supports the validity of this assumption. This is, of course, tied to one of the fundamental open questions in deep learning, but the consistency of empirical results across architectures and datasets provides practical justification for using the UFM as an analysis tool.
>
> We will add a section to the appendix to further motivate the UFM and clarify these assumptions. If you still have concerns about the model, we would be grateful if you could elaborate on what additional evidence you believe would be most useful for us to provide.
>
> **W2B: The authors claim that they established DNC as “they establish DNC as a unifying mechanism shaping curvature, gradient alignment, and weight structure”, yet it is essentially impossible to establish a connection between their results and the UFM model to generalization on real datasets.**
>
> Thank you for raising this point. We agree that the UFM—being intentionally data-agnostic—cannot directly address generalization. This is indeed a deliberate tradeoff. Our goal in this work is to explain optimization-induced structures—curvature, gradient alignment, and weight structure—all of which arise prior to evaluating generalization. As discussed above, the UFM is uniquely well suited for isolating and analyzing these phenomena in a clean and tractable setting.
>
> While these low-dimensional structures have been linked to generalization in prior work (as summarized in the Related Work section), our contribution is different in scope: we establish DNC as a single mechanism that generates these spectral structures within the UFM. Understanding the origin of these phenomena is a necessary step before one can rigorously analyze their downstream implications for generalization. We view such analysis as outside the scope of the present paper but important future work now that we have unified such empirical observations.
>
> **W2C: The connection to the spectral dynamics of real neural networks on real datasets is not even established empirically in the paper… I find it hard to expect that the Hessian and gradients in practice always have these outlier eigenvalues exactly corresponding to different classes.**
>
> We would like to clarify that we do evaluate the spectral dynamics of real networks trained on real datasets in Appendix F. These experiments show very strong agreement with our theoretical predictions, Can the reviewer please clarify if they meant these experiments are insufficient, since we are happy to produce further metrics.
>
> We placed these empirical validations in the appendix for space reasons, but we are happy to reorganize the paper If the reviewer believes this would be useful. Nonetheless, we hope that these experiments are considered, despite the fact they are in an appendix, given that they address the exact concern you are raising.
>
> **W3: I am concerned that weights being low rank is too general ... the singular vectors are important.**
>
> We completely agree and emphasize that Theorem 4 gives closed-form expressions for both the singular values and the singular vectors: they are the class-feature means of the adjacent layers. While we did not test this prediction directly due to our focus on Hessian structure, we will add an experiment confirming this in real networks. In advance of this, we note that several empirical results in Appendix F (e.g., the Hessian eigenvectors in Figures 11 and 14) implicitly depend on the predicted singular-vector structure of the weights, providing indirect validation of our theoretical claim.

---

> > ### Author Response · Authors · 2025-11-20
> > **Weight Singular Vectors Experiment**
> >
> > As an update on the singular vector expression and conclusion: we tested whether $\hat{\mu}^{(l+1)}_c$, for $c=1,...,K$, are left singular vectors of $W_l$ by looking at the cosine similarity between $\hat{\mu}^{(l+1)}_c$ and $W_l W_l^T \hat{\mu}^{(l+1)}_c$ for the same layer of the ResNet used for the experiments in Appendix F on MNIST after training was completed. We justify why this metric is suitable in the paper, and it takes value 1 if they are eigenvectors. We also perform a similar test to ascertain the extent to which $\hat{\mu}^{(l)}_c$, for $c=1,...,K$, are right singular vectors of $W_l$.
> >
> > Across the 20 predicted left and right singular vectors at that layer, the lowest cosine similarity was 0.99971, so our theoretical prediction aligns strongly with experiment. We are happy to update the paper to include this result along with the exact evolution trajectory throughout training.

---

> ### Author Response · Authors · 2025-11-25
> **Summary of Revisions in Response to Reviewer Comments**
>
> We thank the reviewer for their thoughtful and constructive feedback. Their comments helped us clarify the presentation, strengthen the connection between our theoretical model and empirical evidence, and better highlight the paper’s core contributions. Below, we summarize the key revisions made in response to the reviewer’s suggestions.
>
> **Title clarity**
>
> To address the concern that the original title was too vague, we have revised it to “Unifying Low-Dimensional Spectra in Deep Learning.” This title more accurately reflects the focused goals of our theoretical analysis.
>
> **Clearer section takeaways**
>
> We have substantially revised the Contributions section to more explicitly address each main result.
>
> Additionally, we have added a detailed introductory paragraph to Section 3 that summarizes the key takeaways of each major part of the paper, improving readability and overall guidance for the reader.
>
> **Justification for why UFM captures real network dynamics**
>
> We have added a dedicated appendix (Appendix A) that consolidates:
>
> * the extensive empirical evidence supporting UFM models,
> * new theoretical evidence from Súkeník et al. [1], and
> * clear justification for the modelling assumptions used.
>
> This should make the motivation and validity of UFM as a model for real networks considerably more transparent.
>
> **Clearer connection between spectral dynamics of real networks and the model**
>
> To strengthen the link between our theoretical model and empirical behavior, we have moved parts of the MNIST and CIFAR-10 full-network experiments from the appendix into the main text. Their strong alignment with our model’s predictions is now more visible to the reader.
>
> **Highlighting singular vector and eigenvector results**
>
> We have revised portions of both the introduction and conclusion to more prominently emphasize our characterization of singular vectors and eigenvectors across all matrices studied in the paper.
>
> We would especially like to thank the reviewer for noting that these singular vector and eigenvector results were previously under-emphasized. They are, in fact, unique contributions of our work—unobserved even empirically in prior literature—and we have revised the manuscript to clearly highlight their significance.
>
> We welcome any additional feedback you may have on these revisions, as well as your broader thoughts on how effectively we have addressed your concerns. We are also preparing additional experiments, including full trajectories of the weights singular vector metrics (beyond the final-time values included in our earlier response), which will be added in the next update.
>
> **References**
>
> [1] Peter Súkeník, Christoph H. Lampert, Marco Mondelli. Neural Collapse is Globally Optimal in Deep Regularized ResNets and Transformers. arXiv preprint, 2025.

---

> ### Author Response · Authors · 2025-12-01
>
> Dear Area Chair,
>
> We would like to provide an update on new experiments related to this reviewer's concerns.
>
> The reviewer expressed skepticism about our weight–singular-vector results. In the revised submission, we have added plots illustrating the alignment between the predicted singular vectors and the true singular vectors of the weight matrix for an overparameterized network trained on MNIST (see Appendix G: Further Numerical Experiments, Figure 16). These results show excellent agreement with our theoretical predictions, demonstrating that our analysis extends to real networks. We hope this additional evidence will be taken into account.

---

### Official Review · Reviewer_9iP3 · 2025-10-28

**Soundness:** 3
**Presentation:** 3
**Contribution:** 2
**Rating:** 4
**Confidence:** 4

**Summary:**

This paper studied the low-dimensional structures of deep unconstrained feature models (UFMs), from weights, gradients and hessian matrices. The papers demonstrated the driving factors and solutions of deep neural collapse (DNC) and that the full Hessian inherits the layer-wise Hessian structures.

**Strengths:**

This study provided an unified perspective towards the dimensional structures in weights, gradients and hessian matrices, leading to explanations towards the DNC phenomenon.

**Weaknesses:**

1. The contributions of this paper is limited in significance, and the conclusions on DNC is rather unclear; It could be further emphasized in the concluding remarks.
2. The optimization perspective is studied less in this paper, for which the low dimensional structure interplay with learning dynamics is also a critical factor to the DNC phenomenon.
3. The scope of the numerical experiments are limited in the main paper.

**Questions:**

1. As an further demonstration of the DNC phenomenon, can the authors further present the low-dimensional structure formation in learning, and towards solving the DNC problem?
2. The story-telling could be improved on the full Hessian spectrum side, e.g. how may this structure derivation assist future studies, and what research questions may benefit from this conclusion.

---

> ### Author Response · Authors · 2025-11-16
>
> We thank the reviewer for their thoughtful evaluation and constructive comments. In the responses below, we address each concern in detail. We hope these clarifications help convey the strength and scope of the work, and we respectfully ask the reviewer to consider raising their score if they find the responses satisfactory.
>
> **W1: The contributions of this paper is limited in significance, and the conclusions on DNC is rather unclear; It could be further emphasized in the concluding remarks.**
>
> We appreciate the opportunity to clarify the significance of our findings. A substantial body of work has sought to understand the spectral properties of deep learning—Hessian spectra, gradient covariance, Fisher information matrices, and weight spectra—because these structures are tied to generalization, optimization efficiency, and robustness (as detailed in the Related Work section). Yet, despite this effort, theoretical analyses have only captured isolated aspects of the empirically observed phenomenology.
>
> Our work makes a step forward by providing the first analytic explanation that simultaneously reproduces all low-dimensional spectral phenomena described by Papyan and others. Importantly, we do not simply match eigenvalue counts: we characterize eigenvectors and singular vectors explicitly in terms of class feature means. Prior work has not provided such a complete and interpretable description, even empirically. We believe this is a significant step given the extent of previous efforts to make such characterizations. We will revise the introduction to include these details on the significance of our results.
>
> We are slightly unsure what is meant by “the conclusions on DNC are rather unclear.” What we demonstrate in the paper is that DNC definitively causes the low-dimensional spectra and gradient observations reported empirically (as detailed in the introduction). There is no other work in the literature that definitively demonstrates and formalizes this theory. Our conclusion is admittedly brief, and we will update it to make sure this contribution is clearer. If we have misinterpreted your concern, could you please clarify?
>
> **Q2: The story-telling could be improved on the full Hessian spectrum side, e.g. how may this structure derivation assist future studies, and what research questions may benefit from this conclusion.**
>
> Thank you for this helpful suggestion. We are happy to clarify the significance of our Hessian results. By showing that the full Hessian inherits its structure directly from DNC, our analysis provides a principled explanation for several empirical findings, and it offers a foundation for more systematic design choices in future work. For instance, our work makes clear that the deep UFM enables controlled exploration of how hyperparameters influence local flatness, which has been linked to generalization. Additionally, recent work argues that flatness—rather than NC—is the key driver of generalization [1]; our results demonstrate that these two notions are not independent under standard training, and therefore our analytic expressions can guide new architectures or regularization schemes that target flatness by modifying the NC geometry.
>
> Beyond concrete applications, a clearer understanding of Hessian structure also contributes to core theoretical questions, such as why overparameterized models avoid overfitting. The low-dimensional structures we characterize arise robustly across many design choices, which suggests they encode fundamental aspects of deep learning and explains why they have been so heavily studied in the literature. Providing an explanation for the full spiked Hessian spectrum—beyond the partial accounts in prior work—represents a meaningful step forward in such theoretical understanding.
>
> We will update our introduction to include this explanation.

---

> ### Author Response · Authors · 2025-11-16
>
> **W2: The optimization perspective is studied less in this paper, for which the low dimensional structure interplay with learning dynamics is also a critical factor to the DNC phenomenon.**
>
> Thank you for raising this point. The goal of our work is to explain mathematically why the empirical observations of low-dimensional structure, as described in the introduction, arise. The conditions under which these observations appear are most clearly explained in the original NC paper by Papyan et al., which identifies overparameterization and the terminal phase of training as the key regimes. These conditions inform our perspective, since the unconstrained feature assumption underpinning our model corresponds to the overparameterization limit, and the consideration of critical points of the loss corresponds to the terminal phase of training. In particular, we provide a complete characterization of the UFM in these limits, which required substantial theoretical development.
>
> We agree that studying learning dynamics can further illuminate how DNC emerges. Other researchers have considered this question [2,3], for which different modeling approaches—such as mean-field theory—are more appropriate. Because such analyses rely on different tools and our theoretical contribution is already quite extensive, we view dynamic analyses as beyond the scope of this project and as promising future work.
>
> **W3: The scope of the numerical experiments are limited in the main paper.**
>
> Thank you for raising this point. In the main paper we prioritized presenting the full theoretical framework, and therefore placed several supporting experiments—including those validating our predictions in real networks on standard datasets—in Appendix F. We recognize that reviewers are not expected to read the full appendix, but hope these experiments will be considered when assessing the empirical component. If the reviewer feels that some of these results should be moved into the main text, we are happy to do so. If, even including Appendix F, the experimental scope appears insufficient, we would be glad to provide additional evaluations and would welcome suggestions on what would be most useful.
>
> **Q1: Can the authors further present the low-dimensional structure formation in learning, and towards solving the DNC problem?**
>
> Thank you for the suggestion. We are happy to provide additional experiments, though we are not entirely sure what is meant by “presenting the low-dimensional structure formation in learning.” Our interpretation is that this refers to visualizing or quantifying how the predicted structure emerges during training. We have already included such analyses—for example, Figures 1 and 4 in the main text, and Figures 11 and 14 in Appendix F for deep networks on MNIST and CIFAR-10—but we would be glad to add further metrics if the reviewer has something specific in mind. Alternatively, if we have misunderstood your request, could you please clarify?
>
> We are also unsure what is intended by “towards solving the DNC problem.” If this refers to elaborating on a theoretical explanation for the emergence of DNC or clarifying its implications, we would be glad to do so.

---

> > ### Author Response · Authors · 2025-11-16
> > **References**
> >
> > [1] Ting Han, Linara Adilova, Henning Petzka, Jens Kleesiek and Michael Kamp. Flatness is Necessary, Neural Collapse is Not: Rethinking Generalization via Grokking. ArXiv preprint, 2025.
> >
> > [2] Diyuan Wu and Marco Mondelli. Neural Collapse Beyond the Unconstrained Features Model: Landscape, Dynamics, and Generalization in the Mean-Field Regime. ArXiv preprint, 2025.
> >
> > [3] Wanli Hong and Shuyang Ling. Beyond Unconstrained Features: Neural Collapse for Shallow Neural Networks with General Data. ArXiv preprint, 2024.

---

> > ### Comment · Reviewer_9iP3 · 2025-11-23
> >
> > The reviewer's effort and patience in addressing my concerns are greatly appreciated.
> >
> > **W1/W2**: The author's comments have largely resolved my confusion; the strength and the scope of the paper are now much clearer to me.
> >
> > **W3**: Regarding the empirical studies, some of my personal suggestions follow, which are more of a discussion of presentation methodologies, which will not impact my paper rating: 1. Figures 11 and 14 from the Appendix. F can be merged into a single plot and be supplied in the main paper, as an important demonstration and comparison to Figures 1 and 4, the training of deep UFMs (given the extended page limit). 2. Consider using additional log-rank plots for Figure 5 and Figure 12 to present the layer-wise Hessian spectra, and demonstrate the outlier eigenvalues w.r.t. the heavy-tail eigenvalues.  3. The mean and standard deviation of outlier eigenvalue distributions can be plotted against the growing epoch counts, demonstrating the trend in training dynamics. 4. Statistical correlations between predicted eigenvector alignment and gradient alignment can be reported and discussed for single models and against different models, potentially as additional numerical evidence to the paper's claim. 5. While further experiments can always be extended to CIFAR-100 and Vision Transformers (ViT), they can be considered as non-essential to this study.
> >
> > **Q1**: Potential metric that comes to my mind include the Spectral (Shannon) entropy [1,2] and the Effective rank [3,4] of matrices (gradients and Hessian). It would be nice to see numerical evidence on low-dimensional structure formation and include these metrics in the proposed theoretical landscape. Nevertheless, I may have to rely on the authors to clarify whether they are good fits for the claim or not.
> >
> > **Q2**: The discussions on the improved understandings and potential implications of the Hessian structure brought by this study are intriguing and could be considered as meaningful contributions to the field.
> >
> > I would like to personally appraise the author's professionalism, and I would like to consider raising my score during the rebuttal stage.
> >
> > [1] Anand, Kartik, Ginestra Bianconi, and Simone Severini. "Shannon and von Neumann entropy of random networks with heterogeneous expected degree." Physical Review E—Statistical, Nonlinear, and Soft Matter Physics 83.3 (2011): 036109.
> >
> > [2] Jha, Nandan Kumar, and Brandon Reagen. "Spectral Scaling Laws in Language Models: How Effectively Do Feed-Forward Networks Use Their Latent Space?." arXiv preprint arXiv:2510.00537 (2025).
> >
> > [3] Roy, Olivier, and Martin Vetterli. "The effective rank: A measure of effective dimensionality." 2007 15th European signal processing conference. IEEE, 2007.
> >
> > [4] Yang, Jiang, Yuxiang Zhao, and Quanhui Zhu. "Effective Rank and the Staircase Phenomenon: New Insights into Neural Network Training Dynamics." arXiv preprint arXiv:2412.05144 (2024).

---

> ### Author Response · Authors · 2025-11-25
> **Summary of Revisions in Response to Reviewer Comments**
>
> Thank you very much for your thoughtful comments and for the positive assessment of our work. We have uploaded a revised version of the manuscript that incorporates your suggestions and addresses the points you raised. For your convenience, we summarize the updates specific to your evaluation below:
>
> **Clarified significance:**
>
> We have added a dedicated Significance section to more clearly articulate the contributions and context of our results, including the broader narrative regarding the Hessian structure, as discussed in our earlier response.
>
> **Strengthened concluding remarks:**
>
> The concluding section has been expanded and rewritten to better emphasize the implications and takeaways of our theoretical findings.
>
> **Revised numerical experiment presentation:**
>
> Following your suggestion, we have merged the original Figures 11 and 14 (formerly in Appendix F) into a single plot and moved it into the main text, facilitating more direct comparison with the corresponding modeling results.
>
> **Additional empirical suggestions:** We greatly appreciate your ideas regarding further analyses, and we agree that these directions are highly valuable. We are currently working on incorporating several of these suggestions and aim to provide the corresponding plots in a revision before the rebuttal deadline.
>
> Due to time constraints, we will not be able to run the additional experiments on CIFAR-100 or Vision Transformers before the deadline, but we view these as valuable directions and will consider them for future work or for a camera-ready version.
>
> Thank you again for the constructive feedback and for considering raising your score. We sincerely appreciate your engagement with our work.

---

> ### Author Response · Authors · 2025-12-01
>
> We provide here a follow-up on the experiments that the reviewer suggested might be useful, while noting that they stated these would not impact their score.
>
> **2. Consider using additional log-rank plots for Figure 5 and Figure 12 to present the layer-wise Hessian spectra, and demonstrate the outlier eigenvalues w.r.t. the heavy-tail eigenvalues.**
>
> We have decided against including such experiments. While these plots are indeed used by Papyan [1], whose work helps motivate ours, the layer-wise Hessians in our setting exhibit significantly less noise than the full Hessian. Consequently, the bulk near zero is much smaller, and logarithmic scaling is unnecessary to observe clear separations among the eigenvalues. In our current experiments, the standard linear scale already makes these separations apparent, and effectively communicates that the empirical behavior aligns with our theoretical predictions.
>
> **3. The mean and standard deviation of outlier eigenvalue distributions can be plotted against the growing epoch counts, demonstrating the trend in training dynamics.**
>
> We have now included such plots in Figure 18 of Appendix G. These results again agree with our theoretical predictions, and we provide an expanded discussion of the trends in the same appendix.
>
> **4. Statistical correlations between predicted eigenvector alignment and gradient alignment can be reported and discussed for single models and against different models, potentially as additional numerical evidence to the paper's claim.**
>
> Our work does not claim that eigenvector alignment should necessarily correlate with gradient alignment. Instead, our focus is on DNC, and our claim regarding its importance is supported by explicit constructions of the alignment variables and eigenvectors. With this framing, we consider our existing empirical results sufficient to demonstrate how the occurrence of DNC relates to these other phenomena.
>
> **Potential metric that comes to my mind include the Spectral (Shannon) entropy [1,2] and the Effective rank [3,4] of matrices.**
>
> We have added plots of the effective rank of both the weights and the layer-wise Hessian matrices in Appendix G (Figure 17). These results again show precise agreement with our theory. Because the effective rank is simply the exponential of the spectral entropy, we chose not to include both metrics.
>
> If the area chair believes that additional experiments would be valuable, or if they disagree with our justifications above, we are very happy to conduct further experiments prior to the camera-ready deadline.
>
> [1] Vardan Papyan. Traces of class/cross-class structure pervade deep learning spectra. Journal of Machine Learning Research, 2020

---

### Official Review · Reviewer_LCvk · 2025-10-30

**Soundness:** 2
**Presentation:** 3
**Contribution:** 3
**Rating:** 4
**Confidence:** 4

**Summary:**

The paper derives results about the structure of the Hessian associated with a variant of deep learning classification networks called UFM. The main difference between UFM and standard deep nets is the fact that the input are themselves assumed to be features computed via such a highly flexible function that one can treat them as optimization parameters.  With this approximation, analytical derivations provide support for many empirical or semi-empirical observations reported in earlier papers.

One intriguing property of this approach, which I would like to see commented, is the fact that making the inputs subject to optimization allows the network to simply make them directly express what we are trying to predict. So in that sense, the optimal solution for an UFM is trivial. Although we can still compute gradients and Hessians, they only describe what's happening in the layers that follow a collapsed representation. These layers, essentially, do nothing.

However it would be too fast to say that this observation totally undermines the paper. The empirical fact remains that the observed Hessian in actual deep networks display the same structure as the Hessians of these rather degenerate UFMs. That in itself demands an explanation!

**Strengths:**

+ analytical derivations on UFMs match empirical observations on DNNs.

**Weaknesses:**

- needs a more intentional discussion of the degenerate nature of UFMs

**Questions:**

- please discuss the degenerate solutions of the UFM formulation.
- please discuss the possible causes for the match between the empirical DNN Hessians and the UFM ones.

---

> ### Author Response · Authors · 2025-11-16
>
> We thank the reviewer for the thoughtful and constructive feedback. Your comments make it clear that we should better motivate the UFM formulation and clarify its assumptions, particularly regarding its degeneracy and its relation to empirical deep networks. We have addressed these points below and will add the corresponding discussion to an Appendix. We would also be grateful if, after reviewing these clarifications, you might consider raising your score.
>
> If anything in our response remains unclear, or if our answers do not sufficiently address your questions, please let us know. Additionally, we note that the weakness you describe concerns the model detailed in the background section. If you have any feedback on the results in Sections 3 and 4, we would be grateful for the opportunity to address that as well.
>
> **needs a more intentional discussion of the degenerate nature of UFMs / please discuss the degenerate solutions of the UFM formulation**
>
> We agree that the UFM admits an apparent degeneracy: because the feature map is assumed to be arbitrarily expressive, the model can map each input directly to its corresponding label. While such degeneracy may initially seem problematic, it is in fact designed to mirror the overparameterized regime of real deep networks. In this regime, the network has sufficient capacity to fit the data exactly, and additional layers become redundant in the sense that they “do nothing’’ beyond achieving perfect interpolation—exactly as you point out in your review. Thus, this property is not a flaw but an intentional feature of the model, intended to isolate the overparameterized setting so that its geometric consequences can be analyzed cleanly.
>
> Importantly, this degeneracy does not trivialize the model. Recovering the label matrix is far weaker than exhibiting deep neural collapse (DNC). Many non-DNC solutions perfectly interpolate the labels, yet they do not arise as optimization endpoints in the linear UFM, and they appear only in narrow hyperparameter regimes in the nonlinear case. Indeed, proving that DNC is the global optimum in the linear model [1] requires nontrivial matrix-completion arguments, demonstrating that this structure is far from automatic.
>
> The UFM’s degeneracy should therefore be understood in the same spirit as many implicit-bias analyses—such as linear networks trained on separable data under exponentially tailed losses or matrix-factorization models—where large expressivity creates a family of trivial interpolators, yet optimization systematically selects highly structured members of this family. The UFM provides a particularly clean and analyzable setting in which to study the selection mechanism of overparameterization and its consequences for curvature and spectral structure.
>
> **The observed Hessian in actual deep networks display the same structure as the Hessians of these rather degenerate UFMs … please discuss the possible causes for the match between the empirical DNN Hessians and the UFM ones.**
>
> We agree that it is indeed surprising that the UFM reproduces the spectral structure of Hessians observed in genuinely overparameterized deep networks. The UFM is designed precisely to emulate the behavior of the later layers of such networks, abstracting away the specifics of the optimization trajectory while retaining the essential geometric and spectral properties at convergence. The key assumption enabling this correspondence is, once again, the unconstrained feature assumption discussed above. Your question therefore ultimately boils down to why this assumption appears empirically valid.
>
> The first point to emphasize is that the network must overparameterize the dataset, which depends strongly on the intrinsic complexity of the data. In many deep-learning contexts—such as the classification settings studied in our work and in related experimental papers [2,3]—this appears to be a reasonable assumption. However, we would expect it to break down in more constrained regimes or for sufficiently complex datasets.
>
> In our view, the most interesting challenge to this assumption is the fact that discovering good features is genuinely difficult for real networks, given the complexity of the loss landscape, whereas it is much simpler in the UFM. This concern is partially mitigated by the fact that the model is meant only to reflect the network at critical points of the loss; we do not require the optimization trajectories themselves to match.
>
> That said, understanding why real networks reliably reach solutions that are close to globally optimal remains one of the foundational open questions in deep learning theory, and lies beyond the scope of our work. While there exist proposals for why deep-learning loss surfaces may be navigable in practice, most current explanations remain debated.
>
> If you have a specific reason to believe that the UFM and real networks should not show corresponding results, we would be happy to address it.

---

> > ### Author Response · Authors · 2025-11-16
> > **References**
> >
> > [1] Hien Dang,Tan Minh Nguyen,Tho Tran,Hung The Tran, and Nhat Ho. Neural collapse in deep linear networks: From balanced to imbalanced data. In International Conference on Machine Learning, 2023.
> >
> > [2] Vardan Papyan. Traces of class/cross-class structure pervade deep learning spectra. In Journal of Machine Learning Research, 2020.
> >
> > [3] Vardan Papyan, Xuemei Han, and David L. Donoho. Prevalence of neural collapse during the terminal phase of deep learning training. In Proceedings of the National Academy of Sciences of the United States of America, 2020.

---

> > ### Comment · Reviewer_LCvk · 2025-11-19
> > **The burden of the proof**
> >
> > "If you have a specific reason to believe that the UFM and real networks should not show corresponding results, we would be happy to address it."  -- I believe the burden of the proof is on you, that is, explaining why UFMs should produce good predictions of what is seen in real networks. Alternatively, recognize that this is an empirical observation without satisfying theoretical basis (which is the most interesting bit in the paper in my opinion.)

---

> ### Author Response · Authors · 2025-11-21
>
> We apologize for the misunderstanding. We fully agree that the justification of the UFM is our responsibility; our intention in asking whether you had a specific counterexample in mind was simply to ensure that we were addressing your concern as specifically as possible, rather than giving a general response that may have missed a precise objection you had.  We simply wanted to make sure we had understood your perspective as clearly as possible.
>
> Below we provide an expanded and more structured justification of the UFM as a modeling tool. We would be grateful for any further guidance on how to strengthen this part of the paper.
>
> **(A) The UFM reproduces previously documented phenomena**
>
> A first criterion for the usefulness of any abstract model is its ability to capture known empirical behavior. The UFM has been the central object of a large body of work, and has repeatedly matched empirical observations across settings—including neural collapse [1-5], its variants for different losses [6-8], DNC [9-12], and geometric structures in high-class-count regimes [13].
>
> Our paper continues this line of evidence by showing that UFM predictions agree with the spectral and gradient-related phenomena empirically observed in modern deep networks, as summarized in the introduction. This empirical validation is the primary reason the model has been considered meaningful in previous work, and forms the basis of its use in our setting as well.
>
> **(B) The UFM makes new predictions that are subsequently validated**
>
> A stronger justification for a model is its ability to generate novel predictions that turn out to be correct. The literature provides many such case studies for UFMs (e.g. predictions of NC behavior for imbalanced classes [14-17], new low-rank solutions [18,19], extensions to regression [20] and graph neural networks [21], and analogues in language models [22,23]).
>
> Our work adds another instance of this: the eigenvector structure of DNN Hessians and other deep learning matrices detailed in our paper had not been predicted previously. The UFM provides explicit analytic formulas, and our experiments demonstrate that real networks follow these predictions closely.
>
> We agree with the reviewer that these two points—agreement with known phenomena and predictive power for new ones—carry more weight than any direct argument about the plausibility of assumptions in isolation. Without empirical alignment, those assumptions would be irrelevant; with alignment, they become scientifically meaningful.
>
> **(C) Clarifying the theoretical basis of the UFM**
>
> In our previous response we summarized the intuition behind the UFM assumptions. For completeness, we provide a more formal justification here.
>
> The UFM corresponds to the hypothesis that, in the overparameterized limit, the learned feature representations at global minima coincide with choosing the optimal feature representations directly. Under extremely mild assumptions on the data (e.g. no data point appears twice with different labels), universal approximation results ensure that in the overparameterization limit the feature map can express the function that places the feature vectors at an optimal location. Thus, at global minima and in the high-parameter regime, the optimal features and the features of the actual network coincide. This is the mathematical foundation underlying the model.
>
> Of course, this foundation idealizes many aspects of real training. As we noted earlier, deep networks may not reach global minima exactly, and the overparameterized limit is not literally attained in practice. These are precisely the reasons why empirical validations—points (A) and (B)—are essential.
>
> Importantly, the UFM is not unusual in relying on idealizations. Many influential deep learning models operate under assumptions known not to hold exactly: NTK theory [24] (infinite-width limit), spin-glass analogies [25] (Hessian spectra deviate sharply from semicircle laws), linearized models [26] (which omit feature learning), etc. Yet these models have been highly valuable precisely because, despite their idealizations, they align with key empirical behaviors of real networks and have thereby produced meaningful insights for learning theory. We view the UFM in the same vein: its utility is grounded not in literal fidelity of assumptions, but in its repeated empirical alignment with phenomena observed in practical overparameterized networks.
>
> We will incorporate these clarifications—including a dedicated appendix motivating the UFM—into the revised manuscript. We appreciate the reviewer’s push for greater clarity, and we would be grateful for any additional suggestions on how to strengthen the exposition.

---

> > ### Author Response · Authors · 2025-11-21
> > **References**
> >
> > [1]  Dustin G Mixon, Hans Parshall, and Jianzong Pi. Neural collapse with unconstrained features. Sampling Theory, Signal Processing, and Data Analysis, 2022.
> >
> > [2] Jianfeng Lu and Stefan Steinerberger. Neural collapse under cross-entropy loss. Applied and Computational Harmonic Analysis, 2022.
> >
> > [3] Zhihui Zhu, Tianyu Ding, Jinxin Zhou, Xiao Li, Chong You, Jeremias Sulam, and Qing Qu. A geometric analysis of neural collapse with unconstrained features. In Conference on Neural Information Processing Systems, 2021
> >
> > [4] Wenlong Ji, Yiping Lu, Yiliang Zhang, Zhun Deng, and Weijie J Su. An unconstrained layer-peeled perspective on neural collapse. In ICLR, 2022.
> >
> > [5] Peng Wang, Huikang Liu, Can Yaras, Laura Balzano, and Qing Qu. Linear convergence analysis of neural collapse with unconstrained features. In NeurIPS Workshop, 2022.
> >
> > [6] Jinxin Zhou, Xiao Li, Tianyu Ding, Chong You, Qing Qu, and Zhihui Zhu. On the optimization landscape of neural collapse under MSE loss: Global optimality with unconstrained features. In ICML, 2022.
> >
> > [7] X. Y. Han, Vardan Papyan, and David L Donoho. Neural collapse under mse loss: Proximity to and dynamics on the central path. In ICLR, 2022.
> >
> > [8]  Jinxin Zhou, Chong You, Xiao Li, Kangning Liu, Sheng Liu, Qing Qu, and Zhihui Zhu. Are all losses created equal: A neural collapse perspective. In Conference on Neural Information Processing Systems, 2022.
> >
> > [9] Peter Súkeník, Marco Mondelli, and Christoph H. Lampert. Deep neural collapse is provably optimal for the deep unconstrained features model. In Conference on Neural Information Processing Systems, 2023.
> >
> > [10] Hien Dang, Tan Nguyen, Tho Tran, Hung Tran, and Nhat Ho. Neural collapse in deep linear network: From balanced to imbalanced data. In ICML, 2023.
> >
> > [11] Tom Tirer and Joan Bruna. Extended unconstrained features model for exploring deep neural collapse. In ICML, 2022.
> >
> > [12] Tom Tirer, Haoxiang Huang, and Jonathan Niles-Weed. Perturbation analysis of neural collapse. In ICML, 2023.
> >
> > [13] Jiachen Jiang, Jinxin Zhou, Peng Wang, Qing Qu, Dustin G Mixon, Chong You, and Zhihui Zhu. Generalized neural collapse for a large number of classes. In Conference on Parsimony and Learning, 2023.
> >
> > [14] Cong Fang, Hangfeng He, Qi Long, and Weijie J Su. Exploring deep neural networks via layer-peeled model: Minority collapse in imbalanced training. In Proceedings of the National Academy of Sciences, 2021.
> >
> > [15] Christos Thrampoulidis, Ganesh Ramachandra Kini, Vala Vakilian, and Tina Behnia. Imbalance trouble: Revisiting neural-collapse geometry. In Conference on Neural Information Processing Systems, 2022.
> >
> > [16] Wanli Hong and Shuyang Ling. Neural collapse for unconstrained feature model under cross-entropy loss with imbalanced data. JMLR, 2024.
> >
> > [17]  Hien Dang, Tho Tran Huu, Tan Minh Nguyen, and Nhat Ho. Neural collapse for cross-entropy class-imbalanced learning with unconstrained relu features model. In ICML, 2024.
> >
> > [18]  Peter Súkeník, Marco Mondelli, and Christoph H. Lampert. Neural collapse versus low-rank bias: Is deep neural collapse really optimal? Conference on Neural Information Processing Systems, 2024.
> >
> > [19]  Connall Garrod and Jonathan P Keating. The persistence of neural collapse despite low-rank bias: An analytic perspective through unconstrained features. arXiv preprint, 2024.
> >
> > [20] George Andriopoulos, Zixuan Dong, Li Guo, Zifan Zhao, and Keith Ross. The prevalence of neural collapse in neural multivariate regression. In Conference on Neural Information Processing Systems, 2024.
> >
> > [21] Vignesh Kothapalli, Tom Tirer, and Joan Bruna. A neural collapse perspective on feature evolution in graph neural networks. In Conference on Neural Information Processing Systems, 2023.
> >
> > [22] Christos Thrampoulidis. Implicit optimization bias of next-token prediction in linear models. In Conference on Neural Information Processing Systems, 2024.
> >
> > [23] Yize Zhao, Tina Behnia, Vala Vakilian, and Christos Thrampoulidis. Implicit geometry of next-token prediction: From language sparsity patterns to model representations. In First Conference on Language Modeling, 2024.
> >
> > [24] Arthur Jacot, Franck Gabriel and Clement Hongler. Neural tangent kernel: Convergence and generalization in neural networks. In Advances in neural information processing systems, 2018.
> >
> > [25] Anna Choromanska, MIkael Henaff, Michael Mathieu, Gerard Ben Arous and Yann Le Cun. The Loss Surfaces of Multilayer Networks. In Proceedings of the Eighteenth International Conference on Artificial Intelligence and Statistics, 2015.
> >
> > [26] Andrew M. Saxe, James L. McClelland, and Surya Ganguli. Exact solutions to the nonlinear dynamics of learning in deep linear neural networks. ICLR, 2014.

---

> ### Author Response · Authors · 2025-11-25
> **Summary of Revisions in Response to Reviewer Comments**
>
> As an update, we have now uploaded a revised version of the paper. To reduce your reading burden, we summarize below the changes made in direct response to your comments. Thank you again for the time and effort you have dedicated to evaluating our work.
>
> **Motivating the UFM:**
>
> We have provided a new Appendix A to further justify the use of the UFM as a model for overparameterized neural networks. In particular, following a discussion with another reviewer, we now reference recent results by Sukenik et al. [1], which establish a correspondence between UFMs and neural networks in the depth limit for specific architectural designs. We hope this strengthens the theoretical foundation of our modeling approach and helps address your concerns.
>
> **Further highlighting empirical results:**
>
> We have also moved a portion of the empirical results into the main text to more clearly emphasize the strong correspondence between the deep UFM and real neural networks on canonical datasets.
>
> We would greatly appreciate any additional feedback you may have on these revisions, as well as your overall thoughts on our efforts to address the concerns you raised. In particular, please let us know if you feel it would be helpful to add to Appendix A the discussion of the degenerate nature of the UFM and the prevalence of such degeneracies in the implicit-bias literature.
>
>
> [1]  Peter Súkeník, Christoph H. Lampert, Marco Mondelli. Neural Collapse is Globally Optimal in Deep Regularized ResNets and Transformers. ArXiv preprint, 2025.

---

> ### Author Response · Authors · 2025-12-01
>
> Dear Area Chair,
>
> Given that the reviewer is unable to provide guidance on whether a discussion of the degenerate nature of the UFM should be included, we have elected to add such a section to Appendix A. The full content of this new section is provided below.
>
> **Why the UFM Can Exactly Fit the Data**
>
> A defining feature of the UFM is that its feature map is assumed to be arbitrarily expressive. Consequently, the model can, in principle, map each input directly to its corresponding label and achieve zero training error. At first glance this might seem problematic. Below, we explain why this property is intentional and foundational to the design of the UFM.
>
> **Modelling the Overparameterized Regime:** The central purpose of the UFM is to isolate and study the geometric structures that emerge in the limit of extreme overparameterization. In such regimes, real neural networks possess enough capacity to fit the training data exactly, and additional layers or parameters contribute progressively less to reducing the loss once interpolation is achievable.
>
> The UFM captures this limiting behavior by assuming a feature map flexible enough to represent the data perfectly. This is not a flaw but an explicit modelling choice: it reflects the empirical fact that modern networks often operate far beyond the interpolation threshold. By doing so, the UFM allows us to analyze the organization of learned representations when capacity is not a limiting factor, which is precisely the setting emphasized in the original Neural Collapse (NC) work of Papyan et al. [1] and numerous other empirical studies [2,3,4]. This controlled abstraction enables a clean examination of the geometric consequences of overparameterization without confounding details.
>
> **Exact Fitting as a Standard Setting for Studying Implicit Bias:** The assumption that the model can fit the data exactly is also firmly rooted in a long tradition of theoretical work on implicit bias in optimization and deep learning. Many influential papers deliberately study training dynamics in interpolation settings—settings where the loss can be driven to zero—to reveal how gradient-based methods or depth implicitly favor certain solutions.
>
> To highlight just a few canonical examples:
>
> * Saxe et al. [5] analyze how depth and gradient flow induce the sequential emergence of features in linear networks trained with MSE.
> * Soudry et al. [6] and Lyu & Li [7] show that gradient descent on losses with exponential tails implicitly maximizes the margin in linearly separable classification.
> * Arora et al. [8] study deep matrix factorization and demonstrate that depth induces low-rank biases that improve generalization.
> * Gunasekar et al. [9] characterize how different optimization methods lead to different implicit biases in separable linear classification problems.
>
> For a survey of this line of work, see the review by Gal Vardi [10].
>
> The UFM should be understood as operating squarely within this tradition. NC and DNC manifest across many architectures and datasets, yet common loss functions contain no explicit term encouraging simplex equiangular tight frames or orthogonal class means. Their emergence instead arises as a consequence of implicit bias induced by overparameterization. The UFM provides an analytically tractable model for studying this bias in the limit of high overparameterization, and its correspondence is supported by extensive empirical evidence summarized previously in this appendix.

---

> > ### Author Response · Authors · 2025-12-01
> > **References**
> >
> > [1] Vardan Papyan, Xuemei Han, and David L. Donoho. Prevalence of neural collapse during the terminal phase of deep learning training. In Proceedings of the National Academy of Sciences of the United States of America, 2020.
> >
> > [2] Vardan Papyan. Traces of class/cross-class structure pervade deep learning spectra. In Journal of Machine Learning Research, 2020.
> >
> > [3] Vardan Papyan. Measurements of three-level hierarchical structure in the outliers in the spectrum of deep net Hessians. ArXiv preprint,2019
> >
> > [4] Vardan Papyan. The Full Spectrum of Deep Net Hessians At Scale: Dynamics with Sample Size, ArXiv preprint, 2018.
> >
> > [5] Andrew M. Saxe, James L. McClelland, and Surya Ganguli. Exact solutions to the nonlinear dynamics of learning in deep linear neural networks. ICLR, 2014.
> >
> > [6] Daniel Soudry, Elad Hoffer, Mor Shpigel Nacson, Suriya Gunasekar and Nathan Srebro. The implicit bias of gradient descent on separable data. Journal of Machine Learning Research, 2018.
> >
> > [7] Kaifeng Lyu and Jian Li. Gradient Descent Maximizes the Margin of Homogeneous Neural Networks. International Conference on Learning Representations, 2019.
> >
> > [8] Sanjeev Arora, Nadav Cohen, Wei Hu and Yuping Luo. Implicit regularization in deep matrix factorization. Advances in neural information processing systems, 2019.
> >
> > [9] Suriya Gunasekar, Jason Lee, Daniel Soudryl and Nathan Srebro. Characterizing implicit bias in terms of optimization geometry. International Conference on Machine Learning. 2018.
> >
> > [10] Gal Vardi. On the implicit bias in deep-learning algorithms. Communications of the ACM, 2023.

---

### Author Response · Authors · 2025-11-25
**Revised Submission: Summary of Changes**

Dear Reviewers,

Thank you for your thoughtful feedback and for the helpful suggestions on how to improve our submission. We have carefully incorporated your comments and substantially expanded and clarified the manuscript. Below is a summary of the key changes in the revised version:

* **Title:** Updated the paper title to “Unifying Low Dimensional Spectra in Deep Learning.” We will update the OpenReview title upon confirmation of the appropriate procedure.
* **Introduction:** Added expanded context from Papyan et al. [1] to better situate our work within existing literature.
* Introduced an explicit Contributions subsection to clearly articulate the main advances of our paper.
* Added a Significance subsection to highlight the broader value and implications of our results.
* Expanded the discussion of prior random matrix theory (RMT) approaches used to study Hessians in deep learning.
* **Background:** Adapted the definition of DNC3 and provided explicit definitions for normalized vectors to improve clarity.
* **Theoretical Results:** Added a summary at the beginning of Section 3 outlining the purpose and contribution of each subsection.
* Refined Theorem 6 and its discussion to clarify the assumptions underlying the theory and to explain complexities introduced by specific feature map architectures.
* Revised commentary related to the derivative of ReLU to make the modelling assumptions more explicit.
* **Numerical Experiments:** Moved experiments verifying eigenvectors of layer-wise Hessians and the associated gradient alignment (ResNet-20 on MNIST and CIFAR-10) into the main text for improved accessibility and emphasis.
* **Conclusion:** Expanded concluding remarks to more clearly synthesize our findings and articulate our contributions.
* **Appendix A: Justifying the UFM as a Model of Overparameterization:** Added an extensive appendix documenting prior empirical validation of UFM predictions.
* Included detailed justification of modelling assumptions.
* Added discussion of recent theoretical support from Sukenik et al. [2], demonstrating that the UFM captures real-network behavior for certain architectures.

We hope these revisions meaningfully improve the clarity, rigor, and impact of the paper. We sincerely appreciate the reviewers’ constructive comments and the time taken to evaluate our work. We plan to incorporate the suggested additional experiments at a later date.

Thank you again for your consideration.

**References**

[1] Vardan Papyan, Xuemei Han, and David L. Donoho. Prevalence of neural collapse during the terminal phase of deep learning training. Proceedings of the National Academy of Sciences of the United States of America, 2020.

[2] Peter Súkeník, Christoph H. Lampert, Marco Mondelli. Neural Collapse is Globally Optimal in Deep Regularized ResNets and Transformers. ArXiv preprint, 2025.

---

> ### Author Response · Authors · 2025-12-01
> **Second Revision**
>
> We have since prepared a second revision that includes the following additions:
>
> Added Appendix A.3, which provides justification for the “degenerate nature” of the UFM as noted by Reviewer LCvk.
>
> Included additional experiments (Figures 16–18) in response to the requests from Reviewer 9iP3 and Reviewer SQKm.

---

### Author Response · Authors · 2025-12-01
**Summary of Discussions**

Dear Area Chair,

We would like to provide a concise summary of the outcomes of the discussions with the reviewers to aid in your evaluation of our work. Following the initial reviews and subsequent exchanges, we undertook extensive revisions to address all concerns raised.

With Reviewers 9iP3 and BQ76, the discussions were exceptionally constructive. Through detailed clarification and technical engagement, we were able to demonstrate the correctness of our approach on several key issues and reach shared agreement on others. Both reviewers explicitly praised our engagement during the discussion phase and indicated that they would raise their evaluation scores.

Although we did not receive follow-up responses from the remaining reviewers before the change in review process, we carefully addressed each of their points in our revision:

**Reviewer LCvk:** Their sole criticism concerned insufficient explanation of the UFM and the rationale for its degenerate nature. In response, we added a new appendix (Appendix A) that provides (i) a detailed account of extensive empirical evidence from prior work, (ii) new theoretical support for the model, (iii) clear motivation for the underlying assumptions, and (iv) a concise explanation of why the UFM’s degeneracy is an intentional modelling choice aligned with a substantial body of research on implicit bias in deep learning.


**Reviewer SQKm:** the majority of their concerns stemmed from an earlier ambiguous title, insufficiently highlighted key results, and the placement of empirical validations in the appendices. We have since addressed each of their concerns, including: revising the title for clarity, emphasizing our key contributions more prominently in the introduction and main text, and moving the most important empirical validations into the main body.

We hope that the improvements to the paper generated by productive discussions with reviewers are accounted for in your evaluations of our work.

---

### Meta-Review · Area_Chair_1YR2 · 2025-12-30

**Summary:**

This paper studies the low-dimensional structures emerging in the spectra of Hessians (and also gradients and weights matrices) connecting them to the deep neural collapse, which is argued as a unifying explanation. The theoretical results concern unconstrained feature models (UFMs), which are commonly considered in the neural collapse literature, and some of the numerical evidence is obtained with more practical deep neural networks. More precisely, the authors start with the deep linear UFM analyzing Hessian spectra, gradient alignment with outlier eigenspace and weight matrices, and they then move to the deep ReLU UFM.

The reviewers have raised a number of objections both to the overall narrative and to certain specific conclusions, and there has been quite a lengthy discussion before the rebuttal period had to come to an abrupt conclusion. While the initial reviewers were negative, upon reading in detail the exchanges and looking at the revision, my opinion is that the authors have been able to address at least part of the concerns raised by the reviewers. This makes the paper, in its current state, borderline. However, even if the reviewers had been able to participate fully in the discussion, I find it rather unlikely that a consensus would have been reached towards accepting the paper. The manuscript has already undergone rather significant changes (some of which are structural, in terms of the narrative and the positioning of the work) and, based on my reading, I think that the paper would still need to undergo another round of review before it can be accepted.

For these reasons, I recommend a rejection of the paper in its current state and I encourage the authors to resubmit a revised version to a future venue.

**Reviewer Concerns:**

Reviewers LCvk and SQKm: the concerns here are rather structural and they concerns the overall narrative, positioning and significance of the results of the paper. The authors have done significant work to address them, both in their rebuttal and in the revision. However, it is not fully clear whether such issues have been resolved.

Reviewers 9iP3 and BQ76: most of the issues have been solved here. However, one notable exception is the discussion on Theorem 6 and the progressive neural collapse concern of reviewer BQ76, which remain outstanding.

**Reviewer Scores:**

Reviewer BQ76 had already raised the score to 6 and, while I cannot say this for sure, I would find it fair if Reviewer 9iP3 had done the same, given the tone of the exchange. However, I find it unlikely that either reviewer would have championed the paper expressing a more clear acceptance in the subsequent discussion. Thus, taking into account all 4 reviews, I find it unlikely that there would have been an overall recommendation towards accepting the paper.

---

### Decision · Program_Chairs · 2026-01-26

Reject